# ENHANCING THE SCALABILITY AND APPLICABILITY OF KOHN-SHAM HAMILTONIANS FOR MOLECULAR SYSTEMS

**Yunyang Li**♠∗  **Zaishuo Xia**♡∗  **Lin Huang**♣✉∗  **Xinran Wei**♣  **Han Yang**♣  **Sam Harshe**♠
**Zun Wang**♣  **Chang Liu**♣  **Jia Zhang**♣✉  **Bin Shao**♣  **Mark B. Gerstein**♠✉
✉{huang.lin, jia.zhang}@microsoft.com  mark@gersteinlab.org
♠Yale University  ♡UC Davis  ♣MSR AI4Science

## ABSTRACT

Density Functional Theory (DFT) is a pivotal method within quantum chemistry and materials science, with its core involving the construction and solution of the Kohn-Sham Hamiltonian. Despite its importance, the application of DFT is frequently limited by the substantial computational resources required to construct the Kohn-Sham Hamiltonian. In response to these limitations, current research has employed deep-learning models to efficiently predict molecular and solid Hamiltonians, with roto-translational symmetries encoded in their neural networks. However, the scalability of prior models may be problematic when applied to large molecules, resulting in non-physical predictions of ground-state properties. In this study, we generate a substantially larger training set (PubChemQH) than used previously and use it to create a scalable model for DFT calculations with physical accuracy. For our model, we introduce a loss function derived from physical principles, which we call Wavefunction Alignment Loss (WALoss). WALoss involves performing a basis change on the predicted Hamiltonian to align it with the observed one; thus, the resulting differences can serve as a surrogate for orbital energy differences, allowing models to make better predictions for molecular orbitals and total energies than previously possible. WALoss also substantially accelerates self-consistent-field (SCF) DFT calculations. Here, we show it achieves a reduction in total energy prediction error by a factor of 1347 and an SCF calculation speed-up by a factor of 18%. These substantial improvements set new benchmarks for achieving accurate and applicable predictions in larger molecular systems.

## 1 INTRODUCTION

Density functional theory (DFT) (Kohn & Sham, 1965; Hohenberg & Kohn, 1964; Martin, 2020) has been widely used in physics (Argaman & Makov, 2000; Jones, 2015; Van Mourik et al., 2014), chemistry (Levine et al., 2009; Van Mourik et al., 2014), and materials science (March, 1999; Neugebauer & Hickel, 2013) to study the electronic properties of molecules and solids. This methodology is particularly valued for its balanced blend of computational efficiency and accuracy, rendering it a versatile choice for investigating electronic structure (Kohn et al., 1996; Parr & Yang, 1995), spectroscopy (Neese, 2009; Orio et al., 2009), lattice dynamics (Dal Corso et al., 1997; Wang et al., 2021), transport properties (Bhamu et al., 2018), and more. The most critical step in applying DFT to a molecule is constructing the Kohn-Sham Hamiltonian, which consists of the kinetic operator, the external potential, the Coulomb potential (also known as the Hartree potential), and the exchange-correlation potential (Kohn & Sham, 1965; Hohenberg & Kohn, 1964; Martin, 2020).

The Hamiltonian matrix contains crucial information about molecular systems and their quantum states (Kohn & Sham, 1965; Hohenberg & Kohn, 1964; Yu et al., 2023b; Zhang et al., 2024). This matrix facilitates the extraction of various properties, including the highest occupied molecular orbital (HOMO) and lowest unoccupied molecular orbital (LUMO) energies, the HOMO-LUMO gap, total

---

∗Equal contribution.

energy, and spectral characteristics (Eisberg & Resnick, 1985; Zhang et al., 2024). These properties are vital for analyzing conformational energies (St.-Amant et al., 1995), reaction pathways (Farberow et al., 2014), and vibrational frequencies (Watson & Hirst, 2002). However, the efficiency of DFT is constrained by the self-consistent field (SCF) iterations required to solve the Kohn-Sham equations and achieve consistent charge density. These iterations scale as $O(N^3T) \sim O(N^4T)$ (Tirado-Rives & Jorgensen, 2008; Yu et al., 2023a), where $N$ represents the number of electrons and $T$ denotes the number of SCF cycles, making them particularly resource-intensive for large systems.

Consequently, this computational intensity underscores the critical need to develop more efficient methodologies to determining the Hamiltonian without relying on SCF iterations. This challenge has catalyzed interest in leveraging deep learning to predict Hamiltonians directly from atomic configurations while adhering to the inherent symmetries of molecular systems (Unke et al., 2021a; Kochkov et al., 2021; Yu et al., 2023b; Li et al., 2022; Schütt et al., 2019; Gastegger et al., 2020; Hermann et al., 2020; Yin et al., 2024; Zhang et al., 2024; Luo et al., 2025). Notably, the Hamiltonian matrix is subject to unitary transformations under molecular rotations due to its spherical harmonics component. To tackle these transformations, researchers have developed SE(3)-equivariant neural networks such as PhisNet (Unke et al., 2021b) and QHNet (Yu et al., 2023b). These networks employ high-order spherical tensors and the Clebsch-Gordon tensor product to construct the predicted Hamiltonians.

However, current implementations of SE(3)-equivariant neural networks face considerable scalability challenges when applied to large molecular structures. Notably, prevailing techniques for predicting the Hamiltonian matrix predominantly utilize mean absolute error (MAE) or mean square error (MSE) as the loss function (Unke et al., 2021b; Yu et al., 2023b). We contend that these *elementwise losses alone are insufficient* for accurately learning Hamiltonians in large systems. To underscore this point, we introduce the PubChemQH dataset, which comprises molecular Hamiltonians for structures with atom counts ranging from 40 to 100. In contrast, the previously curated dataset (Yu et al., 2023a) is limited to molecules with no more than 31 atoms. Figure 1 demonstrates a phenomenon enabled by the introduction of the PubChemQH dataset, which we term *Scaling-Induced MAE-Applicability Divergence* (SAD). The applicability of the Hamiltonian is assessed by evaluating the system energy error derived from `pyscf` (Sun et al., 2020; Sun, 2015; Sun et al., 2018). While small molecules yield system energies close to the ground truth with relative Hamiltonian MAEs up to 200%, larger molecules exhibit profound inaccuracies. Energy errors escalate to as much as 1,000,000 kcal/mol at only 0.01% relative MAE, rendering the Hamiltonians inapplicable. This striking disparity emphasizes the inadequacy of MAE for large systems and highlights the pressing need for new methodologies to enhance the scalability[1] and applicability of predicted Hamiltonians.

To address these limitations, this work aims to enhance Hamiltonian learning for large systems by utilizing wavefunctions and their corresponding energies as surrogates to improve Hamiltonian applicability. The wavefunction is crucial for Hamiltonian prediction as it encapsulates the quantum state of a system, enabling the validation of Hamiltonians based on their ability to accurately reflect the system's energy and dynamics. However, learning the electron wavefunction is nontrivial; inaccuracies in the machine learning-based Hamiltonian may lead to significant error in both electronic structure and wavefunctions. This challenge motivated us to introduce the *Wavefunction Alignment Loss* function (WALoss), which aligns the eigenspaces of predicted and ground-truth Hamiltonians without explicit backpropagation through eigensolvers. Additionally, to improve the scalability of Hamiltonian learning, we present *WANet*, a modernized architecture for Hamiltonian prediction that leverages eSCN (Passaro & Zitnick, 2023) convolution and a sparse mixture of pair experts. Our pipeline and main contributions are summarized in Figure 2.

## 2 BACKGROUND

In the framework of Density Functional Theory (DFT), a molecular system is defined by its nuclear configuration $\mathcal{M} := \{\mathbf{Z}, \mathbf{R}\}$, where $\mathbf{Z}$ represents the atomic numbers of the nuclei and $\mathbf{R}$ their positions within the molecule. DFT focuses on determining the ground state of a system consisting of $N$ electrons by minimizing the total electronic energy with respect to the electron density $\rho(\mathbf{r})$. Here, $\rho(\mathbf{r})$ is a functional of the set of $N$ one-electron orbitals $\{\psi_i(\mathbf{r})\}_{i=1}^N$, where $\mathbf{r} \in \mathbb{R}^3$ specifies the spatial coordinates of an electron.

---

[1]We define the scalability as the model's accuracy under large molecules.

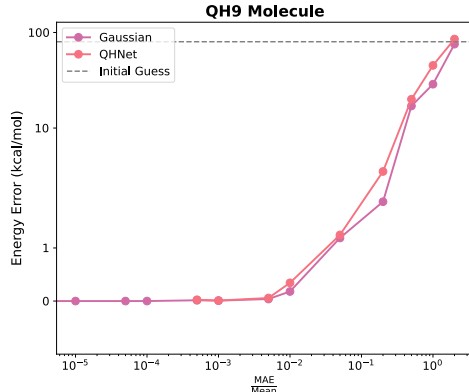 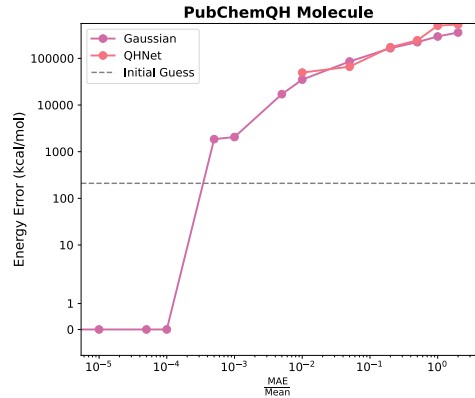

Figure 1: Visualization of the SAD phenomenon. The y-axis (in symmetrical log scale (Webber, 2012)) represents the system energy error derived from the perturbed Hamiltonian, while the x-axis shows the relative MAE, defined as $\frac{\mathrm{MAE}(\hat{\mathbf{H}}, \mathbf{H}^\star)}{\mathrm{Mean}(\mathbf{H}^\star)}$, where $\mathbf{H}^\star$ represents the ground-truth Hamiltonian and $\hat{\mathbf{H}}$ denotes the predicted or perturbed Hamiltonian. The relative MAE is induced by model learning errors or Gaussian perturbation. The Gaussian perturbation ensures that the perturbed matrix remains Hermitian. The initial guess line is derived from `minao` (Almlöf et al., 1982; Van Lenthe et al., 2006). Current state-of-the-art models, such as QHNet, achieve a relative MAE of up to $10^{-2}$ on PubChemQH molecules. For small molecules, a $10^{-2}$ relative MAE is sufficient for accurate system energy predictions (left panel), but this accuracy does not extend to large systems (right panel).

For computational efficiency, these orbitals are represented using a basis set $\{\phi_\alpha(\mathbf{r})\}_{\alpha=1}^B$ depending on the molecular geometry, $B$ denotes the number of basis and varies for different basis sets. The expansion coefficients of the orbitals are organized in a matrix $\mathbf{C} \in \mathbb{R}^{B \times N}$, allowing each orbital $\psi_i(\mathbf{r})$ to be expressed as: $\psi_i(\mathbf{r}) = \sum_{\alpha=1}^B C_{\alpha i} \phi_\alpha(\mathbf{r})$. DFT seeks the minimal electronic energy by solving for the optimal coefficient matrix $\mathbf{C}$ via the Kohn-Sham equations, represented as $\mathbf{H}(\mathbf{C})\mathbf{C} = \mathbf{SC}\boldsymbol{\epsilon}$, where $\mathbf{H}(\mathbf{C}) \in \mathbb{R}^{B \times B}$ denotes the Hamiltonian matrix. Notice that $\mathbf{H}$ is a function of $\mathbf{C}$, and can be computed using *density fitting* with a complexity of $O(B^3)$. $\mathbf{S} \in \mathbb{R}^{B \times B}$ is the overlap matrix with elements $S_{\alpha\beta} := \int \phi_\alpha^\dagger(\mathbf{r})\phi_\beta(\mathbf{r})\mathrm{d}\mathbf{r}$, where $\phi^\dagger$ denotes the complex conjugate of $\phi$. $\boldsymbol{\epsilon}$ represents a diagonal matrix containing the orbital energies. This forms a generalized eigenvalue problem, where $\mathbf{C}$ comprises the eigenvectors and the diagonal elements of $\boldsymbol{\epsilon}$ are the eigenvalues. However, the solution to this problem is complicated by the interdependence between $\mathbf{H}(\mathbf{C})$ and $\mathbf{C}$. To resolve this, traditional DFT employs the self-consistent field (SCF) method, an iterative process that refines the coefficients $\mathbf{C}^{(k)}$ through successive approximations of the Hamiltonian matrix $\mathbf{H}^{(k)}$. Each iteration commences by computing $\mathbf{H}^{(k)}$ leveraging $\mathbf{C}^{(k-1)}$ and solves a new eigenvalue problem:

$$\mathbf{H}^{(k)}\left(\mathbf{C}^{(k-1)}\right)\mathbf{C}^{(k)} = \mathbf{SC}^{(k)}\boldsymbol{\epsilon}^{(k)},$$

converging to a stable Hamiltonian $\mathbf{H}^\star$ and its corresponding eigenvectors $\mathbf{C}^\star$, which define the electron density and other molecular properties derived from the Kohn-Sham equations.

**Problem Formulation** The objective of Hamiltonian prediction is to obviate the need for self-consistent field (SCF) iteration by directly estimating the target Hamiltonian $\mathbf{H}^\star$ from a given molecular structure $\mathcal{M}$. To achieve this, one could parameterize a machine learning model $\hat{\mathbf{H}}_\theta(\mathcal{M})$. The learning process is guided by an optimization process defined as:

$$\theta^\star = \arg\min_\theta \frac{1}{|\mathcal{D}|} \sum_{(\mathcal{M}, \mathbf{H}^\star_\mathcal{M}) \in \mathcal{D}} \mathrm{dist}\left(\hat{\mathbf{H}}_\theta(\mathcal{M}), \mathbf{H}^\star_\mathcal{M}\right), \tag{1}$$

where $|\mathcal{D}|$ denotes the cardinality of dataset $\mathcal{D}$, and $\mathrm{dist}(\cdot, \cdot)$ is a predefined distance metric.

**SE(3) Equivariant Networks** SE(3)-equivariant neural networks incorporate strong prior knowledge through equivariance. These networks utilize equivariant irreducible representation (irreps) features built from vector spaces of irreducible representations to achieve 3D rotation equivariance. The vector spaces are $(2\ell+1)$-dimensional, where degree $\ell \in \mathbb{N}$ represents the angular frequency of the vectors. Higher $\ell$ values are critical for tasks sensitive to angular information, such as predicting the Hamiltonian matrix. Vectors of degree $\ell$, referred to as type-$\ell$ vectors, are rotated using Wigner-D matrices $D^{(\ell)} \in \mathbb{R}^{(2\ell+1) \times (2\ell+1)}$ when rotating coordinate systems. Eigenfunctions of rotation in $\mathbb{R}^3$ can be projected into type-$\ell$ vectors using the real spherical harmonics $Y^{(\ell)} : \mathbb{S}^2 \to \mathbb{R}^{2\ell+1}$, where $\mathbb{S}^2 = \{\hat{\mathbf{r}} \in \mathbb{R}^3 : \|\hat{\mathbf{r}}\| = 1\}$ is the unit sphere. Equivariant GNNs update irreps features through message passing of transformed irreps features between nodes, using tensor prod-

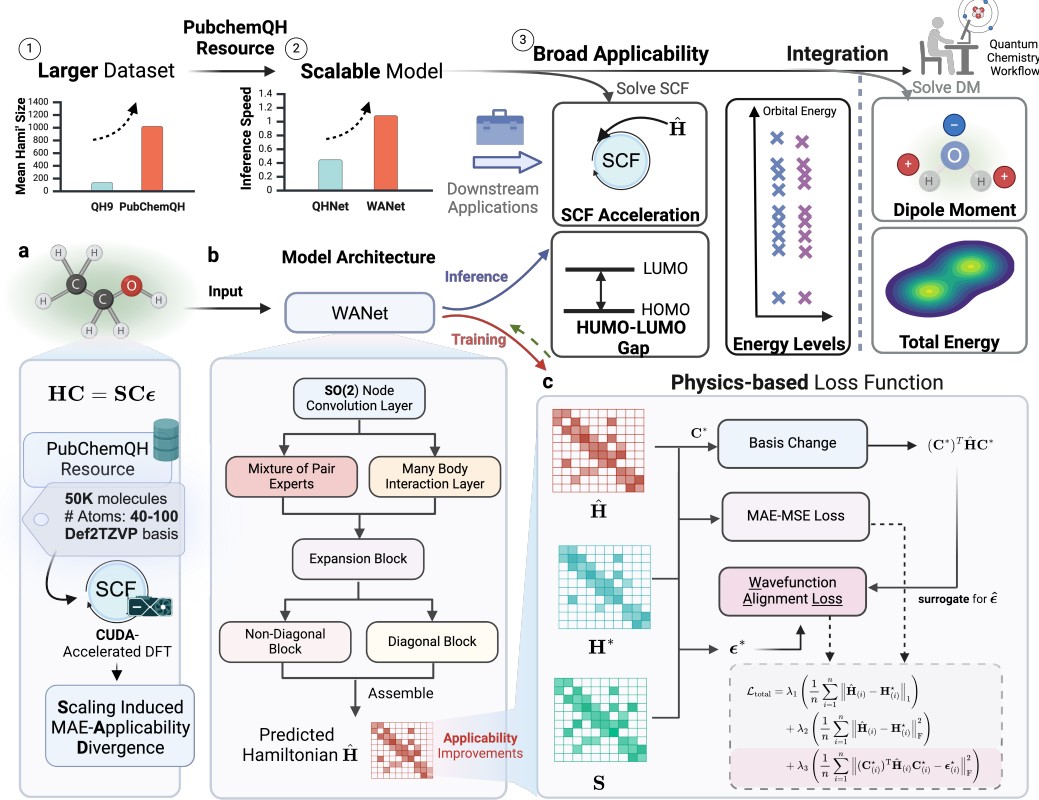

Figure 2: (a) We introduce PubChemQH, a new resource for Hamiltonian learning that facilitates the exploration of the scalability challenge known as SAD. (b) We present WANet, a modern architecture designed for accurate Hamiltonian prediction. WANet incorporates $SO(2)$ convolution, a mixture of pair experts, and a many-body interaction layer. The mixture of pair experts constructs the non-diagonal block, while the many-body interaction layer constructs the diagonal block. (c) Our loss module, WALoss, performs a basis transformation of the predicted Hamiltonian using the ground-truth Hamiltonian and the overlap matrix. This enhancement aims to improve the applicability of the predicted Hamiltonian in real-world scenarios. Our final loss function combines MAE-MSE loss with WALoss.

ucts: $(u^{\ell_1} \otimes v^{\ell_2})^{\ell_3}_{m_3} = \sum_{m_1=-\ell_1}^{\ell_1} \sum_{m_2=-\ell_2}^{\ell_2} C^{(\ell_3,m_3)}_{(\ell_1,m_1),(\ell_2,m_2)} u^{\ell_1}_{m_1} v^{\ell_2}_{m_2}$, where $C$ denotes the Clebsch-Gordan coefficient, $\ell_3$ satisfies $|\ell_1 - \ell_2| \leq \ell_3 \leq \ell_1 + \ell_2$. Note that $m$ denotes the $m$-th element in the irreducible representation with $-\ell \leq m \leq \ell$ and $m \in \mathbb{N}$. In equivariant graph neural networks, the message function $m_{ts}$ from source node $s$ to target node $t$ is calculated using $SO(3)$ convolution. The $\ell_0$-th degree of $m_{ts}$ can be expressed as $m^{(\ell_o)}_{ts} = \sum_{\ell_i,\ell_f} W_{\ell_i,\ell_f,\ell_o} \left( x^{(\ell_i)}_s \otimes Y^{(\ell_f)}(\hat{\mathbf{r}}_{ts}) \right)^{\ell_o}$, where $W_{\ell_i,\ell_f,\ell_o}$ are the weight vectors, $x_s$ represents the irreducible representation of the source node $s$ and $\hat{\mathbf{r}}_{ts} := \frac{\mathbf{r}_{ts}}{\|\mathbf{r}_{ts}\|_2}$.

# 3 WAVEFUNCTION-ALIGNMENT LOSS

The applicability of the predicted Hamiltonian $\hat{\mathbf{H}}$ depends significantly on the accuracy of its eigenvalues (orbital energies) and eigenvectors (basis coefficients). To assess the alignment between $\hat{\mathbf{H}}$ and the actual Hamiltonian $\mathbf{H}^\star$ with respect to their eigenspaces, we present the following theorem:

**Theorem 1.** *Let $\mathbf{H}$ and $\hat{\mathbf{H}}$ represent Hamiltonian matrices, and $\mathbf{S}$ the overlap matrix. Define the perturbation matrix as $\Delta\mathbf{H} := \hat{\mathbf{H}} - \mathbf{H}$. Let $\lambda_i(\mathbf{H}, \mathbf{S})$ and $\lambda_i(\hat{\mathbf{H}}, \mathbf{S})$ be the $i^{th}$ generalized eigenvalues of $\mathbf{H}$ and $\hat{\mathbf{H}}$, respectively. Assume a spectral gap $\delta$ separates the generalized eigenvalues of $\mathbf{H}, \hat{\mathbf{H}}$. $\kappa(\cdot)$ denotes the conditional number of a given matrix, $\|\cdot\|_2$ represents the spectral norm. $\|\Delta\mathbf{H}\|_{1,1} = \sum_{i,j} |\Delta\mathbf{H}_{ij}|$. Then, the difference in eigenvalues and the angle $\theta$ between the eigenspace of $\mathbf{H}$ and $\hat{\mathbf{H}}$ satisfy:*

$$|\lambda_i(\hat{\mathbf{H}}, \mathbf{S}) - \lambda_i(\mathbf{H}, \mathbf{S})| \leq \frac{\kappa(\mathbf{S})}{\|\mathbf{S}\|_2} \cdot \|\Delta\mathbf{H}\|_{1,1}, \quad \sin\theta \leq \frac{\kappa(\mathbf{S})}{\|\mathbf{S}\|_2} \cdot \frac{\|\Delta\mathbf{H}\|_{1,1}}{\delta}.$$

*Proof.* The proof is delegated to Appendix J.1. □

**Corollary 1** (Perturbation Sensitivity Scaling). *Assume* $\lambda_{min}(S)$ *of the overlap matrix $S$ scales as:* $\lambda_{min}(S) = c + \dfrac{A}{1 + \left(\dfrac{B}{N_0}\right)^\alpha}$, *where $c > 0$, $A > 0$, and $\alpha > 0$ describe the saturation behavior. Then, for a perturbation matrix $\Delta H$, under the first-order perturbation approximation, the eigenvalue perturbation is bounded by:*

$$\left| \lambda_i(\hat{\mathbf{H}}, \mathbf{S}) - \lambda_i(\mathbf{H}, \mathbf{S}) \right| \leq O\left( \sigma B^{1/2} + B|\mu| \right),$$

*Proof.* The proof is delegated to Appendix J.2. □

**Remark** The theorem highlights that the difference between the predicted and actual Hamiltonian matrices, when only considering the element-wise norm, can lead to unbounded differences in eigenvalues/eigenvectors due to a significant $\frac{\kappa(\mathbf{S})}{\|\mathbf{S}\|}$ ratio. The corollary further elucidates the sensitivity of eigenvalue perturbations to the size of the basis $B$. This phenomenon underscores the catastrophic scaling associated with increasing $B$, which is a manifestation of the aforementioned SAD phenomenon. We provide a thorough discussion on $\alpha$ in Appendix J.3. To validate our theoretical analysis, we empirically evaluated the distribution of $\frac{\kappa(\mathbf{S})}{\|\mathbf{S}\|_2}$ across both the QH9 and PubChemQH datasets (Figure 6). The results demonstrate that molecules in PubChemQH exhibit substantially higher ratios compared to QH9, indicating increased perturbation sensitivity in larger molecular systems. This provides strong empirical evidence for the SAD phenomenon and aligns with our theoretical predictions. Consequently, these findings suggest that when designing an effective supervisory signal for learning or optimization tasks involving Hamiltonian matrices, it is crucial to take into account the interaction of the overlap matrix and the corresponding Hamiltonian to mitigate the potential instability caused by perturbations.

In light of this, we propose a new loss function: the Wavefunction Alignment Loss. It is designed to preserve the integrity of the eigenstructure related to molecular orbitals. Let $\hat{\boldsymbol{\epsilon}}$ and $\boldsymbol{\epsilon}^\star$ represent the eigenvalues (orbital energies) of $\hat{\mathbf{H}}$ and $\mathbf{H}^\star$, respectively, and $\hat{\mathbf{C}}$ and $\mathbf{C}^\star$ denote the corresponding eigenvectors (basis coefficients). We define a primary form of the loss function by directly applying the Frobenius norm to the eigenvalues $\mathcal{L}_{\text{align}} = \frac{1}{n} \sum_{i=1}^{n} \|\hat{\boldsymbol{\epsilon}}_{(i)} - \boldsymbol{\epsilon}^\star_{(i)}\|_F^2$, where $\hat{\boldsymbol{\epsilon}}_{(i)}$ and $\boldsymbol{\epsilon}^\star_{(i)}$ are derived from solving the generalized eigenvalue problems: $\hat{\mathbf{H}}\hat{\mathbf{C}} = \mathbf{S}\hat{\mathbf{C}}\hat{\boldsymbol{\epsilon}}$ and $\mathbf{H}^\star\mathbf{C}^\star = \mathbf{S}\mathbf{C}^\star\boldsymbol{\epsilon}^\star$, respectively.

However, this formulation has notable limitations: (1) Generalized eigenvalue problems are susceptible to numerical instabilities due to ill-conditioned matrices, leading to erroneous gradients during backpropagation through iterative eigensolvers, complicating optimization. (2) The loss function assigns uniform weights to all orbital energies, which is not practical as some orbital energies hold more significance. To address these issues, we begin by applying the following algorithm (Ghojogh et al., 2019; Golub & Van Loan, 2013) to perform a simultaneous reduction of the matrix pair $(\mathbf{H}^\star, \mathbf{S})$:

---

**Algorithm 1** Simultaneous reduction of a matrix pair $(\mathbf{H}^\star, \mathbf{S})$

---

**Require:** Ground-truth Hamiltonian matrix $\mathbf{H}^\star$ and overlap matrix $\mathbf{S}$
**Ensure:** Diagonal matrix $\boldsymbol{\epsilon}^\star$ and matrix $\mathbf{C}^\star$ such that $(\mathbf{C}^\star)^\top \mathbf{S}\mathbf{C}^\star = \mathbf{I}$ and $(\mathbf{C}^\star)^\top \mathbf{H}^\star\mathbf{C}^\star = \boldsymbol{\epsilon}^\star$
  1: Compute the Cholesky decomposition $\mathbf{S} = \mathbf{G}\mathbf{G}^\top$.
  2: Define $\mathbf{M}^\star = \mathbf{G}^{-1}\mathbf{H}^\star\mathbf{G}^{-\top}$.
  3: Apply the symmetric QR algorithm to find the Schur form $(\mathbf{Q}^\star)^\top \mathbf{M}^\star\mathbf{Q}^\star = \boldsymbol{\epsilon}^\star$.
  4: Compute $\mathbf{C}^\star = \mathbf{G}^{-\top}\mathbf{Q}^\star$.

---

When the overlap matrix $\mathbf{S}$ is ill-conditioned, the eigenvalues $\boldsymbol{\epsilon}^\star$ computed by Algorithm 1 can suffer from significant roundoff errors. To mitigate this, we modify the algorithm by replacing the Cholesky decomposition of $\mathbf{S}$ with its eigen (Schur) decomposition $\mathbf{V}^\top \mathbf{S}\mathbf{V} = \boldsymbol{\Sigma}$, where $\mathbf{V}$ is the matrix of eigenvectors and $\boldsymbol{\Sigma}$ is the diagonal matrix of eigenvalues. We then substitute $\mathbf{G}$ with $\mathbf{V}\boldsymbol{\Sigma}^{-1/2}$. This modification effectively reorders the entries of $\mathbf{M}^\star$, placing larger values towards the upper left-hand corner, thereby enhancing the precision in computing smaller eigenvalues (Golub & Van Loan, 2013; Wilkinson, 1988).

**Claim 1.** *Under optimal convergence condition, the ground truth eigenvector $\mathbf{C}^\star$ should diagonalize the predicted transformed matrix $\hat{\mathbf{H}}$, thereby satisfying the relation $(\mathbf{C}^\star)^\top \hat{\mathbf{H}} \mathbf{C}^\star = \boldsymbol{\epsilon}^\star$.*

Accordingly, we propose a refined loss function:

$$\mathcal{L}_{\text{WA}} = \frac{1}{n} \sum_{i=1}^{n} \|(\mathbf{C}_{(i)}^\star)^\top \hat{\mathbf{H}}_{(i)} \mathbf{C}_{(i)}^\star - \boldsymbol{\epsilon}_{(i)}^\star\|_{\text{F}}^2, \tag{2}$$

where the subscript $(\cdot)_{(i)}$ denotes the $i$-th sample. This modified loss function explicitly penalizes deviations from the expected eigenstructure. It is important to note that while $(\mathbf{C}^\star)^\top \hat{\mathbf{H}} \mathbf{C}^\star$ may not yield a diagonal matrix, the diagonal nature of $\boldsymbol{\epsilon}^\star$ implicitly enforces the predicted eigenstructure through the non-diagonal elements of the loss function. Here, we provide another derivation of the WALoss by drawing a parallel to first-order perturbation theory. In perturbation theory, for a Hamiltonian $\hat{\mathbf{H}} = \mathbf{H}^\star + \Delta\mathbf{H}$, where $\Delta\mathbf{H}$ is the perturbation. The first-order energy shift is $\delta\boldsymbol{\epsilon} = \int \psi^{\star\dagger} \Delta\mathbf{H} \psi^\star d\mathbf{r}$, using the unperturbed eigenvectors $\psi^\star$ of $\mathbf{H}^\star$, with the assumption that these eigenvectors remain fixed while adjusting the eigenvalues. Similarly, WALoss employs the ground-truth eigenvectors $\mathbf{C}^\star$ from $\mathbf{H}^\star \mathbf{C}^\star = \mathbf{S}\mathbf{C}^\star \boldsymbol{\epsilon}^\star$ to evaluate the predicted Hamiltonian $\mathbf{H}^\star + \Delta\mathbf{H}$. The loss could be rewritten as $\mathcal{L}_{\text{WA}} = \frac{1}{n} \sum_{i=1}^{n} \|(\mathbf{C}_{(i)}^\star)^\top (\mathbf{H}^\star + \Delta\mathbf{H})_{(i)} \mathbf{C}_{(i)}^\star - \boldsymbol{\epsilon}_{(i)}^\star\|_{\text{F}}^2$, assessing how well $\mathbf{H}^\star + \Delta\mathbf{H}$ reproduces $\boldsymbol{\epsilon}^\star$ in the $\mathbf{C}^\star$ basis. Here, $(\mathbf{C}^\star)^\top (\mathbf{H}^\star + \Delta\mathbf{H}) \mathbf{C}^\star = \boldsymbol{\epsilon}^\star + (\mathbf{C}^\star)^\top \Delta\mathbf{H} \mathbf{C}^\star$, where $(\mathbf{C}^\star)^\top \Delta\mathbf{H} \mathbf{C}^\star$ mirrors the perturbation term $\int \psi^{\star\dagger} \Delta\mathbf{H} \psi^\star d\mathbf{r}$.

It is widely known that energies associated with occupied molecular orbitals (and LUMO) make up the most energy of the molecule. Thus, to capture the most relevant part of the eigenspectrum, we calculate the loss function with increased weight on the $k + 1$ lowest eigenvalues. Here, $k$ corresponds to the number of occupied orbitals [2], and an additional eigenvalue accounts for the Lowest Unoccupied Molecular Orbital (LUMO). Let $\mathcal{I}$ represent the set of indices corresponding to the $k + 1$ lowest eigenvalues, and $((\cdot)_{(i)})_j$ indexes the $j$-th eigenvalue for the $i$-th sample. The loss function is defined as follows:

$$\mathcal{L}_{\text{WA}} = \frac{1}{n} \sum_{i=1}^{n} \left( \rho \sum_{j \in \mathcal{I}} \|(\mathbf{C}_{(i)}^\star)_j^\top \hat{\mathbf{H}}_{(i)} (\mathbf{C}_{(i)}^\star)_j - (\boldsymbol{\epsilon}_{(i)}^\star)_j\|_{\text{F}}^2 + \xi \sum_{j \notin \mathcal{I}} \|(\mathbf{C}_{(i)}^\star)_j^\top \hat{\mathbf{H}}_{(i)} (\mathbf{C}_{(i)}^\star)_j - (\boldsymbol{\epsilon}_{(i)}^\star)_j\|_{\text{F}}^2 \right), \tag{3}$$

where $n$ denotes the number of samples, and $\rho, \xi$ are hyperparameters where $\rho \gg \xi$. This adaptation ensures that the loss function places greater emphasis on the eigenvectors and eigenvalues corresponding to the occupied orbitals and LUMO. This improves the model's focus on the critical part of the eigenspace, which is crucial for practical applications.

## 4 WANET

Here, we present WANet, a modernized architecture for Hamiltonian prediction. First, unlike previous approaches, we propose a streamlined design for Hamiltonian prediction that consists of two essential components: the Node Convolution Layer and the Hamiltonian Head. The Node Convolution Layer operates on a localized radius graph, performing graph convolution to capture intricate atomic interactions. This block serves a dual purpose: first, it generates an irreducible node representation, providing a powerful input for the subsequent Hamiltonian Head. Second, it can be initialized with a pretrained EGNN or reprogrammed for other downstream tasks, constituting a unified framework for molecular modeling. The Hamiltonian Head constructs both pairwise and many-body irreducible representations using the Clebsch-Gordon tensor product. These representations are then utilized to assemble both the non-diagonal and diagonal components of the Hamiltonian matrix. The model architecture and ablation studies are detailed in Figure 10 and Table 9, respectively.

### 4.1 NODE CONVOLUTION LAYER

For the Node Convolution Layer, we replace the traditional $SO(3)$ convolutions with Equivariant Spherical Channel Network (eSCN) (Passaro & Zitnick, 2023; Liao et al., 2023). The eSCN

---

[2]$k = \lfloor \frac{N}{2} \rfloor$ for paired orbitals.

framework primarily utilizes $SO(2)$ linear operations, optimizing the computation of tensor products involved in the convolution process. Traditionally, $SO(3)$ convolutions operate on input irreducible representation (irrep) features $u_{m_i}^{\ell_i}$ and spherical harmonic projections $Y_{m_f}^{\ell_f}(\hat{\mathbf{r}}_{ts})$. By applying a rotation matrix $D_{ts}$ to $\hat{\mathbf{r}}_{ts}$, aligning it with the canonical axis where $\ell = 0$ and $m = 0$, the spherical harmonic projection $Y_{m_f}^{\ell_f}(D_{ts}\hat{\mathbf{r}}_{ts})$ becomes non-zero exclusively for $m_f = 0$. This condition simplifies the tensor product to $C_{(\ell_i,m_i),(\ell_f,0)}^{(\ell_o,m_o)}$, which remains non-zero only when $m_i = \pm m_o$. Subsequently, eSCN has demonstrated that this reformulation could be reparameterized using $SO(2)$ operations on the rotated tensors, and simplify the computation from $O(L^6)$ to $O(L^3)$, $L$ is the degree of the representaion. A detailed mathematical framework of this method is elaborated in the Appendix D.

## 4.2 Hamiltonian Head

**Sparse Mixture of Long-Short-Range Experts**  We introduce a variant of the Gated Mixture-of-Experts (GMoE) (Shazeer et al., 2017; Clark et al., 2022; Riquelme et al., 2021; Zoph et al., 2022; Jiang et al., 2024) model by incorporating a Mixture-of-Experts (MoE) layer tailored for pairwise molecular interactions. This enhancement draws inspiration from the Long-Short-Range Message Passing framework (Li et al., 2023), which differentiates between handling proximal and distal interactions through specialized layers. Our approach differentiates interaction dynamics based on distance, with closer pairs experiencing distinct interaction profiles compared to more distant pairs. This differentiation is achieved through a novel layer that substitutes the conventional pair interaction layer with a sparse assembly of expert modules, each functioning autonomously as a Pair Construction Layer. We define the Pair Construction layer with the function $F_{\text{pair}}^n$ for the $n$-th expert, and delineate the output of the MoE layer with $N$ experts as: $F_{\text{MoE}}(x_t, x_s) = \sum_{n=1}^{N} p_n(x_t, x_s) \cdot F_{\text{pair}}^n(x_t, x_s)$, where $p_n(x_t, x_s)$ are the gating probabilities computed by the gating network, and $\cdot$ denotes scalar multiplication. The gating probabilities are obtained by applying the *Softmax* function over the gating scores of all experts: $p_n(x_t, x_s) = \frac{\exp(G_n(z))}{\sum_{m=1}^{N} \exp(G_m(z))}$, where $z = \text{rbf}(\|\mathbf{r}_{ts}\|_2)$ applies a radial basis function to the Euclidean distance $\|\mathbf{r}_{ts}\|_2$ with a distance cutoff, and $G_n(z)$ represents the gating score for the $n$-th expert, computed as: $G_n(z) = z \cdot W_{g_n} + \epsilon_n$. Here, $W_{g_n}$ are learned gating weights for expert $n$, and $\epsilon_n$ is injected noise to encourage exploration and promote load balancing among experts. Specifically, we use *Gumbel* noise (Jang et al., 2016): $\epsilon_n = -\log\left(-\log\left(U_n\right)\right), \quad U_n \sim \text{Uniform}(0, 1)$. This noise enables a differentiable approximation of the top-$K$ selection, allowing for sparse expert utilization while maintaining gradient flow during training. To further promote load balancing among the experts, we introduce an auxiliary load balancing loss (Shazeer et al., 2017): $\mathcal{L}_{\text{load\_balancing}} = N \sum_{n=1}^{N} \left(\frac{\sum_{(t,s)} p_n(x_t, x_s)}{\sum_{(t,s)} 1}\right)^2$, which encourages the gating network to allocate routing probabilities evenly across experts, preventing underutilization of any single expert. The Pair Construction layer for each expert is defined as: $\left(F_{\text{pair}}^n(x_t, x_s)\right)^{l_o} = \sum_{l_i, l_j} W_{l_i, l_j, l_o}^n \left(x_s^{l_i} \otimes x_t^{l_j}\right)^{l_o}$, where $x_s^{l_i}$ and $x_t^{l_j}$ are the $l_i$-th and $l_j$-th irreducible representations of source node $s$ and target node $t$, respectively; $W_{l_i, l_j, l_o}^n$ are the learned weights that couple these representations into the output representation $l_o$; and $\otimes$ denotes the tensor product.

**Many-Body Interaction Layer**  Considering many-body interactions for the diagonal components of the Hamiltonian in molecular systems captures essential electron correlation effects (Szabo & Ostlund, 2012; Jensen, 2017). These interactions provide a more accurate representation of the collective behavior of atoms, beyond pairwise approximations. This leads to a precise description of key quantum phenomena like electron delocalization and exchange interactions (Szabo & Ostlund, 2012). For this purpose, we employ the methodologies of the MACE framework (Batatia et al., 2022; Kovács et al., 2023; Batatia et al., 2023). Central to MACE is the adept conversion of first-order features into higher-order features using the so-called *density trick* (Duval et al., 2023). This procedure initiates with the formation of generalized Clebsch-Gordon tensor products from the first-order features:

$$B_{M,\nu}^L = \sum_{lm} C_{lm,\nu}^{LM} \prod_{\xi=1}^{\nu} w_{l_\xi} f_m^l; \quad \text{where } lm = (l_1 m_1, \ldots, l_\nu m_\nu),$$

where $f_m^l$ represents the input tensor, and $B_{M,\nu}^L$ the resultant tensor for the $\nu$-body. The coefficients $C_{lm,\nu}^{LM}$ are the generalized Clebsch-Gordan coefficients, ensuring the $L$-equivariance of the output tensor $B_{M,\nu}^L$. Moreover, $C_{lm,\nu}^{LM}$ is notably sparse and can be pre-computed efficiently.

# 5 EXPERIMENTS

In this section, we evaluate WANet with WALoss on the QH9 and PubChemQH datasets. Our evaluation metrics include MAE for the Hamiltonian, $\epsilon_{\mathrm{HOMO}}$, $\epsilon_{\mathrm{LUMO}}$, $\epsilon_\Delta$, $\epsilon_{\mathrm{occ}}$, cosine similarity for the eigenvectors $\mathbf{C}$, and *relative* SCF iterations compared to the initial guess, which are commonly used in previous works (Unke et al., 2021b; Yu et al., 2023b;a). Additionally, we introduce two new physics-related metrics—MAE for $\epsilon_{\mathrm{orb}}$ and System Energy—to provide a more comprehensive evaluation. Detailed descriptions of these metrics are provided in Appendix C.1.

## 5.1 RESULTS ON THE PUBCHEMQH DATASET

**Dataset Generation Process**  In our study, we investigated the scalability of Hamiltonian learning by utilizing a *CUDA-accelerated SCF implementation* (Ju et al., 2024) to perform computational quantum chemistry calculations, thereby generating the PubChemQH dataset. We began with geometries from the PubChemQC dataset by (Nakata & Maeda, 2023), selecting only molecules with a molecular weight above 400. This filtration process resulted in a dataset comprising molecules with 40 to 100 atoms, totaling over 50,000 samples. We chose the B3LYP exchange-correlation functional (Lee et al., 1988; Beeke, 1993; Vosko et al., 1980; Stephens et al., 1994) and the Def2TZV basis set (Weigend & Ahlrichs, 2005; Weigend, 2006) to approximate electronic wavefunctions. Generating this comprehensive dataset represents a substantial computational effort, requiring approximately *one month of continuous processing using 128 NVIDIA-V100 GPUs*. We provide a comparison between the PubChemQH dataset and the QH9 dataset in Appendix H.

Table 1: PubChemQH experimental results. The energy units are presented in kcal/mol. N/A indicates that the metric is not applicable to a specific model. The best-performing models are highlighted in bold. Detailed training setups are provided in Table 12 and Table 13.

| Model | Hamiltonian MAE ↓ | $\epsilon_{\mathrm{HOMO}}$ MAE ↓ | $\epsilon_{\mathrm{LUMO}}$ MAE ↓ | $\epsilon_\Delta$ MAE ↓ | $\epsilon_{\mathrm{occ}}$ MAE ↓ | $\epsilon_{\mathrm{orb}}$ MAE ↓ | C Similarity ↑ | System Energy MAE ↓ | relative SCF Iterations ↓ |
|---|---|---|---|---|---|---|---|---|---|
| QHNet | 0.7765 | 71.250 | 83.890 | 5.790 | 2087.45 | 1532.672 | 2.32% | 65721.028 | 371% |
| WANet | 0.6274 | 60.140 | 62.35 | 4.723 | 734.258 | 502.43 | 3.13% | 63579.233 | 334% |
| QHNet w/ WALoss | 0.5207 | 13.945 | 14.087 | 4.3982 | 21.805 | 10.930 | 46.66% | 75.625 | 90% |
| PhisNet w/ WALoss | 0.5166 | 11.872 | 12.075 | 4.054 | 20.045 | 8.917 | 46.78% | 60.325 | 90% |
| WANet w/ WALoss | **0.4744** | **0.7122** | **0.730** | **1.327** | **18.835** | **7.330** | **48.03%** | **47.193** | **82%** |
| init guess (minao) | 0.5512 | 29.430 | 28.521 | 4.955 | 42.740 | 35.183 | 0.3293 | 374.313 | 100% |
| Equiformer V2 Regression | N/A | 6.955 | 6.562 | 3.222 | N/A | N/A | N/A | N/A | N/A |
| UniMol+ Regression | N/A | 12.250 | 9.472 | 13.001 | N/A | N/A | N/A | N/A | N/A |
| UniMol 2 Regression | N/A | 9.573 | 7.414 | 10.638 | N/A | N/A | N/A | N/A | N/A |

Table 1 presents a comparative analysis of various models' performance on the PubChemQH dataset. Despite a higher Hamiltonian MAE, WANet with WALoss significantly outperforms the other models in practical utility, as evidenced by the System Energy MAE and the required SCF iterations. Specifically, the System Energy MAE for WANet with WALoss is dramatically reduced from 63579.233 kcal/mol to 47.193 kcal/mol. Additionally, the relative SCF iterations required for WANet with WALoss is only 82%, compared to 371% for QHNet and 334% for WANet without WALoss. This substantial reduction in SCF iterations demonstrates the effectiveness of WALoss in accelerating the convergence process. Furthermore, it is worth noting that the initial guess matrix, although not achieving as low an MAE as QHNet or WANet without WALoss, shows improved utility with a System Energy MAE of 374.313 kcal/mol. This finding reinforces the idea that *elementwise losses are insufficient*. Additional evidence supporting this idea is provided in Appendix C.2.

## 5.2 RESULTS ON THE QH9 DATASET

The QH9 dataset is a comprehensive quantum chemistry resource designed to support the development and evaluation of machine learning models for predicting quantum Hamiltonian matrices. Built upon the QM9 dataset, QH9 contains Hamiltonian matrices for 130,831 stable molecular geometries, encompassing molecules with up to nine heavy atoms of elements C, N, O, and F. These Hamiltonian matrices were generated using `pyscf` with the B3LYP functional (Lee et al., 1988; Beeke, 1993; Vosko et al., 1980; Stephens et al., 1994) and the def2SVP basis set. Table 2 presents a comparative analysis of the performance of our model, WANet, against the baseline model, QHNet, on the QH9 dataset in both stable and dynamic settings. In the QH9-stable experiments, WANet

Table 2: Experimental results on the QH9 dataset. The energy units are presented in kcal/mol.

| | Model | Hamiltonian MAE ↓ | $\epsilon_{occ}$ MAE ↓ | C similarity ↑ |
|---|---|---|---|---|
| QH9 stable | QHNet | 0.0513 | 0.5366 | 95.85% |
| | QHNet w/WAloss | 0.0780 | 0.4901 | 96.35% |
| | WANet | **0.0502** | 0.5231 | 96.86% |
| | WANet w/WAloss | 0.0914 | **0.4587** | **96.95%** |
| QH9 dynamic | QHNet | 0.0471 | 0.2744 | 97.13% |
| | QHNet w/WAloss | 0.0495 | 0.2658 | 98.54% |
| | WAnet | **0.0469** | 0.2614 | 99.68% |
| | WAnet w/WAloss | 0.0512 | **0.2500** | **99.81%** |

demonstrates superior performance, achieving higher accuracy compared to QHNet. Specifically, WANet significantly reduces both the Hamiltonian and occupied energy MAE while improving the cosine similarity of C. For the QH9-dynamic dataset, WANet consistently outperforms QHNet, further enhancing prediction accuracy. These results underscore the robustness and effectiveness of WANet in both stable and dynamic scenarios.

### 5.3 COMPARISON WITH A PROPERTY REGRESSION MODEL

Conventional machine learning approaches typically employ property regression, mapping molecular features directly to the desired property value (Blum & Reymond, 2009; Montavon et al., 2013). A common question arises: why use Hamiltonians instead of a property regression model? We argue that property regression methods often fail to incorporate underlying quantum mechanical principles, limiting their generalization capability. To illustrate this, we compared the performance of WANet with WALoss to a model utilizing Equiformer V2 (Liao & Smidt, 2023), UniMol+ (Lu et al., 2023), and UniMol2 (Xiaohong et al., 2024) with invariant regression heads, using identical training and test sets. As shown in Table 1, WANet with WALoss demonstrates significantly lower MAE values in predicting key quantum chemical properties. Specifically, WANet with WALoss achieves an 88.88% improvement in $\epsilon_{LUMO}$ MAE and a 58.81% improvement in $\epsilon_\Delta$ MAE. Moreover, the Hamiltonian predicted by WANet with WALoss is not limited to specific properties. It enables the accurate calculation of various critical properties, such as electronic densities, dipole moments, and excited-state energies, all from a single model. Additionally, it can be applied to SCF acceleration. In contrast, the Equiformer V2 regression model is constrained to predicting a narrow set of specific properties, necessitating the training of a new model for each new property.

### 5.4 EFFICIENCY EVALUATION OF WANET

WANet exhibits efficiency advantages in two aspects: (1) its application to SCF relative to traditional DFT calculations, and (2) its efficiency and resource usage compared to existing state-of-the-art neural networks. Specifically, WANet can predict Hamiltonians for large molecular systems significantly faster than traditional DFT methods. Figure 3a presents a wall-clock time comparison between WANet-augmented SCF and traditional DFT calculations. Although the neural network evaluation introduces a small overhead, WANet substantially reduces the number of required SCF iterations, resulting in a faster overall computation time. This notable speed-up makes WANet particularly advantageous for applications requiring rapid predictions for large molecular systems, such as high-throughput virtual screening. Furthermore, WANet outperforms QHNet in training and inference efficiency on the PubChemQH dataset, offering faster training times, improved inference speeds, and lower peak GPU memory usage (Figure 3b).

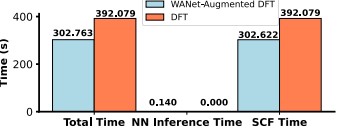

Figure 3(a): Wall-clock comparison of WANet-Augmented DFT with traditional SCF iterations.

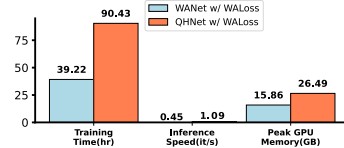

(b): Comparison of training and inference efficiency and resource usage between QHNet and WANet on the PubChemQH dataset.

### 5.5 MOLECULAR PROPERTIES BEYOND ENERGY PREDICTIONS

To validate the versatility of the Hamiltonian predicted by WANet with WALoss, we extended our experiments to include predictions beyond energy properties, specifically dipole moment and electronic spatial extent.

Table 3: Additional property predictions. We evaluated WANet in deriving dipole moment and electronic spatial extent.

| Model | Dipole Moment MAE (D) | Electronic Spatial Extent MAE (a.u.) |
|---|---|---|
| Equiformer V2 Regression | **3.3221** | 0.1098 |
| EGNN Regression | 4.3412 | 0.0789 |
| ViSNet Regression | 4.5211 | 0.0982 |
| Initial Guess | 4.0170 | 0.0161 |
| WANet w/ WALoss | 3.3928 | **0.0076** |

As shown in Table 3, WANet with WALoss performs competitively in dipole moment prediction and excels in predicting electronic spatial extent. This demonstrates the model's ability to generalize across multiple molecular properties, highlighting its potential for broader quantum mechanical

calculations beyond energy-based properties. Further details on the derivation of these properties from the Hamiltonian are provided in Appendix F.

## 5.6 SCALABILITY IN ELONGATED CARBON CHAIN

To evaluate the scalability of our model trained on PubChemQH, we conducted inference using elongated alkanes $(C_xH_{2x+2})^3$, a series of saturated hydrocarbons. We compared three models: our model with WALoss, our model without WALoss, and an initial guess algorithm.

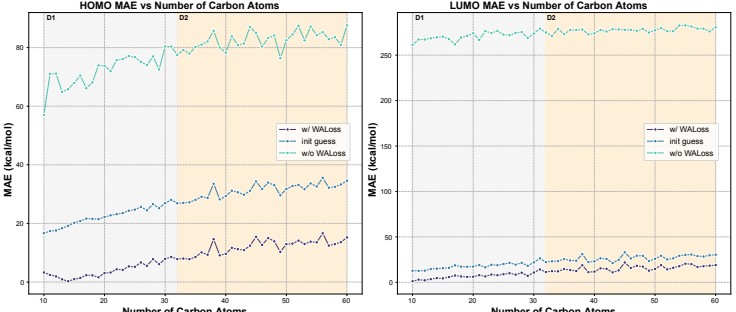

The results, shown in Figure 4, demonstrate that our model with WALoss achieves enhanced performance in predicting LUMO and HOMO energies, particularly in the "D2" region, where the atom count exceeds the range of the PubChemQH dataset. Notably, our model with WALoss also performs well on elongated alkanes with up to 182 atoms—three times the average atom count of the PubChemQH training set (60). These findings highlight the effectiveness of WALoss in enhancing the scalability and applicability

Figure 4: Model performance in predicting HOMO and LUMO energies for elongated alkanes. The left panel shows the MAE for HOMO predictions, while the right panel shows the MAE for LUMO predictions. "D1" indicates that the atom count is within the range of the PubChemQH dataset, whereas "D2" indicates that the atom count exceeds this range. The models compared include WANet with WALoss (w/ WALoss), our model without WALoss (w/o WALoss), and the initial guess (init guess). Notably, our model with WALoss demonstrates superior performance in LUMO predictions and matches the best HOMO performance, particularly in the "D2" region.

of our model for predicting electronic properties in scalable homogeneous series, demonstrating its potential for application to larger and more complex molecular systems. Additional analysis of the scaling performance is provided in Appendix C.3.

## 5.7 ABLATION STUDY ON WALOSS

To evaluate the effectiveness of our proposed WALoss, we conducted an ablation study with three variations: full WALoss, naive WALoss, and WALoss without reweighting. The naive WALoss applies the Frobenius norm to the eigenvalues leveraging backpropagation through eigensolvers, defined as $\mathcal{L}_{\text{naive}} = \frac{1}{n} \sum_{i=1}^{n} \|\hat{\boldsymbol{\epsilon}}_{(i)} - \boldsymbol{\epsilon}^{\star}_{(i)}\|_{\text{F}}^2$, where $\hat{\boldsymbol{\epsilon}}_{(i)}$ and $\boldsymbol{\epsilon}^{\star}_{(i)}$ are derived from solving generalized eigenvalue problems. The WALoss without reweighting calculates the loss uniformly across all eigenvalues. As shown in Table 4, the full WALoss achieves the lowest MAEs across all metrics. The naive WALoss performs poorly, highlighting several challenges associated with optimizing the naive loss function. Removing reweighting also degrades performance, though not as drastically. Overall, these results validate the design choices in formulating WALoss to improve Hamiltonian prediction.

Table 4: Ablation study of WALoss on the PubChemQH dataset. The best-performing models are highlighted in bold.

| Model | Hamiltonian MAE ↓ | $\epsilon_{\text{HOMO}}$ MAE ↓ | $\epsilon_{\text{LUMO}}$ MAE ↓ | $\epsilon_\Delta$ | $\epsilon_{\text{occ}}$ MAE ↓ | $\epsilon_{\text{orb}}$ MAE ↓ | C ↑ | System Energy MAE ↓ | SCF Iteration ↓ |
|---|---|---|---|---|---|---|---|---|---|
| Naive Loss | 0.4912 | 50.174 | 55.630 | 3.634 | 632.220 | 486.322 | 5.36% | 13562.7 | 306% |
| WALoss without Reweighting | 0.4973 | 8.241 | 7.993 | 3.988 | 41.230 | 21.614 | 28.28% | 55.492 | 88% |
| WALoss Complete | **0.4744** | **0.7122** | **0.730** | **1.327** | **18.835** | **7.330** | **48.03%** | **47.193** | **82%** |

## 6 CONCLUSION AND LIMITATIONS

In this work, we introduced WALoss, a loss function designed to improve the accuracy of predicted Hamiltonians. Our experiments demonstrate that incorporating WALoss achieves state-of-the-art performance by reducing prediction errors and accelerating SCF convergence. Additionally, we introduced a new dataset, *PubChemQH*, and an efficient model, *WANet*. However, limitations remain, such as the high computational cost of generating large training sets. Despite these challenges, deep learning approaches incorporating WALoss show great promise in advancing computational chemistry and materials science.

---

[3]Elongated alkanes $(C_xH_{2x+2})$ are only present in the test set.

ACKNOWLEDGMENT

Tha authors would like to thank Gaoyuan Wang from Yale University for the discussion on the scaling of the eigenvalues. This work is supported by ALW professorship funds to Mark Gerstein.

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

**Appendix Table of Contents**

# A  Notation Summary

Table 5: Summary of main notation used in the paper.

| Notation | Description |
|---|---|
| $\mathcal{M} := \{\mathbf{Z}, \mathbf{R}\}$ | Molecular system defined by nuclear charges $\mathbf{Z}$ and positions $\mathbf{R}$ |
| $\mathbf{r} \in \mathbb{R}^3$ | Spatial coordinate of an electron |
| $\psi_i(\mathbf{r})$ | Single-electron orbital/wavefunction $i$ as a function of electron spatial coordinate $\mathbf{r}$ |
| $\rho(\mathbf{r})$ | Electron density as a function of spatial coordinate $\mathbf{r}$ |
| $B$ | Number of basis functions in the basis set used to represent orbitals |
| $\phi_\alpha(\mathbf{r})_{\alpha=1}^B$ | Basis set consisting of $B$ basis functions $\phi_\alpha(\mathbf{r})$ |
| $\mathbf{C} \in \mathbb{R}^{B \times N}$ | Coefficient matrix where each column contains the basis function coefficients for an orbital |
| $\mathbf{H} \in \mathbb{R}^{B \times B}$ | Hamiltonian matrix in the chosen basis representation |
| $\mathbf{S} \in \mathbb{R}^{B \times B}$ | Overlap matrix with elements $S_{\alpha\beta} := \int \phi_\alpha^\dagger(\mathbf{r})\phi_\beta(\mathbf{r})\mathrm{d}\mathbf{r}$ |
| $\boldsymbol{\epsilon} \in \mathbb{R}^{N \times N}$ | Diagonal matrix containing orbital energies (eigenvalues of the Hamiltonian) |
| $\hat{\mathbf{H}}_\theta(\mathcal{M})$ | Machine learning model parameterized by $\theta$ for predicting the Hamiltonian $\mathbf{H}$ from a molecular structure $\mathcal{M}$ |
| $\mathcal{L}_{\mathrm{WA}}$ | Wavefunction Alignment Loss function |
| $\hat{\boldsymbol{\epsilon}}, \boldsymbol{\epsilon}$ | Predicted and ground truth eigenvalues (orbital energies) of $\hat{\mathbf{H}}$ and $\mathbf{H}$ |
| $\hat{\mathbf{C}}, \mathbf{C}$ | Predicted and ground truth eigenvectors (basis coefficients) of $\hat{\mathbf{H}}$ and $\mathbf{H}$ |
| $\mathbf{H}^\star$ | Ground truth Hamiltonian matrix |
| $\hat{\mathbf{M}}$ | Predicted transformed Hamiltonian matrix in an orthogonal basis |
| $\mathbf{G}$ | Matrix obtained from the eigen decomposition of the overlap matrix $\mathbf{S}$ |
| $k$ | Number of occupied orbitals |
| $\mathcal{I}$ | Set of indices corresponding to the $k+1$ lowest eigenvalues |
| $\rho$ | Hyperparameter controlling the weight of occupied and unoccupied orbitals in the loss function |
| $n$ | Number of samples in the dataset / the $n$-th experts in MoE nodel |
| $\mathcal{D}$ | Dataset used for training the machine learning model |
| $N$ | Number of electrons in the system / total number of experts in MoE model |
| $\mathbf{H}^{(0)}$ | Initial guess for the Hamiltonian matrix |
| $\mathbf{H}^{(k)}$ | Hamiltonian matrix at the $k$-th SCF iteration |
| $\mathbf{C}^{(k)}$ | Coefficient matrix at the $k$-th SCF iteration |
| $\boldsymbol{\epsilon}^{(k)}$ | Diagonal matrix of orbital energies at the $k$-th SCF iteration |
| $\delta$ | Convergence threshold for the SCF procedure |
| $\ell$ | Degree of the irreducible representation (irrep) in SO(3) equivariant networks |
| $D^{(\ell)}(g)$ | Wigner D-matrix representation of group element $g \in \mathrm{SO}(3)$ of degree $\ell$ |
| $Y^{(\ell)}(\hat{\mathbf{r}})$ | Real spherical harmonics of degree $\ell$ evaluated at unit vector $\hat{\mathbf{r}}$ |
| $C^{(\ell_3, m_3)}_{(\ell_1, m_1),(\ell_2, m_2)}$ | Clebsch-Gordan coefficients coupling irreps of degree $\ell_1$ and $\ell_2$ into irrep of degree $\ell_3$ |
| $\mathbf{r}_{ts}$ | Vector pointing from node $s$ to node $t$ |
| $\hat{\mathbf{r}}_{ts}$ | Unit vector pointing from node $s$ to node $t$ |
| $\mathcal{L}_{\mathrm{total}}$ | Total loss function combining $\mathcal{L}_{\mathrm{align}}$ and mean squared error (MSE) loss |
| $\lambda_1, \lambda_2, \lambda_3$ | Hyperparameters controlling the weights of different loss terms in $\mathcal{L}_{\mathrm{total}}$ |

## B  ADDITIONAL RELATED WORK

**Predicting Kohn-Sham Hamiltonians**    Early work on predicting Kohn-Sham Hamiltonians used kernel ridge regression (Hegde & Bowen, 2017), while newer approaches use neural networks (NNs). Some NNs predict the wavefunction itself (Schütt et al., 2019; Gastegger et al., 2020; Hermann et al., 2020), while others use equivariant (Unke et al., 2021a; Kochkov et al., 2021; Yu et al., 2023b; Li et al., 2022; Zhong et al., 2023) or hybrid architectures (Yin et al., 2024) to predict the molecular Hamiltonian. A novel training method has been proposed to address the scarcity of labeled data (Zhang et al., 2024), and two benchmark datasets aim to standardize evaluation of molecular Hamiltonian prediction (Khrabrov et al., 2022; Yu et al., 2023a; Khrabrov et al., 2024).

**Equivariant Graph Neural Networks (EGNNs)**    It is often desired that machine learning (ML) models exhibit equivariance to rotations, translations, or reflections, which guarantee that they respect certain physical symmetries. Foundational work introduced group equivariant convolutional neural networks (ECNNs) (Cohen & Welling, 2016), whose importance was underscored by a proof that equivariance and convolutional structure are equivalent given certain ordinary constraints (Kondor & Trivedi, 2018). One appealing way to implement convolution is with geometric graph neural networks (geometric GNNs), which apply naturally to atomic systems by encoding them as graphs embedded in $\mathbb{R}^3$. Two important families of geometric GNNs are invariant GNNs and Cartesian equivariant GNNs. Over the past several years, invariant GNNs have achieved state-of-the-art results in predicting properties of molecules, crystals, and other materials  (Schütt et al., 2018) (Sanyal et al., 2018) (Chen et al., 2019) (Gasteiger et al., 2020) (Liu et al., 2022) (Gasteiger et al., 2021) (Wang et al., 2022), as well as in predicting the folding structure of proteins (Jumper et al., 2021). Cartesian equivariant GNNs have seen success in similar areas, benefiting from the greater flexibility of their representations (Jing et al., 2020) (Satorras et al., 2021) (Du et al., 2022) (Simeon & De Fabritiis, 2024). Cartesian equivariant GNNs have also seen recent innovation in Cartesian equivariant transformer layers (Frank et al., 2022). A third significant family of geometric GNNs is spherical equivariant GNNs, which use spherical tensors rather than Cartesian tensors. As as a result, spherical equivariant GNNs behave more naturally under rotations and avail themselves of many results of the representation theory of $SO(3)$. They have shown dexterity in tasks in geometry, physics, and chemistry (Thomas et al., 2018); modeled dynamic molecular systems (Anderson et al., 2019); enabled $SO(3)$- and $SE(3)$-equivariant transformer layers (Fuchs et al., 2020) (Liao & Smidt, 2022); accuarately and efficiently calculated interatomic potentials (Batzner et al., 2022) (Batatia et al., 2022) (Musaelian et al., 2023); enabled $E(3)$-equivariant fluid mechanical modeling  (Toshev et al., 2023); and improved efficiency by reducing certain convolution computations in $SO(3)$ to equivalent ones in $SO(2)$ (Passaro & Zitnick, 2023).  Li et al. (2025) propose to use Wiger-$6j$ Convolution to shift computation from edges to nodes.

## C  ADDITIONAL EXPERIMENTS

### C.1  EVALUATION METRICS

In this section, we provide a detailed description of the metrics in the main-text:

**MAE for Hamiltonian**    The MAE for Hamiltonian assesses the accuracy of the predicted Hamiltonian matrices. This metric is crucial for evaluating the quality of the predicted electronic structure and its components, which are foundational in quantum chemistry calculations.

**MAE for $\epsilon_{\text{HOMO}}$**    The MAE for the Highest Occupied Molecular Orbital (HOMO) energy ($\epsilon_{\text{HOMO}}$) evaluates the precision of the predicted HOMO levels. Accurate HOMO predictions are essential as they influence a molecule's chemical reactivity and stability.

**MAE for $\epsilon_{\text{LUMO}}$**    The MAE for the Lowest Unoccupied Molecular Orbital (LUMO) energy ($\epsilon_{\text{LUMO}}$) measures the accuracy of LUMO level predictions. Accurate LUMO predictions are critical for understanding a molecule's electron affinity and chemical behavior.

**MAE for $\epsilon_{\Delta}$**    The MAE for the energy gap between HOMO and LUMO ($\epsilon_{\Delta}$) assesses the precision of this important property, which determines the electronic properties and conductivity of materials.

**MAE for $\epsilon_{\text{occ}}$**    The MAE for occupied orbital energies ($\epsilon_{\text{occ}}$) assesses the accuracy of predicted energies for all occupied molecular orbitals, providing a comprehensive measure of how well the model captures the electronic structure.

**MAE for $\epsilon_{\text{orb}}$**    The MAE for all orbital energies ($\epsilon_{\text{orb}}$) measures the discrepancies in predicted energies for both occupied and unoccupied orbitals. This metric evaluates the overall accuracy of the model in predicting the entire spectrum of orbital energies.

**MAE on Total Energy**    The MAE on total energy assesses the accuracy of the predicted total energies derived from Hamiltonian matrices using `pyscf`. This metric is crucial for validating the model's accuracy in predicting the overall energy of the system, which is fundamental for understanding molecular stability and reactions.

**Cosine Similarity for Wavefunction/Eigenvectors (C)**    The cosine similarity for wavefunction/eigenvectors (C) measures the similarity between the predicted and actual wavefunctions or eigenvectors of the system. High cosine similarity indicates that the predicted wavefunction distribution closely matches the actual distribution, which is important for accurately modeling electronic properties.

**SCF Iteration**    The SCF (Self-Consistent Field) iteration count evaluates the number of iterations required to achieve convergence in DFT (Density Functional Theory) calculations using the predicted Hamiltonian matrices when comparing with the inital guess. Mathmatically, it is defined as:

$$\sigma = \frac{\text{Predicted Hamiltonian Iteractions}}{\text{Inital Hamiltonian Iteractions}}.$$

This metric assesses the efficiency of the predicted matrices in expediting the DFT calculations.

### C.2  SAD PHENOMENON

We present an additional graphical illustration of the SAD phenomenon discussed in the main text, depicting the learning curve of the QHNet model using only elementwise loss on a *subset* of the *PubChemQH* dataset [4]. Figure 5 shows that as the MAE decreases, the system energy exhibits significant fluctuations. Notably, when the Hamiltonian's MAE is around 0.3, the system energy MAE reaches 30,000 kcal/mol. This highlights the non-monotonic relationship between the Hamiltonian MAE and the resulting system energy. This extreme instability in derived properties, despite a seemingly small Hamiltonian MAE, is a key characteristic of the SAD phenomenon, hindering the applicability of the predicted Hamiltonian.

---

[4]This differs from the dataset used in the maintexts.

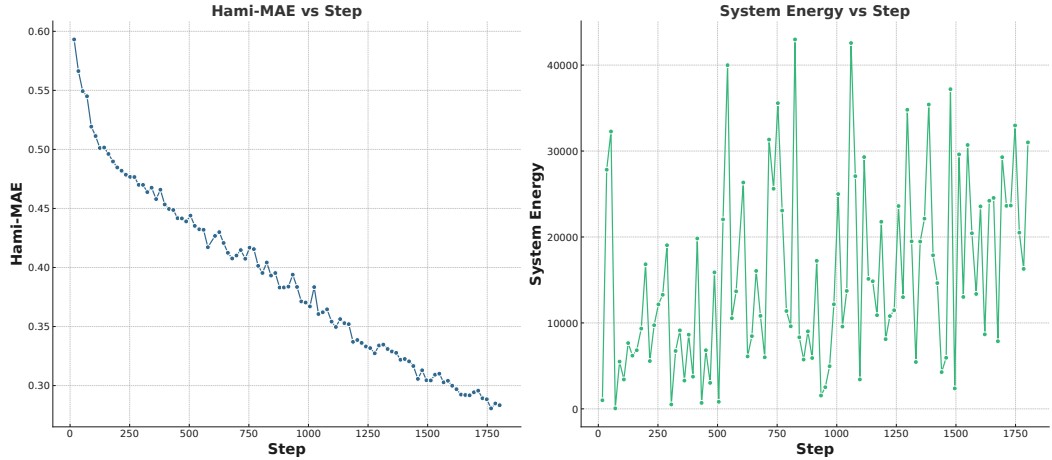

Figure 5: An additional graphical illustration of the SAD phenomenon discussed in the main text, depicting the learning curve of the QHNet model using only elementwise loss. The y-axis represents the metric, and the x-axis represents the steps. The figure shows that as the MAE decreases, the system energy exhibits significant fluctuations.

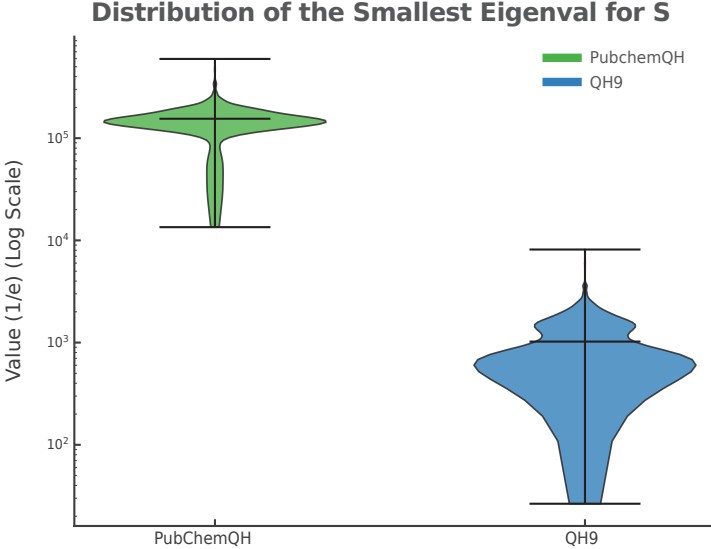

Figure 6: the distribution of $\frac{\kappa(\mathbf{S})}{\|\mathbf{S}\|}$ on QH9 and PubChemQH dataset

### C.3 ERROR SCALING ANALYSIS

In Figure 4 and Table 6, we analyzed the error trend scaling for "intensive" properties such as $\epsilon_{\text{HOMO}}$ and $\epsilon_{\text{LUMO}}$ with an increasing number of electrons. This challenge, widely recognized in the field (Yu et al., 2023a; Zhang et al., 2024), highlights a difficult out-of-distribution (OOD) generalization scenario: models trained on smaller molecules often struggle to generalize to larger ones, limiting their applicability. Previous state-of-the-art models have exhibited significant scaling errors when extrapolating to larger systems (Yu et al., 2023a; Zhang et al., 2024).

We conducted an error scaling analysis for saturated carbon chains to assess the scalability of our model. The error scaling coefficients for the HOMO and LUMO energies, as well as the energy gap, are presented in Table 6. Our results show significantly lower error scaling compared to baseline models for saturated carbon chains. Our model substantially outperforms the one without WALoss, as reflected by the scaling coefficients in Table R2, marking a considerable improvement in extrapolation capabilities within the field.

We further analyze the smallest eigenvalue distribution which is now included in Figure 6.

Table 6: Scaling Coefficients of HOMO, LUMO, and Energy Gap with Carbon Atom Count

| Model | HOMO Scaling Coefficients | LUMO Scaling Coefficients | Gap Scaling Coefficients |
|---|---|---|---|
| WANet (with WALoss) | **0.4211** | **0.4003** | **0.0206** |
| Initial Guess | 0.4221 | 0.4006 | 0.0214 |
| Without WALoss | 1.3639 | 2.7646 | 1.4007 |

## C.4 OVERFITTING RISK EVALUATION

We evaluated the risk of overfitting to the PubChemQH dataset. As shown in Table 7, the evaluation metrics on unseen data are comparable to those on the training set, indicating that our model generalizes well and does not overfit.

Table 7: Overfitting Risk Evaluation

| PubChem | Hamiltonian MAE $\downarrow$ | $\epsilon_{HOMO}$ MAE $\downarrow$ | $\epsilon_{LUMO}$ MAE $\downarrow$ | $\epsilon_\Delta$ | $\epsilon_{occ}$ MAE $\downarrow$ | $\epsilon_{orb}$MAE $\downarrow$ | C$\uparrow$ | System Energy MAE $\downarrow$ |
|---|---|---|---|---|---|---|---|---|
| Training MAE | 0.4736 | 6.073 | 4.414 | 3.753 | 17.850 | 6.159 | 48.01% | 46.036 |
| Test MAE | 0.4744 | 0.7122 | 0.730 | 1.327 | 18.835 | 7.330 | 48.03% | 47.193 |

## C.5 ENERGY PREDICTIONS FOR ELONGATED ALKANES

We have incorporated additional plots (Figure 7) that illustrate the predicted and actual values for both LUMO and HOMO energies, as well as the HOMO-LUMO gap. As the number of carbon atoms increases, both HOMO and LUMO energies exhibit an initial rise followed by a plateau with slight increases. Our model, utilizing the WALoss method, effectively captures this trend, outperforming baseline methods. Notably, Equiformer V2's HOMO predictions fail to generalize to saturated carbon systems, as these systems were not included in the training set. This observation underscores the advantage of Hamiltonian-based models in capturing physical principles and achieving better generalization compared to property regression models.

For isolated molecular systems, such as those studied here, computational chemists are primarily concerned with per-electron energy levels. To demonstrate this, we plot the energy levels within a reasonable window centered around the HOMO level and compare these with ground truth values obtained from DFT calculations. As shown in Figure 7 **A**, our model closely replicates the ground truth energy levels, outperforming baseline models. These results highlight the effectiveness of our approach in accurately predicting the electronic structure of molecules.

## C.6 ADDITIONAL ABLATION STUDY

In contrast to prior research, our study addresses the substantial challenges presented by the scale of the PubChemQC t-zvp dataset, particularly in providing the model with a strong numeric starting point. To overcome this difficulty, we shifted our focus to a more tractable objective: making predictions based on an easily obtainable initial guess. Our ablation study shows that while predictions based on the initial guess significantly improve performance in Hamiltonian prediction, they struggle when applied to the prediction of physical properties.

Table 8: Performance Comparison of Baseline and Initial Guess Models on PubChemQH Dataset. The best models are bolded.

| Model | Hamiltonian MAE $\downarrow$ | $\epsilon_{HOMO}$ MAE $\downarrow$ | $\epsilon_{LUMO}$ MAE $\downarrow$ | $\epsilon_\Delta$ MAE $\downarrow$ | $\epsilon_{occ}$ MAE $\downarrow$ | $\epsilon_{orb}$MAE $\downarrow$ | C Similarity $\uparrow$ | System Energy MAE $\downarrow$ | relative SCF Iterations $\downarrow$ |
|---|---|---|---|---|---|---|---|---|---|
| WANet | 0.6274 | 60.14 | 62.35 | 4.723 | 734.258 | 502.43 | 3.13% | 63579.233 | 334% |
| WANet w/ Initial Guess | **0.0379** | 54.107 | 56.628 | 3.618 | 695.418 | 481.513 | 4.42% | 60078.633 | 306% |
| WANet w/ Initial Guess & WALoss | 0.4744 | **0.7122** | **0.730** | **1.327** | **18.835** | **7.330** | **48.03%** | **47.193** | **82%** |

To evaluate the contributions of each component in the WANet model, we conducted ablation studies. The results are summarized in the Table 9. The ablation studies confirm the importance of each architectural component in the WANet model.

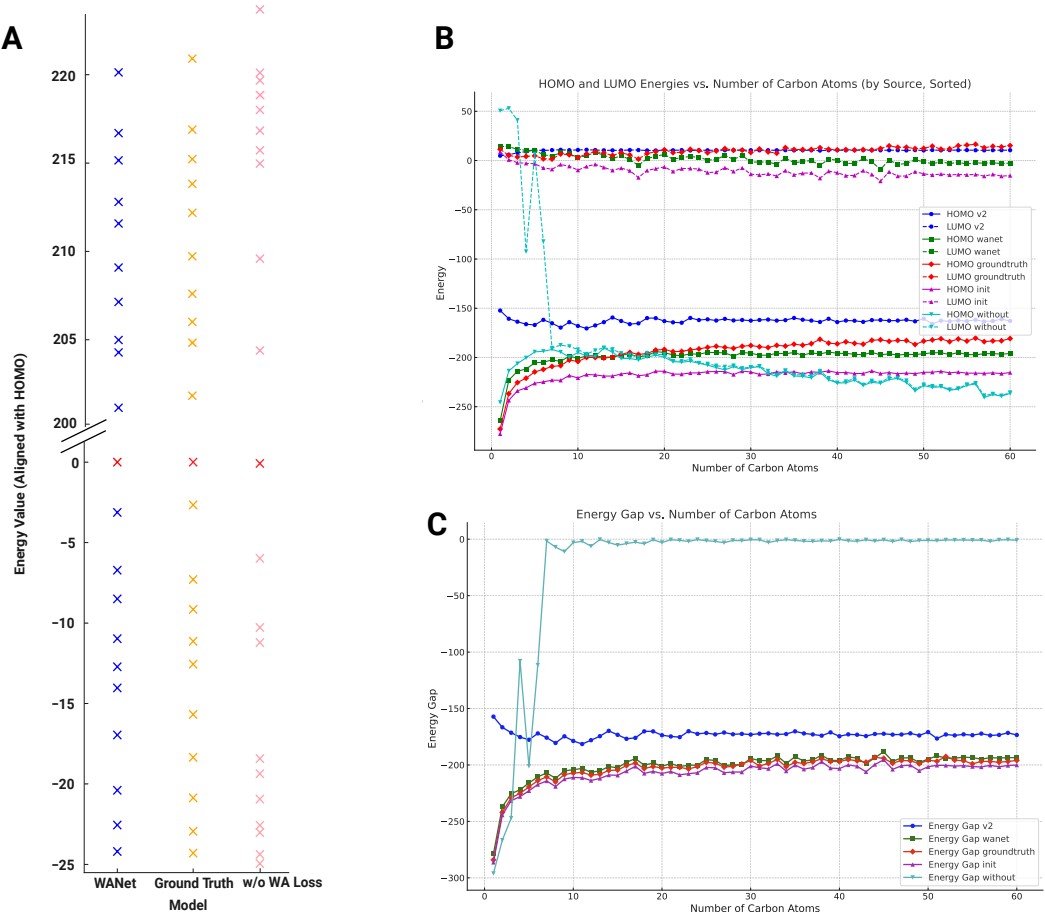

Figure 7: (**A**) The energy level for a saturated hydrocarbon system ($C_{20}H_{42}$). The energy is centered at HOMO for comparison. (**B**) The predicted energy for differnet models on the satured hydrocarbon system. The y-axis indicates the energy in kcal/mol. The x-axis denotes the number of the carbon atoms in the elongated carbon-chains. v2 indicates Equiformer V2. 'wanet' indicates WANet with WALoss. 'groundtruth' indicates DFT calculations. 'init' indicates a Fock matrix initialization algorithm using minao. 'without' indicates WANet without WALoss. (**C**) The HOMO-LUMO gap prediction for different models.

Table 9: Ablation study results for the WANet model. The table shows the impact of different architectural components.

| w/ SO(2) | w/ LSR-MoE | w/ Many Body | Inference Speed (it/s) | GPU Memory | Hamiltonian MAE | $\epsilon_{\text{HOMO}}$ MAE $\downarrow$ | $\epsilon_{\text{LUMO}}$ MAE $\downarrow$ | $\epsilon_{\Delta}$ MAE $\downarrow$ |
|---|---|---|---|---|---|---|---|---|
| ✓ | ✓ | | 1.34 | 13.87 | 0.5895 | 15.39 | 27.50 | 4.651 |
| ✓ | | ✓ | 1.13 | 12.80 | 0.4883 | 2.27 | 4.01 | 2.906 |
| | ✓ | ✓ | 0.51 | 24.47 | 0.4792 | 0.75 | 0.73 | 1.594 |
| ✓ | ✓ | ✓ | 1.09 | 15.86 | 0.4744 | 0.71 | 0.73 | 1.327 |

## D  ESCN CONVOLUTION

The equivariant Spherical Channel Network (eSCN) is a graph neural network whose approach to SO(3)-equivariant convolutions significantly reduces their computational burden. To see this, let's brush up on the notation used in the formula for the $\ell_o$-th degree of the message $m_{ts}$ from source node $s$ to target node $t$ in an SO(3) convolution: $W_{\ell_i,\ell_f,\ell_o}$ is a learnable weight; $x_s^{(\ell_i)}$ is the $\ell_i$-th degree of the irrep feature of node $s$, the source node; $\otimes$ is the tensor product; $Y^{(\ell_f)}(\hat{\mathbf{r}}_{ts})$ is the $\ell_f$-th degree spherical harmonic projection of $\hat{\mathbf{r}}_{ts}$; and $\hat{\mathbf{r}}_{ts}$ is the (normalized) relative position vector. Then, the message is given by

$$m_{ts}^{(\ell_o)} = \sum_{\ell_i,\ell_f} W_{\ell_i,\ell_f,\ell_o}\Big(x_s^{(\ell_i)} \otimes Y^{(\ell_f)}(\hat{\mathbf{r}}_{ts})\Big)^{(\ell_o)}.$$

Traditionally, this involves an ordinary $SO(3)$ tensor product, which is decomposed into its irreps using the Clebsch-Gordon coefficients, $C^{(\ell_o,m_o)}_{(\ell_i,m_i),(\ell_f,m_f)}$:

$$m^{(\ell_o)}_{ts} = \sum_{\ell_i,\ell_f} W_{\ell_i,\ell_f,\ell_o} \bigoplus_{m_o} \sum_{m_i,m_f} \left(x^{(\ell_i)}_s\right)_{m_i} C^{(\ell_o,m_o)}_{(\ell_i,m_i),(\ell_f,m_f)} \left(Y^{(\ell_f)}(\hat{\mathbf{r}}_{ts})\right)_{m_f}.$$

The tensor product, $\otimes$, has been exchanged for the proper Clebsch-Gordon coefficient. The direct sum, $\bigoplus$, is included because the space of tensors of degree $\ell_o$ has a basis indexed by $m_o$.

However, the full $SO(3)$ tensor product is compute-intensive, $O(L^6)$, where $L$ is the tensor degree. In practice, this means that only tensors up to degree 2 or 3 are used, which is especially unfortunate because higher-order tensors allow more precise representation of angular information. To lessen this computational burden, eSCN rotates the irrep features, $x^{(\ell_i)}_s$, of a node $x_s$, along with the relative position vector, $\hat{\mathbf{r}}_{ts}$, by a rotation chosen so that the relative position vector aligns with the $y$-axis. Thus, it may be interpreted that eSCN changes the irreps into a more convenient basis, computes the convolution, and changes back into the original basis. This requires multiplication by a change-of-basis Wigner D-matrix before and its inverse after, but it is worthwhile because it reduces the convolution to $O(L^3)$ by sparsifying the Clebsch-Gordon coefficients. In particular, eSCN guarantees that the output coefficient is nonzero only if both $m_i = \pm m_o$ and $m_f = 0$ hold. The output tensors of some order, therefore, are linear combinations of input tensors of that order. It is not necessary to perform tensor multiplication and decomposition. The non-zero Clebsch-Gordon coefficients can then be denoted $C^{(\ell_o,m)}_{(\ell_i,m),(\ell_f,0)}$, with eSCN further guaranteeing that $C^{(\ell_o,m)}_{(\ell_i,m),(\ell_f,0)} = C^{(\ell_o,-m)}_{(\ell_i,-m),(\ell_f,0)}$, and $C^{(\ell_o,-m)}_{(\ell_i,m),(\ell_f,0)} = -C^{(\ell_o,m)}_{(\ell_i,-m),(\ell_f,0)}$. Thus, eSCN guarantees that all but a few Clebsch-Gordon coefficients will be zero, and provides simple formulas involving those that remain.

More formally, let $R$ be a rotation matrix chosen so that $R \cdot \hat{\mathbf{r}}_{ts} = (0,1,0)$, and let $D^{(\ell)}$ be a Wigner D-matrix representation of $R$ of degree $\ell$ (which would more properly read $D^{(\ell)}(R)$, but which is truncated for readability). Then, the message can be written

$$m^{(\ell_o)}_{ts} = \left(D^{(\ell_o)}\right)^{-1} \sum_{\ell_i,\ell_f} W_{\ell_i,\ell_f,\ell_o} \left(D^{(\ell_i)}x^{(\ell_i)}_s \otimes Y^{(\ell_f)}(R \cdot \hat{\mathbf{r}}_{ts})\right)^{(\ell_o)}.$$

Rewriting the tensor products using the Clebsch-Gordon coefficients yields

$$m^{(\ell_o)}_{ts} = \left(D^{(\ell_o)}\right)^{-1} \sum_{\ell_i,\ell_f} W_{\ell_i,\ell_f,\ell_o} \bigoplus_{m_o} \sum_{m_i,m_f} \left(D^{(\ell_i)}x^{(\ell_i)}_s\right)_{m_i} C^{(\ell_o,m_o)}_{(\ell_i,m_i),(\ell_f,m_f)} \left(Y^{(\ell_f)}(R \cdot \hat{\mathbf{r}}_{ts})\right)_{m_f}.$$

Since the Clebsch-Gordon coefficients are non-zero only when $m_f = 0$, there is no need to sum over $m_f$:

$$m^{(\ell_o)}_{ts} = \left(D^{(\ell_o)}\right)^{-1} \sum_{\ell_i,\ell_f} W_{\ell_i,\ell_f,\ell_o} \bigoplus_{m_o} \sum_{m_i} \left(D^{(\ell_i)}x^{(\ell_i)}_s\right)_{m_i} C^{(\ell_o,m_o)}_{(\ell_i,m_i),(\ell_f,0)} \left(Y^{(\ell_f)}(R \cdot \hat{\mathbf{r}}_{ts})\right)_0.$$

The rotation $R$ has been chosen so that $R \cdot \hat{\mathbf{r}}_{ts}$ yields a simple result, so the spherical harmonic term drops out:

$$m^{(\ell_o)}_{ts} = \left(D^{(\ell_o)}\right)^{-1} \sum_{\ell_i,\ell_f} W_{\ell_i,\ell_f,\ell_o} \bigoplus_{m_o} \sum_{m_i} \left(D^{(\ell_i)}x^{(\ell_i)}_s\right)_{m_i} C^{(\ell_o,m_o)}_{(\ell_i,m_i),(\ell_f,0)}.$$

Now, since one of the rotations on the righthand side has already dropped out, to make things simpler, define $\tilde{x}^{(\ell_i)}_s = D^{(\ell_i)}x^{(\ell_i)}_s$, yielding

$$m^{(\ell_o)}_{ts} = \left(D^{(\ell_o)}\right)^{-1} \sum_{\ell_i,\ell_f} W_{\ell_i,\ell_f,\ell_o} \bigoplus_{m_o} \sum_{m_i} \left(\tilde{x}^{(\ell_i)}_s\right)_{m_i} C^{(\ell_o,m_o)}_{(\ell_i,m_i),(\ell_f,0)}.$$

Since the Clebsch-Gordon coefficients are non-zero only when $m_i = \pm m_o$, the second summation can be omitted:

$$m^{(\ell_o)}_{ts} = \left(D^{(\ell_o)}\right)^{-1} \sum_{\ell_i,\ell_f} W_{\ell_i,\ell_f,\ell_o} \bigoplus_{m_o} \left(\left(\tilde{x}^{(\ell_i)}_s\right)_{m_o} C^{(\ell_o,m_o)}_{(\ell_i,m_o),(\ell_f,0)} + \left(\tilde{x}^{(\ell_i)}_s\right)_{-m_o} C^{(\ell_o,m_o)}_{(\ell_i,-m_o),(\ell_f,0)}\right).$$

Now, to avoid the need to sum over $\ell_f$, rather than learning parameters $W_{\ell_i,\ell_f,\ell_o}$, eSCN learns parameters $\tilde{W}_m^{(\ell_i,\ell_o)}$, defined for $m \geq 0$ as

$$\tilde{W}_m^{(\ell_i,\ell_o)} = \sum_{\ell_f} W_{\ell_i,\ell_f,\ell_o} C_{(\ell_i,m),(\ell_f,0)}^{(\ell_o,m)} = \sum_{\ell_f} W_{\ell_i,\ell_f,\ell_o} C_{(\ell_i,-m),(\ell_f,0)}^{(\ell_o,-m)},$$

and for $m < 0$ as

$$\tilde{W}_m^{(\ell_i,\ell_o)} = \sum_{\ell_f} W_{\ell_i,\ell_f,\ell_o} C_{(\ell_i,-m),(\ell_f,0)}^{(\ell_o,m)} = -\sum_{\ell_f} W_{\ell_i,\ell_f,\ell_o} C_{(\ell_i,m),(\ell_f,0)}^{(\ell_o,-m)}.$$

There exists a linear bijection between $W$ and $\tilde{W}$, so this parameterization loses no information. Finally, defining

$$\begin{aligned}
\left(y_{ts}^{\ell_i,\ell_o}\right)_{m_o} &= \tilde{W}_{m_o}^{(\ell_i,\ell_o)}\left(\tilde{x}_s^{\ell_i}\right)_{m_o} - \tilde{W}_{-m_o}^{(\ell_i,\ell_o)}\left(\tilde{x}_s^{\ell_i}\right)_{-m_o}, & m &> 0; \\
\left(y_{ts}^{(\ell_i,\ell_o)}\right)_{m_o} &= \tilde{W}_{m_o}^{(\ell_i,\ell_o)}\left(\tilde{x}_s^{\ell_i}\right)_{-m_o} + \tilde{W}_{-m_o}^{(\ell_i,\ell_o)}\left(\tilde{x}_s^{\ell_i}\right)_{m_o}, & m &< 0; \\
\left(y_{ts}^{(\ell_i,\ell_o)}\right)_{m_o} &= \tilde{W}_{m_o}^{(\ell_i,\ell_o)}\left(\tilde{x}_s^{\ell_i}\right)_{m_o}, & m &= 0,
\end{aligned}$$

the message equation can very concisely be written as

$$m_{ts}^{\ell_o} = \left(D^{\ell_o}\right)^{-1} \sum_{\ell_i} \bigoplus_{m_o} \left(y_{ts}^{\ell_i,\ell_o}\right)_{m_o}.$$

There is another way to interpret this that is perhaps more intuitive. Fixing the direction of the relative position vector, $\hat{\mathbf{r}}_{st}$ leaves a single rotational degree of freedom: the roll rotation about this axis. Thus, eSCN reduces $SO(3)$ convolution to $SO(2)$ convolution. More formally, define the colatitude angle $\theta \in [0, \pi]$ and longitudinal angle $\phi \in [0, 2\pi]$. Now, using Legendre polynomials $P_m^{(\ell)}(\theta)$, which depend only on the colatitude angle, $\theta$, the real spherical harmonic basis functions can be written

$$Y_m^{(\ell)}(\theta, \phi) = P_m^{(\ell)}(\theta)e^{im\phi}.$$

Aligning the relative position vector with the $y$-axis fixes $\theta$, leaving behind basis functions of the form $e^{im\phi}$, which are the circular harmonic basis functions, used in $SO(2)$ convolution. A convolution about $\phi$ can take advantage of the Convolution Theorem, reducing convolution to point-wise multiplication, further harvesting efficiency gains.

# E  ADDITIONAL BACKGROUND

Quantum mechanics is most often approached as the study of the Schrödinger equation, a linear partial differential equation whose solutions are called wavefunctions. In this paper, as is typical in molecular modeling, the time-independent Schrödinger equation will be used:

$$H\psi = E\psi,$$

where $H$, the Hamiltonian operator, corresponds to the total energy of the system, $E$ is an eigenvalue of $H$ corresponding to the energy of $\psi$, and $\psi$ is an eigenfunction of $H$, also called a wavefunction, or a solution to the Schrödinger equation.

As alluded to, the Hamiltonian operator contains all the information regarding the kinetic and potential energies for all particles of a system. Thus,

$$H = -\frac{1}{2}\sum_{i=1}^{N}\nabla_i^2 - \frac{1}{2}\sum_{A=1}^{M}\frac{1}{M_A}\nabla_A^2 - \sum_{i=1}^{N}\sum_{A=1}^{M}\frac{Z_a}{r_{iA}}$$
$$+ \sum_{i=1}^{N}\sum_{j>1}\frac{1}{r_{ij}} + \sum_{A=1}^{M}\sum_{B>A}\frac{Z_A Z_B}{R_{AB}},$$

where the first summation is due to the kinetic energy of the electrons, the second summation is due to the kinetic energy of the nuclei, the first double summation is due to the potential of the attraction between electrons and nuclei, the second double summation is due to the potential of the repulsion between electrons and electrons, and the third double summation is due to the potential of the repulsion between nuclei.

This formula is unwieldy, however, and it can be simplified significantly without discarding much information that would be useful in chemical prediction. The mass difference between electrons and protons, which is the minimum mass difference between electrons and nuclei, is more than 3 orders of magnitude. Thus, given the same kinetic energy, electrons will be traveling several times faster than nuclei. From the perspective of the electrons, the nuclei are nearly fixed, and from the perspective of the nuclei, the electrons change their position instantaneously. Therefore, it suffices to consider the positions of the nuclei fixed, setting

$$-\frac{1}{2}\sum_{A=1}^{M}\frac{1}{M_A}\nabla_A^2,$$

nuclear kinetic energy, to 0, and

$$\sum_{A=1}^{M}\sum_{B>A}\frac{Z_A Z_B}{R_{AB}},$$

nuclear repulsive potential, to a constant. This is called the Born-Oppenheimer Approximation, and it leaves behind the so-called electronic Hamiltonian,

$$H_{\text{elec}} = -\frac{1}{2}\sum_{i=1}^{N}\nabla_i^2 - \sum_{i=1}^{N}\sum_{A=1}^{M}\frac{Z_a}{r_{iA}} + \sum_{i=1}^{N}\sum_{j>1}\frac{1}{r_{ij}},$$

which can more succinctly be written

$$H_{\text{elec}} = T + V_{\text{Ne}} + V_{\text{ee}}.$$

The electronic Hamiltonian is still solved for its eigenfunctions, giving

$$H_{\text{elec}}\Psi_{\text{elec}}(\mathbf{r}_1, \ldots, \mathbf{r}_N) = E_{\text{elec}}\Psi_{\text{elec}}(\mathbf{r}_1, \ldots, \mathbf{r}_N),$$

ignoring for simplicity's sake electron spin to write $\Psi_{\text{elec}}$ as a function of the positions of the electrons only. For readability, the subscripts of $H_{\text{elec}}$, $E_{\text{elec}}$, and $\Psi_{\text{elec}}$ are henceforth omitted.

For a system of $N$ electrons, $\Psi$ is a function of $3N$ arguments, one for each spatial dimension of each electron. The Hartree-Fock Approximation, also called the Hartree Product, further simplifies

this equation by decomposing the wavefunction into the product of $N$ wave functions, each of three arguments, corresponding to each electron individually:

$$\Psi(\mathbf{r}_1, \ldots, \mathbf{r}_n) \approx \psi_1(\mathbf{r}_1) \cdots \psi_N(\mathbf{r}_N).$$

It is impossible to observe the wavefunction itself, but the wavefunction can be used to derive the probability of observing the system's electrons anywhere in space. In particular, the probability of observing the electrons at positions $\mathbf{r}_1$ through $\mathbf{r}_N$ is the square of the amplitude of the wavefunction with $\mathbf{r}_1$ through $\mathbf{r}_N$ as input:

$$|\Psi(\mathbf{r}_1, \ldots, \mathbf{r}_N)|^2 = \Psi^\dagger(\mathbf{r}_1, \ldots, \mathbf{r}_N)\Psi(\mathbf{r}_1, \ldots, \mathbf{r}_N).$$

Given the Hartree-Fock Approximation, this equation can be converted into an electron density function that sums over the wavefunction of each electron individually,

$$n(\mathbf{r}) = 2 \sum_{i=1}^{N} \psi_i^\dagger(\mathbf{r})\psi_i(\mathbf{r}),$$

multiplying by 2 to account for both spin up and spin down, which were neglected previously. This means that $n(\mathbf{r})$ gives the density of electrons at a point in space, $\mathbf{r}$. This is a function of only 3 inputs, but it contains much of the information that is observable from the full wavefunction, which, recall, is a function of $3N$ inputs.

Density Functional Theory (DFT), which allows the electron density function, $n(\mathbf{r})$, to be exploited, rests on two fundamental theorems. First, the ground-state energy, $E_0$, which is the smallest eigenvalue of the Hamiltonian and corresponds to the energy of the lowest-energy wavefunction, is a unique functional of the electron density function. Second, the electron density that minimizes the energy of this functional is the true electron density, which means that it is the electron density that corresponds to the full solution of the Schrödinger equation. Therefore, after the problem has been reduced from one of $3N$ inputs to one of 3 inputs, significant information about the original problem can still be found.

It is worth examining the first statement in more detail. Recall that a functional is a function that maps functions to scalars. For example, $F$, defined by

$$F[f(x)] = \int_{-1}^{1} f(x)dx,$$

is a functional that takes in an arbitrary real-valued function and gives out its integral from $-1$ to $1$. Thus, for example, if $f(x) = x^2 + 1$, then $F[f(x)] = \frac{8}{3}$. The first statement, then, holds that there exists a functional that uniquely determines the ground state energy of a particular electron density function. It promises that no more information is needed. Written as an equation, $E_0 = F(n(\mathbf{r}))$.

However, this set-up still relies on some means of finding the individual-electron wave function. This is provided by the Kohn-Sham Equation:

$$\left[\frac{1}{2}\nabla^2 + V(\mathbf{r}) + V_{\mathrm{H}}(\mathbf{r}) + V_{\mathrm{XC}}(\mathbf{r})\right]\psi_i(\mathbf{r}) = \epsilon_i\psi_i(\mathbf{r}).$$

The meaning of each term is as follows. The value $\frac{1}{2}\nabla^2$ is kinetic energy. The value $V(\mathbf{r})$ is the potential due to the interaction between the electron and the nuclei. The value $V_{\mathrm{H}}(\mathbf{r})$, called the Hartree potential, is defined

$$V_{\mathrm{H}}(\mathbf{r}) = \int \frac{n(\mathbf{r}')}{|\mathbf{r} - \mathbf{r}'|}d^3\mathbf{r}',$$

and it is due to the Coulomb repulsion between the electron and the electron density function, defined by all electrons in the system. This means that the Hartree potential includes the Coulomb repulsion between the electron and itself, since the electron itself is included in the electron density function, $n(\mathbf{r})$. This is an unphysical result, and it is one of several effects accounted for in the exchange-correlation potential, $V_{\mathrm{XC}}(\mathbf{r})$, which is defined

$$V_{\mathrm{XC}}(\mathbf{r}) = \frac{\delta E_{\mathrm{XC}}(\mathbf{r})}{\delta n(\mathbf{r})}.$$

This is the functional derivative of the exchange-correlation energy with respect to electron density. Exchange-correlation potential is due to quantum-chemical effects, and its form is not known because the exact form of $E_{\mathrm{XC}}(\mathbf{r})$ is not known. The Kohn-Sham Equation can be summarized as

$$H_{\mathrm{KS}}\psi_i(\mathbf{r}) = \epsilon_i\psi_i(\mathbf{r}).$$

It is evident from the definitions of the terms in the Kohn-Sham Equation that the present approach is circular. The Kohn-Sham Equation count on the Hartree potential, $V_H(\mathbf{r})$. The Hartree potential counts on the electron density function, $n(\mathbf{r})$. The electron density function counts on the single-electron wavefunctions, $\psi_i(\mathbf{r})$. And the single-electron wavefunctions count on the Kohn-Sham Equation.

This is no problem. The following process leverages this circularity to check the soundness of a particular $n(\mathbf{r})$:

1. Define an initial trial electron density function, $n_{\mathrm{trial}}(\mathbf{r})$.
2. Solve the Kohn-Sham Equation using the trial electron density function, $n_{\mathrm{trial}}(\mathbf{r})$, to find the single-electron wavefunctions, $\psi_i(\mathbf{r})$.
3. Calculate the electron density function, $n_{\mathrm{KS}}(\mathbf{r})$, implied by these single-electron wavefunctions, by $n_{\mathrm{KS}}(\mathbf{r}) = 2\sum_{i=1}^{N}\psi_i^{\dagger}(\mathbf{r})\psi_i(\mathbf{r})$.
4. Compare the calculated electron density, $n_{\mathrm{KS}}(\mathbf{r})$, with the trial electron density, $n_{\mathrm{trial}}(\mathbf{r})$. If the two densities are the same, or nearly so, then this electron density is accepted as correct, and it can be used to find the ground-state energy, $E_0$. If not, the trial electron density is updated somehow, and the process repeats.

To make this process simpler and more efficient, it is common to represent the single-electron wavefunctions as linear combinations of some predefined basis set $\{\phi_\alpha(\mathbf{r})\}_{\alpha=1}^{B}$, where $B$ is defined as the cardinality of the basis. Often, the basis set is composed of atomic orbitals, especially Gassian-type orbitals, which are particularly convenient in calculation. The expansion coefficients of these wavefunctions can be organized in a matrix $\mathbf{C} \in \mathbb{R}^{B \times N}$, where each column $i$ contains the coefficients of wavefunction $i$. Each wave function can then be recovered as

$$\psi_i(\mathbf{r}) = \sum_{\alpha=1}^{B}\mathbf{C}_{\alpha i}\phi_\alpha(\mathbf{r}).$$

This permits the Kohn-Sham Equation as a whole to be written in matrix form. Recall the form $H_{\mathrm{KS}}\psi_i(\mathbf{r}) = \epsilon_i\psi_i(\mathbf{r})$. Now, it is possible to rewrite $\psi_i(\mathbf{r})$ according to its decomposition in the basis:

$$H_{\mathrm{KS}}\sum_{\alpha=1}^{B}\mathbf{C}_{\alpha i}\phi_\alpha(\mathbf{r}) = \epsilon_i\sum_{\alpha=1}^{B}\mathbf{C}_{\alpha i}\phi_\alpha(\mathbf{r}).$$

Now, for any $\phi_\beta(\mathbf{r})$ in the basis, left-multiplying both sides by $\phi_\beta^{\dagger}(\mathbf{r})$ yields

$$\phi_\beta^{\dagger}(\mathbf{r})H_{\mathrm{KS}}\sum_{\alpha=1}^{B}\mathbf{C}_{\alpha i}\phi_\alpha(\mathbf{r}) = \phi_\beta^{\dagger}(\mathbf{r})\epsilon_i\sum_{\alpha=1}^{B}\mathbf{C}_{\alpha i}\phi_\alpha(\mathbf{r}),$$

and integrating with respect to $\mathbf{r}$ yields

$$\int\phi_\beta^{\dagger}(\mathbf{r})H_{\mathrm{KS}}\sum_{\alpha=1}^{B}\mathbf{C}_{\alpha i}\phi_\alpha(\mathbf{r})d\mathbf{r} = \int\phi_\beta^{\dagger}(\mathbf{r})\epsilon_i\sum_{\alpha=1}^{B}\mathbf{C}_{\alpha i}\phi_\alpha(\mathbf{r})d\mathbf{r}.$$

More succinctly, defining $(H_{\mathrm{KS}})_{\beta\alpha} = \int\phi_\beta^{\dagger}H_{\mathrm{KS}}\phi_\alpha = \langle\phi_\beta|H_{\mathrm{KS}}|\phi_\alpha\rangle$ and $S_{\beta\alpha} = \int\phi_\beta^{\dagger}\phi_\alpha = \langle\phi_\beta|\phi_\alpha\rangle$, this is

$$\sum_{\alpha=1}^{B}(H_{\mathrm{KS}})_{\beta\alpha}\mathbf{C}_{\alpha i} = \epsilon_i\sum_{\alpha=1}^{B}S_{\beta\alpha}\mathbf{C}_{\alpha i}.$$

This allows the construction of the matrix $\mathbf{H}$, with elements $\mathbf{H}_{\beta\alpha}$ defined by $\mathbf{H}_{\beta\alpha} = (H_{\mathrm{KS}})_{\beta\alpha} = \langle\phi_\beta|H_{\mathrm{KS}}|\phi_\alpha\rangle$, and the matrix $\mathbf{S}$, with elements $\mathbf{S}_{\beta\alpha} = \langle\phi_\beta|\phi_\alpha\rangle$. The matrix $\mathbf{H}$ represents the

Hamiltonian matrix, which depends on the coefficient matrix $\mathbf{C}$ and is calculated using a method called Density-Fitting, whose time complexity is $O(B^3)$. The matrix $\mathbf{S}$ is the overlap matrix, which depends on the basis set $\{\phi_\alpha(\mathbf{r})\}_{\alpha=1}^{B}$ and accounts for its non-orthogonality. This means that if the basis set is orthonormal, $\mathbf{S}$ is the identity. The Kohn-Sham Equation in matrix form is therefore

$$\mathbf{HC} = \mathbf{SC}\epsilon,$$

or

$$\mathbf{H}(\mathbf{C})\mathbf{C} = \mathbf{SC}\epsilon,$$

making clearer that $\mathbf{H}$ depends on $\mathbf{C}$, where $\epsilon$ is a diagonal matrix containing the orbital energies. This forms a generalized eigenvalue problem, the principal difficulty of which is the dependence of $\mathbf{H}$ on $\mathbf{C}$.

As a result, traditional DFT uses the Self-Consistent Field method (SCF), which iteratively refines the coefficient matrix by approximating the Hamiltonian. Using superscripts to denote the iteration to which each matrix belongs, at each iteration, the Kohn-Sham Equation is

$$\mathbf{H}^{(k)}\big(\mathbf{C}^{(k-1)}\big)\mathbf{C}^{(k)} = \mathbf{SC}^{(k)}\epsilon^{(k)}.$$

Therefore, $\mathbf{H}^{(k)}$ is computed using $\mathbf{C}^{k-1}$, and the generalized eigenvalue problem is solved for $\mathbf{C}^{(k)}$ and $\epsilon^{(k)}$. This process continues until convergence, which is formalized as $\|\mathbf{H}^{(k+1)} - \mathbf{H}^{(k)}\| \leq \delta$.

The objective of Hamiltonian prediction is to eliminate the need for this computationally expensive SCF iteration by directly estimating the target Hamiltonian, $\mathbf{H}^\star$, for a given molecular structure, $\mathcal{M}$. To predict the Hamiltonian, a machine learning model $\hat{\mathbf{H}}_\theta(\mathcal{M})$ is parameterized by $\theta$, guided by an optimization process defined as

$$\theta^\star = \arg\min_\theta \frac{1}{|\mathcal{D}|} \sum_{(\mathcal{M}, \mathbf{H}^\star_\mathcal{M}) \in \mathcal{D}} \mathrm{dist}(\hat{\mathbf{H}}_\theta(\mathcal{M}), \mathbf{H}^\star_\mathcal{M}),$$

where $\mathcal{D}$ is the dataset, $|\mathcal{D}|$ is its cardinality, and $\mathrm{dist}(\cdot, \cdot)$ is a predefined metric. Machine learning models hold the potential to drastically improve the efficiency of DFT calculations without sacrificing accuracy.

### E.1 EXAMPLE

In the example section, we explore the molecular structure and basis set details of water ($H_2O$), which consists of one oxygen and two hydrogen atoms. We expand each electron's wavefunctions using a basis set, which, in practice, means each atom is represented using a predefined basis set that collectively describes a set of electrons. Specifically, we use the STO-3G minimal basis set, where oxygen is described by five basis functions ($1s, 2s, 2p_x, 2p_y, 2p_z$), and each hydrogen atom has two basis functions ($1s$ for each). Altogether, this totals seven basis functions.

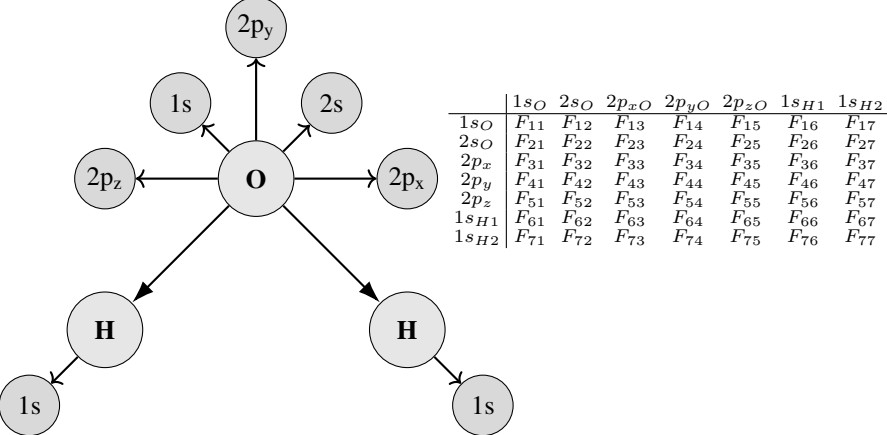

Figure 8: Molecular structure and basis functions for water ($H_2O$).

When considering the effect of rotations on the Hamiltonian matrix, it is important to understand how the matrix elements transform under such operations.

For instance, if we consider a rotation that affects only the $2p_x$, $2p_y$, and $2p_z$ orbitals of the oxygen atom, the corresponding submatrix of the Hamiltonian $\mathbf{H}_p$ within the $2p$ orbital space would transform as:

$$\mathbf{H}_p' = \mathbf{R}_p \mathbf{H}_p \mathbf{R}_p^\top$$

where $\mathbf{H}_p$ is the submatrix containing the interactions between the $2p_x$, $2p_y$, and $2p_z$ orbitals. The rotation matrix $\mathbf{R}_p$ is specific to the rotation in the $2p$ orbital space.

|        | $1s_O$   | $2s_O$   | $2p_x$   | $2p_y$   | $2p_z$   | $1s_{H1}$ | $1s_{H2}$ |
|--------|----------|----------|----------|----------|----------|-----------|-----------|
| $1s_O$    | $F_{11}$ | $F_{12}$ | $F_{13}$ | $F_{14}$ | $F_{15}$ | $F_{16}$  | $F_{17}$  |
| $2s_O$    | $F_{21}$ | $F_{22}$ | $F_{23}$ | $F_{24}$ | $F_{25}$ | $F_{26}$  | $F_{27}$  |
| $2p_x$    | $F_{31}$ | $F_{32}$ | $F_{33}$ | $F_{34}$ | $F_{35}$ | $F_{36}$  | $F_{37}$  |
| $2p_y$    | $F_{41}$ | $F_{42}$ | $F_{43}$ | $F_{44}$ | $F_{45}$ | $F_{46}$  | $F_{47}$  |
| $2p_z$    | $F_{51}$ | $F_{52}$ | $F_{53}$ | $F_{54}$ | $F_{55}$ | $F_{56}$  | $F_{57}$  |
| $1s_{H1}$ | $F_{61}$ | $F_{62}$ | $F_{63}$ | $F_{64}$ | $F_{65}$ | $F_{66}$  | $F_{67}$  |
| $1s_{H2}$ | $F_{71}$ | $F_{72}$ | $F_{73}$ | $F_{74}$ | $F_{75}$ | $F_{76}$  | $F_{77}$  |

|        | $2p_x$    | $2p_y$    | $2p_z$    |
|--------|-----------|-----------|-----------|
| $2p_x$ | $F_{33}'$ | $F_{34}'$ | $F_{35}'$ |
| $2p_y$ | $F_{43}'$ | $F_{44}'$ | $F_{45}'$ |
| $2p_z$ | $F_{53}'$ | $F_{54}'$ | $F_{55}'$ |

This transformation ensures that the physical properties described by the Hamiltonian remain consistent under rotational operations, a fundamental requirement for accurately modeling molecular systems.

## F  DIPOLE MOMENT AND ELECTRONIC SPATIAL EXTENT

In this section, we describe how to compute the dipole moment and the electronic spatial extent using the Kohn-Sham orbitals derived from the Kohn-Sham Hamiltonian.

The Kohn-Sham equations are defined as

$$\left[ -\frac{\hbar^2}{2m} \nabla^2 + V_{\text{eff}}(\mathbf{r}) \right] \psi_i(\mathbf{r}) = \epsilon_i \psi_i(\mathbf{r}), \tag{4}$$

where $\psi_i(\mathbf{r})$ denotes the Kohn-Sham orbitals, $\epsilon_i$ are the orbital energies, and $V_{\text{eff}}(\mathbf{r})$ is the effective potential. The electron density is then expressed as

$$\rho(\mathbf{r}) = \sum_i^{\text{occ}} |\psi_i(\mathbf{r})|^2. \tag{5}$$

The dipole moment $\mathbf{d}$ is computed as the sum of electronic and nuclear contributions. The electronic contribution is given by

$$\mathbf{d}^{\text{elec}} = -e \int \mathbf{r} \rho(\mathbf{r}) \, d\mathbf{r}, \tag{6}$$

and the nuclear contribution is

$$\mathbf{d}^{\text{nuc}} = -e \sum_A Z_A \mathbf{R}_A, \tag{7}$$

where $e$ represents the elementary charge, $Z_A$ denotes the atomic number of nucleus $A$, and $\mathbf{R}_A$ is the position vector of nucleus $A$. The total dipole moment is therefore the sum of these two components:

$$\mathbf{d} = \mathbf{d}^{\text{elec}} + \mathbf{d}^{\text{nuc}}. \tag{8}$$

Next, we define the electronic spatial extent, denoted by $\langle r^2 \rangle$, which provides a measure of the spatial distribution of the electron density. It is calculated as

$$\langle r^2 \rangle = \int r^2 \rho(\mathbf{r}) \, d\mathbf{r}. \tag{9}$$

To compute these quantities in practice, one often works in a basis set representation. The electron density matrix $P_{\mu\nu}$ is defined as

$$P_{\mu\nu} = 2 \sum_i^{\text{occ}} C_{i\mu} C_{i\nu}, \tag{10}$$

where $C_{i\mu}$ are the coefficients of the molecular orbitals in terms of the basis functions $\phi_\mu$. The dipole integrals $\mu_\alpha^{\mu\nu}$ for a Cartesian direction $\alpha$ are expressed as

$$\mu_\alpha^{\mu\nu} = \int \phi_\mu(\mathbf{r}) \, r_\alpha \, \phi_\nu(\mathbf{r}) \, d\mathbf{r}. \tag{11}$$

The electronic contribution to the dipole moment in the basis set representation is then given by

$$d_\alpha^{\text{elec}} = -e \sum_{\mu\nu} P_{\mu\nu} \mu_\alpha^{\mu\nu}. \tag{12}$$

Similarly, the electronic spatial extent in the basis set representation is computed using the integral

$$\langle r^2 \rangle^{\mu\nu} = \int \phi_\mu(\mathbf{r}) \, r^2 \, \phi_\nu(\mathbf{r}) \, d\mathbf{r}, \tag{13}$$

leading to the final expression

$$\langle r^2 \rangle = \sum_{\mu\nu} P_{\mu\nu} \langle r^2 \rangle^{\mu\nu}. \tag{14}$$

## G    GROUP THEORY

If a function is equivariant to the action of a group, it does not matter whether the group acts on the function's input or output. More formally, for vector spaces $V$ and $W$ equipped with arbitrary group representations $D_V(g)$ and $D_W(g)$, a function $f : V \to W$ is equivariant to $G$ if

$$f(D_V(g)v) = D_W(g)f(v)$$

for all $g \in G$ and for all $v \in V$. In the case of SE(3), a function $f$ is equivariant if the output is the same whether the input is slid and rotated, then put through $f$, or whether the input is put through $f$, then slid and rotated in the same way. A function $f$ is invariant to a group $G$ if

$$D_W(g) = e,$$

the identity element in $W$, for all $g \in G$. In the case of SE(3), a function $f$ is equivariant if it gives the same output no matter how its input is slid and rotated. Equivariance is a fundamental notion in the modeling of physical systems. In the context of this paper, it is desirable that the function that predicts the Hamiltonian matrix be SE(3)-equivariant, reflecting that the molecule's energetic properties are equivalent if the molecule is rotated or translated.

Group representations are an instance of the more general notion of group homomorphisms. Given a group $G$ with group operation $\circ$ and another group $H$ with group operation $*$, a group homomorphism is a map $\rho : G \to H$ such that

$$\rho(g_1 \circ g_2) = \rho(g_1) * \rho(g_2).$$

A group homomorphism, then, must preserve the structure of the group, which means that it does not matter whether the group operation is performed in $G$ or $H$. Note, however, that a homomorphism might respect the group structure only trivially. For example, $\rho : G \to H$, defined by $\rho(g) = e$, is a trivial group homomorphism. A group representation is simply a group homomorphism where $H = V$ is some vector space. In plainer language, this means that a group representation is a group written as a set of matrices, whose group operation is matrix multiplication. It is always the case that $V \subseteq GL$, since any matrix that is degenerate or non-square lacks an inverse and therefore fails to satisfy the inverse axiom.

Two group representations $D$ and $D'$ are equivalent if there is a fixed matrix $P$ such that $D(g) = P^{-1}D'(g)P$ for all $g \in G$. In this case, $D$ and $D'$ can be interpreted as the same representation defined with respect to different bases. A representation $D$ is reducible if it acts on independent subspaces of $V$; otherwise, it is irreducible. An irreducible representation is called an irrep. More formally, $D$ is reducible if

$$D(g) = P^{-1} \begin{bmatrix} D^{(\ell_0)}(g) & & & \\ & D^{(\ell_1)}(g) & & \\ & & \ddots & \end{bmatrix} P = P^{-1}\Big( \bigoplus_i D^{(\ell_i)}(g) \Big)P,$$

which means that it $D$ is block-diagonal with respect to some basis. It is convenient to decompose group representations into irreducible representations because this reduces the group operation calculation to several smaller independent calculations. Irreps are the atoms of group representations in the sense that arbitrary representations can be composed with the direct sum of irreps. The irreps of $SO(3)$ are called Wigner D-matrices, with $D^{(\ell)}(g)$ denoting a Wigner D-matrix representation of $g$ of degree $\ell$. Wigner D-matrices are of size $(2\ell + 1) \times (2\ell + 1)$, with higher-degree representations allowing for more precise handling of angular information.

## H EXENSIVE DETAILS ON PUBCHEMQH

Here, we present a detailed comparison between the PubChemQH dataset and the curated QH9 dataset. This comparison aims to highlight the key differences and similarities. Additionally, we provide a comprehensive analysis of the atom number distribution within the PubChemQH dataset, supported by a Figure 9.

Table 10: Comparison of PubChemQH and QH9

| Feature | PubChemQH | QH9 |
|---|---|---|
| Source | PubChem Database | QM9 |
| Number of Molecules | 50,321 | 130,831 (QH9-stable) |
| Functional | B3LYP | B3LYP |
| Basis Set | Def2TZVP | Def2SVP |
| SCF Convergence Tolerance | $10^{-8}$ | $10^{-13}$ |
| SCF Gradient threshold: | $10^{-4}$ | $3.16 \times 10^{-4}$ |
| Grid Density Level | 3 | 3 |
| Mean of Node Number | 61.85 | 18 |
| Mean of Hamiltonian Size | 1025 | 141 |

## I EXTENSIVE DETAILS ON WANET

We here provide extensive details on the motivations of the WANet architecture.The WANet architecture builds upon well-established equivariant neural network designs, which have been extensively studied and validated in the field.

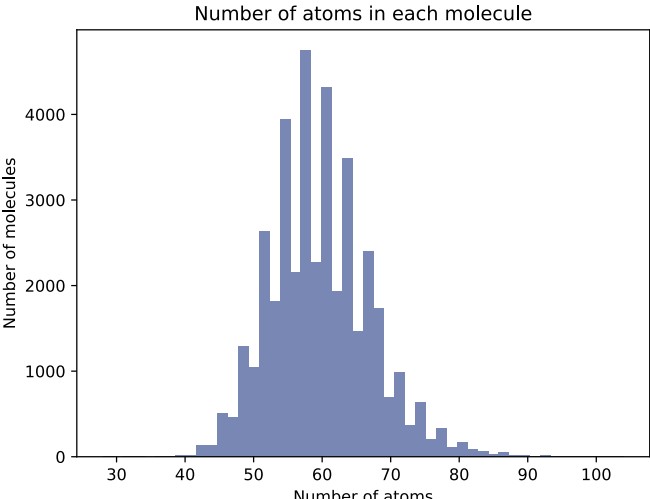

Figure 9: Node atom distribution for PubChemQH Dataset

We provide a visual representation illustrating WANet's architecture and its key components, which is given in Figure10 . This visual aid will help readers better understand how WANet enhances the scalability of Hamiltonian prediction for large molecular systems. Each architectural element was carefully chosen to address specific computational and physical challenges at scale:

First, the motivation behind WANet's architecture stems from a critical limitation in existing Hamiltonian prediction methods—their inability to scale to larger molecular systems. When using larger basis sets, such as Def2-TZVP, higher-order irreducible representations are required to accurately capture angular dependencies in molecular orbitals. This poses a severe computational challenge for traditional SE(3)-equivariant methods, including QHNet, which become extremely expensive due to their computational complexity. WANet addresses this bottleneck by introducing SO(2) convolutions, which reduce computational complexity from $O(L^3)$ to $O(L^6)$ . This improvement enables WANet to process high tensor degrees efficiently. Without SO(2) convolutions, handling the scale of our PubChemQH dataset and larger systems would be significantly more challenging and computationally expensive.

Second, large molecular systems exhibit fundamentally different physics at various distance scales. As molecular size increases, long-range interactions become more prominent. WANet's Mixture-of-Experts architecture is designed to model this complex physics efficiently. It employs specialized experts for different interaction ranges, capturing both short-range effects (like covalent bonding) and long-range phenomena (such as electrostatics). By sparsifying these experts, WANet achieves a rich representation of molecular interactions while maintaining computational efficiency, making it particularly well-suited for large-scale systems.

Third, WANet's architecture is designed to accurately capture the intrinsic properties of molecular systems, particularly for large molecules. The Hamiltonian matrix, which fully characterizes the quantum state and electron distribution, presents unique challenges in prediction due to complex electron correlation effects as the system size grows. To address this, WANet incorporates the MACE architecture's density trick, enabling efficient computation of many-body interactions without explicit calculation of all terms. This approach is crucial for maintaining accuracy as molecular size increases and electron correlation effects become more pronounced, ensuring WANet's scalability and precision in Hamiltonian prediction for large systems.

**Pair Construction Layer**    The objective of the pair construction layer is to extend the model's capacity to consider non-diagonal node pairs by introducing a tensor product filter. This filter modulates the projection of irreducible representations onto the space of node pair irreducible representations, denoted by $f_{ts}$. It is important to note that, in contrast to the Node Convolution Layer, which performs graph convolution on a radius graph or KNN graph, the pair construction

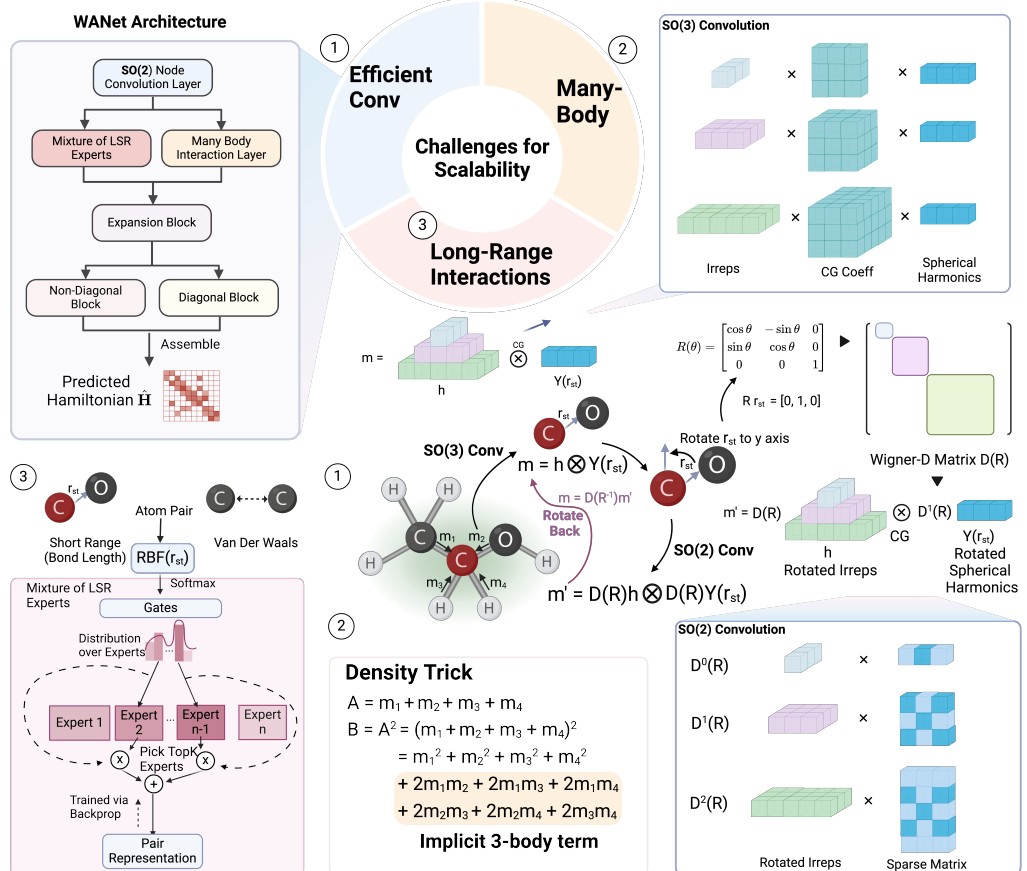

Figure 10: **Scalable architecture for predicting the Hamiltonian matrix ($\hat{\mathbf{H}}$)**. The framework addresses three key scalability challenges in quantum many-body systems: (1) efficient convolutions, (2) many-body interactions, and (3) long-range interactions. (1) Efficient convolutions are achieved through an SO(3)-equivariant convolution reduced to SO(2) for computational efficiency. (2) Many-body interactions are captured using a density trick, which implicitly includes three-body terms by combining pairwise interactions quadratically. (3) Long-range interactions are modeled with a mixture of Local-Structure Representation (LSR) experts, where a softmax-based gating mechanism allocates pairwise inputs to top-performing experts. Finally, the diagonal and non-diagonal components are assembled into the predicted Hamiltonian matrix, providing a scalable and accurate representation of the system dynamics.

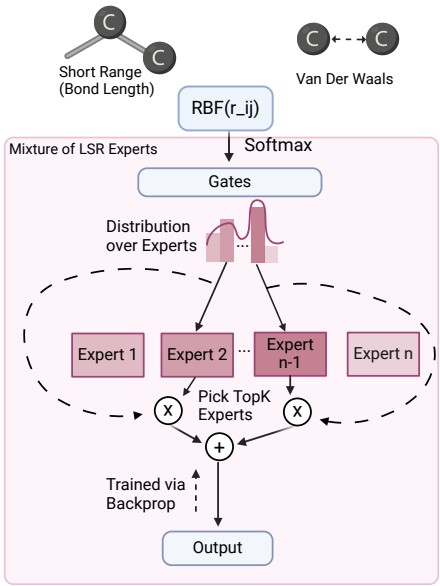

Figure 11: illustration of the Mixture of the Long-Short-Range Pair experts

layer considers all possible node interactions by operating on a complete graph. The mathematical formulation of this layer is given as follows:

$$f_{ts}^{\ell_o} = \sum_{l_i, l_j} W_{l_i, l_j, l_o} \left( x_s^{l_i} \otimes x_t^{l_j} \right)^{l_o}, \tag{15}$$

where $x_s^{l_i}$ and $x_t^{l_j}$ are the $l_i$-th and $l_j$-th irreducible representations of source node $s$ and target node $t$, respectively, and $W_{l_i, l_j, l_o}$ are the learned weights that couple these representations into the output representation $\ell_o$.

To improve the efficiency of the tensor product, we employed the channel-grouped tensor product, where the channels of the first and second tensors are tied to collectively construct the output channel path[5]. Additionally, to accommodate the symmetry inherent in the Hamiltonian matrix, we implement a symmetric structure in the pair representations. Specifically, the representations for node pairs $(s, t)$ and $(t, s)$ must be identical, reflecting the symmetrical nature of physical interactions. This is formulated as:

$$f_{ts}^{\ell_o}{}' = f_{st}^{\ell_o}{}' = \frac{1}{2}(f_{ts}^{\ell_o} + f_{st}^{\ell_o}),$$

where $f_{ts}^{\ell_o}$ and $f_{st}^{\ell_o}$ denote the initial, unsymmetrized tensor products for the node pairs $s$ and $t$ before the application of symmetry. The primed notations $f_{ts}^{\ell_o}{}'$ and $f_{st}^{\ell_o}{}'$ represent the final, symmetrized outputs.

**Graphical Illustration of MoE** Here we provide an additional illustration of the Mixture of the Long-Short-Range Pair experts described in the maintext, which is shown in Figure 11.

**Expansion Block** In the construction of final Hamiltonian blocks that encompass full orbital information using pair irreducible representations and many-body irreducible representation, a tensor expansion operation is employed alongside the filtering process. This expansion is defined by the following relation:

---

[5]This is also called "uuw" tensor product in `e3nn` implementation.

$$\left(\overline{\otimes}_{\ell_o} f^{\ell_o}\right)^{(m_i,m_j)}_{(\ell_i,\ell_j)} = \sum_{m_o=-\ell_o}^{\ell_o} C^{(\ell_o,m_o)}_{(\ell_i,m_i),(\ell_j,m_j)} f^{\ell_o}_{m_o}, \tag{16}$$

where $C$ denotes the Clebsch-Gordan coefficients, and $\overline{\otimes}$ symbolizes the tensor expansion which is the converse operation of tensor product. $u^{\ell_i} \otimes v^{\ell_j}$ can be expressed as a sum over tensor expansions:

$$u^{\ell_i} \otimes v^{\ell_j} = \sum_{\ell_3} W_{l_i,l_j,l_o} \overline{\otimes} f^{\ell_o}, \tag{17}$$

subject to the coupling constraints $|\ell_i - \ell_j| \le \ell_o \le \ell_i + \ell_j$.

**Loss Weighting.** We determined the loss weights through a principled ablation study on a validation set. While keeping $\lambda_1$ and $\lambda_2$ fixed at 1, we varied $\lambda_3$ from 0.5 to 3 to understand the impact of WALoss weighting. We found the method to be robust across a reasonable range of values, with $\lambda_1 = \lambda_2 = 1$, and $\lambda_3 = 2.5$ providing consistently strong performance. The table below summarizes the performance metrics for different values of $\lambda_1$, $\lambda_2$ and $\lambda_3$.

Table 11: Performance metrics for with different $\lambda_1$, $\lambda_2$, and $\lambda_3$ values.

| $\lambda_1$ | $\lambda_2$ | $\lambda_3$ | Hamiltonian MAE $\downarrow$ | $\epsilon_{\text{HOMO}}$ MAE $\downarrow$ | $\epsilon_{\text{LUMO}}$ MAE $\downarrow$ | $\epsilon_\Delta$ | $\epsilon_{\text{occ}}$ MAE $\downarrow$ | $\epsilon_{\text{orb}}$ MAE $\downarrow$ | C $\uparrow$ | System Energy MAE $\downarrow$ |
|---|---|---|---|---|---|---|---|---|---|---|
| 1 | 1 | 3 | 0.5258 | 0.8688 | 1.9116 | 2.0407 | 21.92 | 8.18 | 48.90% | 54.77 |
| 1 | 1 | 2.5 | 0.4744 | 0.7122 | 0.73 | 1.327 | 18.84 | 7.33 | 48.03% | 47.193 |
| 1 | 1 | 2 | 0.4918 | 0.7847 | 1.24981 | 1.46951 | 21.15 | 7.7 | 47.72% | 59.79 |
| 1 | 1 | 1.5 | 0.4586 | 1.50825 | 3.29805 | 3.1123 | 23.57 | 8.56 | 47.29% | 64.857 |
| 1 | 1 | 1 | 0.4807 | 1.1596 | 3.45767 | 3.2068 | 23.16 | 9.2 | 47.15% | 62.61719 |
| 1 | 1 | 0.5 | 0.4223 | 2.28836 | 6.83532 | 5.69888 | 28.96 | 11.78 | 45.30% | 72.9 |

# J ADDITIONAL THEORY

## J.1 PROOF OF THEOREM

**Theorem 2.** *Let* $\mathbf{H}, \hat{\mathbf{H}} \in \mathbb{R}^{n \times n}$ *be symmetric matrices, and let* $\mathbf{S} \in \mathbb{R}^{n \times n}$ *be a symmetric positive definite matrix. Consider the generalized eigenvalue problems:*

$$\mathbf{HC} = \mathbf{SC}\boldsymbol{\epsilon}, \quad \hat{\mathbf{H}}\,\hat{\mathbf{C}} = \mathbf{S}\,\hat{\mathbf{C}}\,\hat{\boldsymbol{\epsilon}},$$

*where* $\boldsymbol{\epsilon}$ *and* $\hat{\boldsymbol{\epsilon}}$ *are diagonal matrices of eigenvalues, and* $\mathbf{C}$ *and* $\hat{\mathbf{C}}$ *are the corresponding eigenvector matrices. Define* $\Delta\mathbf{H} = \hat{\mathbf{H}} - \mathbf{H}$, *and let* $\delta$ *be the minimum distance between the eigenvalue of interest and the rest of the spectrum of* $\mathbf{S}^{-1}\mathbf{H}$. *Then, the following bounds hold:*

1. ***Eigenvalue Differences:***

$$\left| \lambda_i(\hat{\mathbf{H}}, \mathbf{S}) - \lambda_i(\mathbf{H}, \mathbf{S}) \right| \leq \frac{\kappa(\mathbf{S})}{\|\mathbf{S}\|_2} \|\Delta\mathbf{H}\|_{\mathrm{F}} \leq \frac{\kappa(\mathbf{S})}{\|\mathbf{S}\|_2} \|\Delta\mathbf{H}\|_{1,1},$$

   *where* $\kappa(\mathbf{S}) = \|\mathbf{S}\|_2 \|\mathbf{S}^{-1}\|_2$ *is the condition number of* $\mathbf{S}$ *with respect to the spectral norm,* $\|\cdot\|_{\mathrm{F}}$ *denotes the Frobenius norm, and* $\|\cdot\|_{1,1}$ *denotes the element-wise* $l_1$ *norm.*

2. ***Eigenspace Angle:***

$$\sin\theta \leq \frac{\kappa(\mathbf{S})}{\|\mathbf{S}\|_2} \cdot \frac{\|\Delta\mathbf{H}\|_{\mathrm{F}}}{\delta} \leq \frac{\kappa(\mathbf{S})}{\|\mathbf{S}\|_2} \cdot \frac{\|\Delta\mathbf{H}\|_{1,1}}{\delta},$$

   *where* $\theta$ *is the angle between the eigenspaces corresponding to* $\lambda_i(\mathbf{H}, \mathbf{S})$ *and* $\lambda_i(\hat{\mathbf{H}}, \mathbf{S})$.

*Proof.* Consider the generalized eigenvalue problems for $\mathbf{H}$ and $\hat{\mathbf{H}}$ with respect to $\mathbf{S}$:

$$\mathbf{HC} = \mathbf{SC}\boldsymbol{\epsilon}, \quad \hat{\mathbf{H}}\,\hat{\mathbf{C}} = \mathbf{S}\,\hat{\mathbf{C}}\,\hat{\boldsymbol{\epsilon}}.$$

Since $\mathbf{S}$ is symmetric positive definite, it is invertible. We can transform the generalized eigenvalue problems into standard eigenvalue problems by multiplying both sides by $\mathbf{S}^{-1}$:

$$\mathbf{A} = \mathbf{S}^{-1}\mathbf{H}, \quad \hat{\mathbf{A}} = \mathbf{S}^{-1}\hat{\mathbf{H}} = \mathbf{A} + \mathbf{E},$$

where $\mathbf{E} = \mathbf{S}^{-1}\Delta\mathbf{H}$.

**Weyl's Perturbation Theorem**

**Theorem 3** (Weyl's Perturbation Theorem). *Let* $\mathbf{A}, \hat{\mathbf{A}} \in \mathbb{R}^{n \times n}$ *be symmetric matrices with eigenvalues* $\lambda_1 \leq \lambda_2 \leq \cdots \leq \lambda_n$ *and* $\hat{\lambda}_1 \leq \hat{\lambda}_2 \leq \cdots \leq \hat{\lambda}_n$, *respectively. Then, for all* $i = 1, \ldots, n$,

$$\left| \hat{\lambda}_i - \lambda_i \right| \leq \|\hat{\mathbf{A}} - \mathbf{A}\|_2.$$

Applying Weyl's Theorem to $\mathbf{A}$ and $\hat{\mathbf{A}}$, we have:

$$\left| \lambda_i(\hat{\mathbf{A}}) - \lambda_i(\mathbf{A}) \right| \leq \|\mathbf{E}\|_2 = \|\mathbf{S}^{-1}\Delta\mathbf{H}\|_2.$$

**Bounding the Eigenvalue Differences**

Using the sub-multiplicative property of the spectral norm:

$$\|\mathbf{S}^{-1}\Delta\mathbf{H}\|_2 \leq \|\mathbf{S}^{-1}\|_2 \|\Delta\mathbf{H}\|_2.$$

Since the spectral norm is bounded by the Frobenius norm:

$$\|\Delta\mathbf{H}\|_2 \leq \|\Delta\mathbf{H}\|_{\mathrm{F}}.$$

Combining these inequalities:

$$\|\mathbf{S}^{-1}\Delta\mathbf{H}\|_2 \leq \|\mathbf{S}^{-1}\|_2 \|\Delta\mathbf{H}\|_{\mathrm{F}}.$$

The condition number $\kappa(\mathbf{S})$ is defined as:

$$\kappa(\mathbf{S}) = \|\mathbf{S}\|_2 \|\mathbf{S}^{-1}\|_2 \implies \|\mathbf{S}^{-1}\|_2 = \frac{\kappa(\mathbf{S})}{\|\mathbf{S}\|_2}.$$

Substituting back:

$$\|\mathbf{S}^{-1}\Delta\mathbf{H}\|_2 \leq \frac{\kappa(\mathbf{S})}{\|\mathbf{S}\|_2}\|\Delta\mathbf{H}\|_{\mathrm{F}}.$$

Therefore:

$$\left|\lambda_i(\hat{\mathbf{A}}) - \lambda_i(\mathbf{A})\right| \leq \frac{\kappa(\mathbf{S})}{\|\mathbf{S}\|_2}\|\Delta\mathbf{H}\|_{\mathrm{F}}.$$

Since the eigenvalues of $\mathbf{A}$ and $\hat{\mathbf{A}}$ correspond to the generalized eigenvalues of $(\mathbf{H}, \mathbf{S})$ and $(\hat{\mathbf{H}}, \mathbf{S})$, respectively, we have:

$$\left|\lambda_i(\hat{\mathbf{H}}, \mathbf{S}) - \lambda_i(\mathbf{H}, \mathbf{S})\right| \leq \frac{\kappa(\mathbf{S})}{\|\mathbf{S}\|_2}\|\Delta\mathbf{H}\|_{\mathrm{F}}.$$

Since $\|\Delta\mathbf{H}\|_{\mathrm{F}} \leq \|\Delta\mathbf{H}\|_{1,1}$, we can further bound:

$$\left|\lambda_i(\hat{\mathbf{H}}, \mathbf{S}) - \lambda_i(\mathbf{H}, \mathbf{S})\right| \leq \frac{\kappa(\mathbf{S})}{\|\mathbf{S}\|_2}\|\Delta\mathbf{H}\|_{1,1}.$$

### Davis-Kahan $\sin\theta$ Theorem

**Theorem 4** (Davis-Kahan $\sin\theta$ Theorem). *Let $\mathbf{A}, \hat{\mathbf{A}} \in \mathbb{R}^{n \times n}$ be symmetric matrices, and let $\mathcal{U}$ and $\hat{\mathcal{U}}$ be the invariant subspaces of $\mathbf{A}$ and $\hat{\mathbf{A}}$ corresponding to eigenvalues in intervals $\mathcal{I}$ and $\hat{\mathcal{I}}$, respectively. If $\delta = \mathrm{dist}(\mathcal{I}, \hat{\mathcal{I}}^c) > 0$, then*

$$\|\sin\Theta\|_2 \leq \frac{\|\hat{\mathbf{A}} - \mathbf{A}\|_2}{\delta},$$

*where $\Theta$ is the matrix of principal angles between $\mathcal{U}$ and $\hat{\mathcal{U}}$.*

### Bounding the Eigenspace Angle

Applying the Davis-Kahan Theorem to our case:

$$\sin\theta \leq \frac{\|\mathbf{E}\|_2}{\delta} = \frac{\|\mathbf{S}^{-1}\Delta\mathbf{H}\|_2}{\delta}.$$

Using the previously established bound:

$$\|\mathbf{S}^{-1}\Delta\mathbf{H}\|_2 \leq \frac{\kappa(\mathbf{S})}{\|\mathbf{S}\|_2}\|\Delta\mathbf{H}\|_{\mathrm{F}},$$

we obtain:

$$\sin\theta \leq \frac{\kappa(\mathbf{S})}{\|\mathbf{S}\|_2} \cdot \frac{\|\Delta\mathbf{H}\|_{\mathrm{F}}}{\delta}.$$

Similarly, since $\|\Delta\mathbf{H}\|_{\mathrm{F}} \leq \|\Delta\mathbf{H}\|_{1,1}$:

$$\sin\theta \leq \frac{\kappa(\mathbf{S})}{\|\mathbf{S}\|_2} \cdot \frac{\|\Delta\mathbf{H}\|_{1,1}}{\delta}.$$

This completes the proof. $\qquad\qquad\square$

## J.2 PERTURBATION ANALYSIS AND GROWTH OF EIGENVALUES

**Theorem 5** (Generalized Bai-Yin's law). *Let $A$ be an $n \times n$ random matrix with independent entries having mean $\mu$ and variance $\sigma^2$. Then, with high probability, the spectral norm of $A$ is bounded by:*

$$\|A\|_2 \leq |\mu|n + 2\sigma\sqrt{n}.$$

*Proof.* Consider the matrix $A$ decomposed into its mean and fluctuation components:

$$A = \mu J + B,$$

where $J$ is the matrix of all ones and $B$ is a random matrix with zero-mean entries and variance $\sigma^2$.

First, we bound the spectral norm of the mean component $\mu J$. The matrix $J$ is a rank-1 matrix with all entries equal to 1. Its largest singular value is $n$, so:

$$\|\mu J\|_2 = |\mu| \cdot n.$$

Next, we bound the spectral norm of the fluctuation component $B$. Since $B$ has i.i.d. entries with zero mean and variance $\sigma^2$, using Bai-Yin's law, we get:

$$\|B\|_2 \leq 2\sigma\sqrt{n},$$

with high probability.

Combining these bounds using the triangle inequality, we have:

$$\|A\|_2 \leq \|\mu J\|_2 + \|B\|_2 \leq |\mu|n + 2\sigma\sqrt{n}.$$

Thus, with high probability, the spectral norm of $A$ is bounded by:

$$\|A\|_2 \leq |\mu|n + 2\sigma\sqrt{n}.$$

$\square$

**Theorem 6** (Perturbation Sensitivity Scaling). *Consider the generalized eigenvalue problem $\mathbf{H}\mathbf{C} = \mathbf{S}\mathbf{C}\boldsymbol{\epsilon}$, where $\mathbf{H} \in \mathbb{R}^{B \times B}$ is a symmetric Hamiltonian matrix, $\mathbf{S} \in \mathbb{R}^{B \times B}$ is a positive definite overlap matrix, $\mathbf{C} \in \mathbb{R}^{B \times k}$ is the eigenvector matrix, and $\boldsymbol{\epsilon} \in \mathbb{R}^{k \times k}$ is a diagonal matrix of eigenvalues $\lambda_i(\mathbf{H}, \mathbf{S})$. Let $\hat{\mathbf{H}} = \mathbf{H} + \Delta\mathbf{H}$ be the perturbed Hamiltonian, with $\Delta\mathbf{H} \in \mathbb{R}^{B \times B}$ a perturbation matrix whose entries have mean $\mu$ and variance $\sigma^2$. Suppose the smallest eigenvalue of $\mathbf{S}$ satisfies:*

$$\lambda_{\min}(\mathbf{S}) = c + \frac{A}{1 + \left(\frac{B}{N_0}\right)^\alpha},$$

*where $c > 0$, $A > 0$, $\alpha > 0$, and $N_0 > 0$ are fixed constants. Then, for each eigenvalue, there exists a constant $K > 0$ such that:*

$$\left| \lambda_i(\hat{\mathbf{H}}, \mathbf{S}) - \lambda_i(\mathbf{H}, \mathbf{S}) \right| \leq K \left( \sigma B^{1/2} + B|\mu| \right),$$

*for all sufficiently large $B$.*

*Proof.* Consider the unperturbed generalized eigenvalue problem $\mathbf{H}\mathbf{c}_i = \lambda_i \mathbf{S}\mathbf{c}_i$, where $\mathbf{H}$ and $\mathbf{S}$ are symmetric, $\mathbf{S}$ is positive definite, and $\mathbf{c}_i^T \mathbf{S}\mathbf{c}_i = 1$. The perturbed system is $\hat{\mathbf{H}}\hat{\mathbf{c}}_i = \hat{\lambda}_i \mathbf{S}\hat{\mathbf{c}}_i$, with $\hat{\mathbf{H}} = \mathbf{H} + \Delta\mathbf{H}$, $\hat{\mathbf{c}}_i = \mathbf{c}_i + \delta\mathbf{c}_i$, and $\hat{\lambda}_i = \lambda_i + \delta\lambda_i$. Substituting and expanding yields:

$$(\mathbf{H} + \Delta\mathbf{H})(\mathbf{c}_i + \delta\mathbf{c}_i) = (\lambda_i + \delta\lambda_i)\mathbf{S}(\mathbf{c}_i + \delta\mathbf{c}_i).$$

Subtracting the unperturbed equation and projecting onto $\mathbf{c}_i^T$, noting $\mathbf{c}_i^T(\mathbf{H} - \lambda_i\mathbf{S}) = 0$, we obtain:

$$\mathbf{c}_i^T \Delta\mathbf{H}\mathbf{c}_i + \mathbf{c}_i^T \Delta\mathbf{H}\delta\mathbf{c}_i = \delta\lambda_i + \delta\lambda_i\mathbf{c}_i^T \mathbf{S}\delta\mathbf{c}_i.$$

Solving for the perturbation:

$$\delta\lambda_i = \frac{\mathbf{c}_i^T \Delta\mathbf{H}\mathbf{c}_i + \mathbf{c}_i^T \Delta\mathbf{H}\delta\mathbf{c}_i}{1 + \mathbf{c}_i^T \mathbf{S}\delta\mathbf{c}_i}.$$

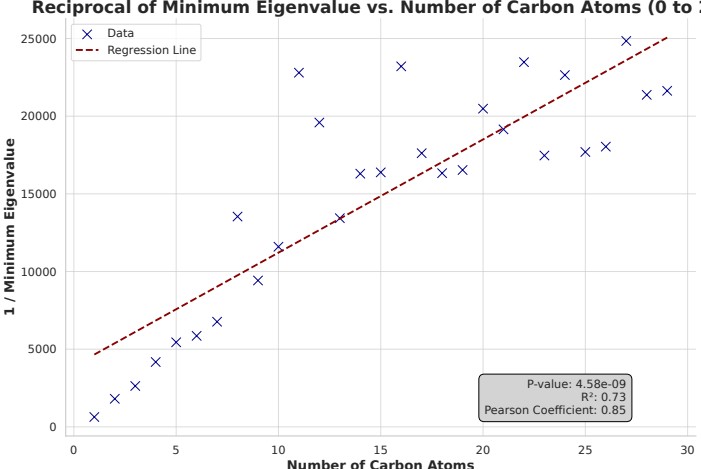

Figure 12: The scaling of the smallest eigenvalue of the overlap matrix in satured hydrocarbons.

For small $\Delta\mathbf{H}$, $\delta\mathbf{c}_i$ is $O(\|\Delta\mathbf{H}\|_2)$, so $\mathbf{c}_i^T\Delta\mathbf{H}\delta\mathbf{c}_i$ and $\mathbf{c}_i^T\mathbf{S}\delta\mathbf{c}_i$ are higher-order terms. Here, we only consider the first-order term arises from evaluating the perturbation with the unperturbed eigenvector. Higher-order terms account for eigenvector shifts. We claim that analyzing the higher-order terms gives rise to unnecessary complication and does not alter the nature of catastrophic scaling. Specifically:

$$\delta\lambda_i \approx \mathbf{c}_i^T\Delta\mathbf{H}\mathbf{c}_i.$$

Bounding this term:

$$|\delta\lambda_i| \le |\mathbf{c}_i^T\Delta\mathbf{H}\mathbf{c}_i| \le \|\mathbf{c}_i\|_2^2\|\Delta\mathbf{H}\|_2.$$

Since $\mathbf{c}_i^T\mathbf{S}\mathbf{c}_i = 1$ and $\mathbf{S} \succeq \lambda_{\min}(\mathbf{S})\mathbf{I}$, we have:

$$\|\mathbf{c}_i\|_2^2 \le \frac{1}{\lambda_{\min}(\mathbf{S})} = \frac{1}{c + \frac{A}{1+\left(\frac{B}{N_0}\right)^\alpha}} = \frac{1+\left(\frac{B}{N_0}\right)^\alpha}{c\left[1+\left(\frac{B}{N_0}\right)^\alpha\right] + A}.$$

For $\Delta\mathbf{H}$ as a random matrix, apply a spectral norm bound (by Bai-Yin law):

$$\|\Delta\mathbf{H}\|_2 \le C_1(\sigma B^{1/2} + B|\mu|),$$

with constant $C_1$, holding with high probability. Combining:

$$|\delta\lambda_i| \le \frac{1+\left(\frac{B}{N_0}\right)^\alpha}{c\left[1+\left(\frac{B}{N_0}\right)^\alpha\right] + A}C_1(\sigma B^{1/2} + B|\mu|).$$

Thus, the leading-order bound is:

$$\left|\lambda_i(\hat{\mathbf{H}}, \mathbf{S}) - \lambda_i(\mathbf{H}, \mathbf{S})\right| = O\left(\frac{1+\left(\frac{B}{N_0}\right)^\alpha}{c\left[1+\left(\frac{B}{N_0}\right)^\alpha\right] + A}\left(\sigma B^{1/2} + B|\mu|\right)\right) = O\left(\sigma B^{1/2} + B|\mu|\right).$$

$\square$

**Remark 1.** *If higher-order terms like $\mathbf{c}_i^T\Delta\mathbf{H}\delta\mathbf{c}_i$ and the denominator correction are included, the perturbation includes an additional term of order $O\left(\left(\frac{1+\left(\frac{B}{N_0}\right)^\alpha}{c\left[1+\left(\frac{B}{N_0}\right)^\alpha\right] + A}\right)^2 (\sigma B^{1/2} + B|\mu|)^2\right)$, reflecting the quadratic contribution of the perturbation strength.*

### J.3 Scaling Law of the Smallest Eigenvalue of Overlap Matrix

Figure 12 shows the relationship between the number of carbon atoms in saturated hydrocarbons and the reciprocal of the smallest eigenvalue of the overlap matrix. The x-axis represents the number of carbon atoms, ranging from 0 to 29. The y-axis represents the reciprocal of the smallest eigenvalue, ranging up to 25,000. Specifically, the regression analysis yields a p-value of $4.58 \times 10^{-9}$, an $R^2$ value of 0.73, and a Pearson coefficient of 0.85, suggesting a strong linear relationship.

We also compare another scaling law suggested by Høyvik (2020). As shown in Figure 13, we investigated the relationship between $\frac{1}{\lambda_{\min}}$ and the number of basis functions for various molecules. Our results indicate a clear power-law scaling, which can be expressed as $y = a \cdot x^b$. The fitted parameters for each molecule are as follows: Eicosane (C20H42): $y = 0.000107 \cdot x^{2.43}$, Graphene (C24): $y = 0.00158 \cdot x^{2.83}$, and Diamond (C10): $y = 0.0178 \cdot x^{4.16}$. This power-law behavior emphasizes the significant impact of the number of basis functions on $\frac{1}{\lambda_{\min}}$, with different scaling exponents for each molecule.

#### J.3.1 $k$-Nearest-Neighbor Overlaps

Here, we analyzed a system where each basis function overlaps with its $k$-nearest neighbors. The overlap matrix $S \succ 0$ in this case is a symmetric banded matrix with bandwidth $2k + 1$:

$$S_{ij} = \begin{cases} 1, & \text{if } i = j, \\ s_{|i-j|}, & \text{if } 0 < |i - j| \le k, \\ 0, & \text{if } |i - j| > k. \end{cases}$$

To derive a scaling equation for the smallest eigenvalue, consider the case where $k = 1$, which corresponds to the nearest-neighbor model. Other cases can be derived similarly using plane waves and discrete Fourier Transform. The nearest-neighbor model holds physical significance in the context of tight-binding models. For example, in chain-like polyenes, the Huckel model is based on a nearest-neighbor analysis, providing a meaningful framework for discussing this case. The overlap matrix $S$ is given as:

$$S = \begin{bmatrix} 1 & s & 0 & \cdots & 0 \\ s & 1 & s & \ddots & \vdots \\ 0 & s & 1 & \ddots & 0 \\ \vdots & \ddots & \ddots & \ddots & s \\ 0 & \cdots & 0 & s & 1 \end{bmatrix}$$

The eigenvalues $\lambda_n$ of this tridiagonal matrix can be derived using the known solutions for such matrices:

$$\lambda_n = 1 + 2s \cos\left(\frac{n\pi}{B+1}\right), \quad n = 1, 2, \ldots, B.$$

The smallest eigenvalue corresponds to $n = B$:

$$\lambda_{\min} = 1 + 2s \cos\left(\frac{B\pi}{B+1}\right).$$

Using the trigonometric identity $\cos(\pi - \theta) = -\cos(\theta)$ and approximating for large $B$:

$$\cos\left(\frac{B\pi}{B+1}\right) = -\cos\left(\frac{\pi}{B+1}\right) \approx -\left(1 - \frac{1}{2}\left(\frac{\pi}{B+1}\right)^2\right).$$

Substituting back:

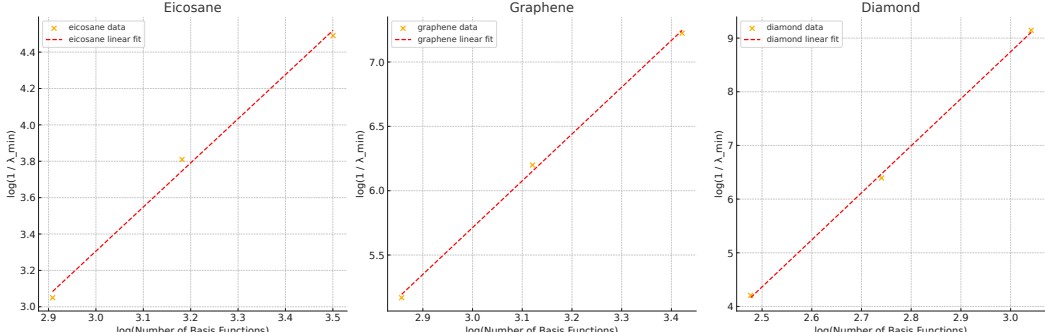

Figure 13: The scaling of the smallest eigenvalue of the overlap matrix in three systems. Data adapted from Høyvik (2020).

$$\lambda_{\min} \approx 1 - 2s + s \left(\frac{\pi}{B}\right)^2.$$

Since $S \succ 0$, this yields $1 - 2s \geq 0$. Here, we investigated the smallest eigenvalue $\lambda_{\min}$ as $B \to \infty$, identifying two distinct cases:

1. **Saturating Case:** When $s < 0.5$ the smallest eigenvalue saturates at a finite value $1 - 2s$ as $B \to \infty$.

2. **Non-Saturating Case:** For $s = 0.5$, $\lambda_{\min}$ decreases indefinitely with increasing $B$.

## K  TRAINING

**Predicting the Gap from Initial Guess**  In contrast to prior research, our study tackles the significant challenges posed by the scale of the PubChemQC t-zvp dataset. It is very challenging for the model to develop a good numeric starting point. To address this, we have shifted our focus towards a more tractable objective: making predictions based on an initial guess, which is easy to obtain. This adjustment is formalized in our objective function as follows:

$$\theta^\star = \arg\min_{\theta} \frac{1}{|\mathcal{D}|} \sum_{(\mathcal{M}, \mathbf{H}_{\mathcal{M}}^\star) \in \mathcal{D}} \text{dist}\left(\hat{\mathbf{H}}_\theta(\mathcal{M}), \mathbf{H}_{\mathcal{M}}^\star - \mathbf{H}_{\mathcal{M}}^{(0)}\right), \quad (18)$$

where $|\mathcal{D}|$ denotes the cardinality of dataset $\mathcal{D}$, and $\text{dist}(\cdot, \cdot)$ is a predefined distance metric, $\mathbf{H}_{\mathcal{M}}^{(0)}$ is the initial guess of the Hamiltonian. We conducted an ablation study, showing that although predictions based on the initial guess significantly improve performance in Hamiltonian prediction, they struggle with predicting physical properties.

**Total Loss Function**  The total loss function used to train our model is a combination of the orbital alignment loss and the mean squared error (MSE) loss between the predicted and true Hamiltonian matrices:

$$\mathcal{L}_{total} = \lambda_1 \left(\frac{1}{n} \sum_{i=1}^{n} \|\hat{\mathbf{H}}_i + \mathbf{H}_i^{(0)} - \mathbf{H}_i^\star\|_{\text{F}}^2\right) + \lambda_2 \left(\frac{1}{n} \sum_{i=1}^{n} \|\hat{\mathbf{H}}_i + \mathbf{H}_i^{(0)} - \mathbf{H}_i^\star\|_{\text{F}}^1\right) + \lambda_3 \mathcal{L}_{\text{align}} \quad (19)$$

where $\lambda_1, \lambda_2, \lambda_3$ are hyperparameters that control the relative importance of each loss component.

**Experimental Setting**  The experimental settings are presented in Table 12. For the PubChemQH dataset, we used an 80/10/10 train/validation/test split, resulting in 40,257 training molecules, 5,032 validation molecules, and 5,032 test molecules. We trained all models of maximum 300,000 steps with a batch size of 8, using early stopping with a patience of 1,000 steps. WANet converges at 278,391 steps, QHNet at 258,267 steps, and PhisNet at 123,170 steps. All models used the Adam

optimizer with a learning rate of 0.001 for PubChemQH, along with a polynomial learning rate scheduler with 1,000 warmup steps. We used gradient clipping of 1, and a radius cutoff of 5 Å. For QHNet and PhisNet, we referred to the official implementation for these two models.

Table 12: Hyperparameter settings for the experimental study using the PubChemQH and QH9 datasets.

| Hyperparameter | PubChemQH (WANet) | QH9 (WANet) | PubChemQH (QHNet) | QH9 (QHNet) | PubChemQH (PhiSNet) | Description |
|---|---|---|---|---|---|---|
| Learning Rate | 0.001 | 5e-4 | 0.001 | 5e-4 | 0.001 | |
| Batch Size | 8 | 32 | 8 | 32 | 8 | |
| Scheduler | Polynomial | Polynomial | Polynomial | Polynomial | Polynomial | |
| LR Warmup Steps | 1,000 | 1,000 | 1,000 | 1,000 | 1,000 | Number of steps to linearly increase the learning rate |
| Max Steps | 300,000 | 300,000 | 300,000 | 300,000 | 300,000 | Maximum number of training steps |
| Model Order | 6 | 4 | 6 | 4 | 6 | Maximum degree of the spherical harmonics |
| Embedding Dimension | 128 | 128 | 128 | 128 | 128 | Dimension of the node embedding |
| Bottle Hidden Size | N/A | N/A | 32 | 32 | N/A | Size of the hidden layer in the bottleneck |
| Number of GNN Layers | 5 | 5 | 5 | 5 | 5 | Number of graph neural network layers |
| Max Radius | 5 | 15 | 5 | 15 | 5 | Maximum distance between nieghboring atoms |
| Sphere Channels | 128 | 128 | 128 | 128 | 128 | Number of channels in the spherical harmonics |
| Edge Channels | 128 | 128 | 32 | 32 | 32 | Number of channels for edge features |
| Drop Path Rate | 0.1 | 0.1 | N/A | N/A | N/A | Probability of dropping a path in the network |
| Projection Drop | 0.0 | 0.0 | N/A | N/A | N/A | Dropout rate for the projection layer |

The training details for the HOMO, LUMO, and GAP predictions are presented in Table 13. For a fair comparison, all regression models used identical dataset splits (40,257 training molecules, 5,032 validation molecules, and 5,032 test molecules). We used batch size of 32 and consistent optimizer settings across all models: Adam optimizer with a learning rate of 0.001 and a polynomial learning rate scheduler with 1,000 warmup steps. WANet followed the same training setup described in Table 12. The complete regression model hyperparameters are detailed in Table 13. For Equiformer V2, Uni-Mol+, and Uni-Mol2 models, we used their original implementations.

Table 13: The training details for the HOMO, LUMO, and GAP predictions.

| Hyperparameter | Uni-Mol+ (Lu et al., 2023) | Uni-Mol2 (Xiaohong et al., 2024) | Equiformer v2 | Description |
|---|---|---|---|---|
| Learning Rate | 0.001 | 0.001 | 0.001 | |
| Batch Size | 32 | 32 | 32 | |
| Scheduler | Polynomial | Polynomial | Polynomial | |
| LR Warmup Steps | 1,000 | 1,000 | 1,000 | Number of steps to linearly increase the learning rate |
| Max Steps | 300,000 | 300,000 | 300,000 | Maximum number of training steps |
| Embedding Dimension | 768 | 768 | 128 | Dimension of the input embedding |
| FFN Embedding Dim | 3072 | 3072 | 512 | Embedding dimension for the feed-forward network |
| Number of Encoder Layers | 6 | 6 | 5 | Number of layers in the transformer encoder |
| Drop Path Rate | 0.0 | 0.0 | 0.1 | Probability of dropping a path in the network |
| Activation Function | GELU | GELU | SiLU | Activation function used in the model |

## L    DISCUSSION

### L.1    DISCUSSION ON SYSTEM ENERGY ERROR

The mean absolute error (MAE) of  47 kcal/mol for system energy prediction is notably above the threshold of chemical accuracy (typically  1 kcal/mol). This highlights the inherent challenges of accurately predicting system energies for large molecular systems.  To our knowledge, this work is *the first* to evaluate Hamiltonian predictions on system energy, which presents a non-trivial implementation challenge. Previous studies have typically focused on metrics such as cosine similarity of eigenvectors or MAE of occupied energy levels. By directly assessing system energy, our approach provides a more comprehensive and practical evaluation of Hamiltonian accuracy, which is critical for large-scale molecular simulations.

Our model demonstrates a significant improvement over baseline models in predicting system energies for these large systems, indicating enhanced prediction accuracy.  The primary goal of this work is to showcase the scalability and applicability of our approach to large molecular systems, where achieving absolute precision is inherently difficult due to their complexity. Despite this, our results represent a meaningful step toward more accurate and efficient predictions in this challenging domain. The advances we have made underscore the potential of our approach to be further refined and applied across a range of complex molecular systems.

Moreover, other critical molecular properties such as the highest occupied molecular orbital (HOMO), the lowest unoccupied molecular orbital (LUMO), and dipole moments are predicted with reasonable accuracy. These properties, often vital in quantum chemistry workflows, demonstrate the practical utility of our model beyond system energy prediction.  The ability to predict multiple important molecular properties with competitive error rates reinforces the broader applicability of our model to a variety of quantum chemistry and materials science applications

### L.2    ADDITIONAL CONFORMATIONAL ENERGY ANALYSIS

In quantum chemistry applications, relative energy differences between conformational states are typically more relevant than absolute energies. To better evaluate our method's practical utility, we conducted an additional conformational energy analysis. For each molecule in our test set, we:

1. Sampled 100 conformations.

2. Computed $\Delta E$ between each conformation and a reference structure using both DFT and WANet.

3. Calculated the MAE of these energy differences.

For sampling, we employed Gaussian perturbation to introduce noise in the atomic positions. Starting from each molecule's initial conformation from the test sets, we generated 100 perturbed geometries by adding random Gaussian noise ($\sigma = 0.1$Å) to the atomic coordinates.  The potential energy values for these perturbed conformations were computed using the B3LYP/def2-TZVP level of theory. The resulting energy standard deviation between sampled conformations ranged from 10 to around 100 kcal/mol (with average std =49.84kcal/mol), depending on the molecule's flexibility. For baseline models (included in the Table 14), the PhisNet model withWALoss achieved 9.90 kcal/mol MAE, the QHNet model with WALoss achieved 10.98 kcal/mol MAE, and the WANet model without WALoss achieved 48.92 kcal/mol MAE. QHNet without WALoss yielded MAEs of 50.56 kcal/mol. Interestingly, we observed that the model without WALoss produce meaningless prediction, demonstrating the effectiveness of WALoss. It worth noting that while more complex sampling strategy could be used, those methods are more computationally expensive as they require DFT calculations.

### L.3    DISCUSSION ON SCF ACCELERATION RATIO

We report an 18% reduction in SCF cycles, which is a notable improvement, especially considering that our work targets significantly larger and more complex molecular systems compared to previous studies such as QH9 and QHNet, where SCF reductions of 18–35% were achieved.  Given the

Table 14: Relative energy differences between conformational states.

| Model | MAE of Energy Differences (kcal/mol) |
|---|---|
| WANet w/WALoss | 1.12 |
| PhisNet w/WALoss | 9.90 |
| QHNet w/WALoss | 10.98 |
| WANet | 48.92 |
| QHNet | 50.56 |

increased complexity and size of our systems, this 18% reduction represents a considerable improvement, reflecting the effectiveness of our model in accelerating convergence for more challenging molecular simulations. This result highlights the scalability of our approach in handling large-scale quantum chemical calculations efficiently.

