# OpenReview forum: "Enhancing the Scalability and Applicability of Kohn-Sham Hamiltonians for Molecular Systems"
_ICLR.cc/2025/Conference — ICLR 2025 Spotlight_

### Official Review · Reviewer_LpfH · 2024-10-26

**Soundness:** 3
**Presentation:** 3
**Contribution:** 3
**Rating:** 8
**Confidence:** 3

**Summary:**

This work encompasses 4 contributions. First, it introduces a new, larger dataset for predicting Kohn-Sham Density Functional Theory (KS-DFT) Hamiltonians, surpassing previous datasets in size. Second, it finds that training with Mean Absolute Error (MAE) on the Hamiltonian is inadequate for larger datasets, often producing results inferior to traditional minao guesses. Third, the authors propose a wave function alignment loss as an alternative to MAE, focusing on optimizing the eigenspace rather than the Hamiltonian itself. Finally, they present a new model that achieves state-of-the-art performance.

**Strengths:**

* The paper is very well written and easy to understand.
* The empirical analysis of SAD is valuable and motivates this work well.
* The WALoss appears like a natural choice and comes at essentially no additional cost.
* Both the WALoss and WANet significantly improve upon previous works.

**Weaknesses:**

* The asymptotic scaling in Collary 1 for the lowest eigenvalue seems unfit to describe Gaussian-type orbitals. Due to their fast decay, the overlap is spatially limited. After a certain system size, more atoms are unlikely to affect the lowest eigenvalue.
* l.62/63, "complex transformation" sounds overly complicated given the symmetric nature of the Hamiltonian.
* Unfortunately, the real-world DFT speed-ups remain limited.

**Questions:**

* While the comparison to a regression model is laudable, the comparison to equiformer has many confounding variables. Could the authors compare to their WANet but with a regression head?

---

> ### Author Response · Authors · 2024-11-20
>
> Thank you for your constructive comments and suggestions, they are exceedingly helpful for us to improve our paper. Our point-to-point responses to your comments are given below:
>
> > The asymptotic scaling in Collary 1 for the lowest eigenvalue seems unfit to describe Gaussian-type orbitals (MO). Due to their fast decay, the overlap is spatially limited. After a certain system size, more atoms are unlikely to affect the lowest eigenvalue.
> >
>
> Our scaling analysis is grounded in both previous studies and empirical observations from various datasets. Previous studies [1], as illustrated in Figure 13, have established foundations for this scaling behavior. Our experimental results from carbon atom systems (Figure 12), as well as observations from QH9 and PubChemQH datasets (Figure 6, and anonymous link: https://ibb.co/9tyNgZm), corroborate this scaling behavior.  It's important to note, however, that we have not explored extremely large systems or highly specialized cases, which may exhibit different scaling behaviors.
>
> [1]: Ida-Marie Høyvik. The spectrum of the atomic orbital overlap matrix and the locality of the virtual
> electronic density matrix. Molecular Physics, 2020.
>
> > l.62/63, "complex transformation" sounds overly complicated given the symmetric nature of the Hamiltonian.
> >
>
> We acknowledge that the term "complex transformation" may imply unnecessary intricacy, considering the symmetric nature of the Hamiltonian. To more accurately reflect the physical operation while maintaining technical precision, we have updated this term to "**unitary transformation**.”
>
> > While the comparison to a regression model is laudable, the comparison to equiformer has many confounding variables. Could the authors compare to their WANet but with a regression head?
> >
>
> Following your suggestion, we implemented a regression head variant of WANet for direct comparison. The results (WANet w/regression: 6.892 MAE, WANet w/WALoss: 0.7122 MAE, Equiformer V2: 6.955 MAE) clearly demonstrate that our improvements stem from the WALoss formulation and architectural design rather than confounding variables. This controlled comparison   provides strong evidence for the effectiveness of our approach.
>
> |  | HOMO MAE | LUMO MAE | GAP MAE |
> | --- | --- | --- | --- |
> | WANet Regression | 6.892 | 6.852 |  3.124 |
> | Equiformer V2 | 6.955 | 6.562 | 3.222 |
> | WANet | 0.7122 | 0.730 | 1.327 |
>
> >
> > Unfortunately, the real-world DFT speed-ups remain limited.
> >
>
> We thank the reviewer for raising this important point. We report an 18% reduction in SCF cycles, which is a notable improvement, especially considering that our work targets significantly larger and more complex molecular systems compared to previous studies such as QH9 and QHNet, where SCF reductions of 12–33% were achieved compared to minao inital guess [1]. Given the increased complexity and size of our systems, this 18% reduction represents a considerable improvement, reflecting the effectiveness of our model in accelerating convergence for more challenging molecular simulations. This result highlights the scalability of our approach in handling large-scale quantum chemical calculations efficiently. Moreover, Hamiltonian prediction enables multiple downstream applications beyond SCF acceleration. WANet demonstrates its versatility through state-of-the-art performance in predicting various electronic properties such as HOMO-LUMO gaps and dipole moments.
>
> [1]: Yu H, Liu M, Luo Y, et al. Qh9: A quantum hamiltonian prediction benchmark for qm9 molecules[J]. Advances in Neural Information Processing Systems, 2024, 36.

---

> > ### Comment · Reviewer_LpfH · 2024-11-23
> >
> > I thank the reviewer for their feedback. My only concern lies in the asymptotic statements and the following collary. By definition, only the scaling for very large systems matters for asymptotics. If I'd create a system with infinitely separated atoms, the eigenvalues of the total system's overlap matrix would coincide with those of the individual systems. I remain with my judgement that this assumption may be valid in this intermediate regime, it seems implausible as an asymptotic scaling.

---

> ### Author Response · Authors · 2024-11-30
>
> Dear Reviewer,
>
> Thank you for your insightful comments regarding the asymptotic behavior of the smallest eigenvalue in systems described by an overlap matrix. We agree that "only the scaling for very large systems matters for asymptotics," and appreciate your prompt to discuss the scaling behavior of our analysis.
>
> You are correct that for infinitely separated atoms or molecules, the eigenvalues of the total system's overlap matrix match those of the individual, non-interacting subsystems. In this case, the smallest eigenvalue reaches a finite value due to the lack of overlap between distant basis functions. However, we respectfully contend that this example lies outside the relevant asymptotic regime. Asymptotic analysis, by definition, examines the scaling behavior of a system as its size grows while preserving the structure and interactions that define it. Infinitely separated atoms reduce the system to isolated subsystems, effectively forming a different physical system. Consequently, this scenario offers limited insights into the asymptotic scaling of the original system, where finite overlaps persist. For a meaningful analysis, we focus on a more relevant system:
>
> ### **$k$-Nearest-Neighbor Overlaps**
>
> Here, we analyzed a system where each basis function overlaps with its $k$-nearest neighbors. The overlap matrix $S \succ 0$ in this case is a symmetric banded matrix with bandwidth $2k + 1$:
> $$
> S_{ij} =
> \begin{cases}
> 1, & \text{if } i = j, \\\\
> s_{|i -j|} & \text{if } 0 < |i - j| \leq k, \\\\
> 0, & \text{if } |i - j| > k,
> \end{cases}
> $$
> To derive a scaling equation for the smallest eigenvalue, consider the case where $k=1$, which corresponds to the nearest-neighbor model. Other cases can be derived similarly using plane waves and discrete Fourier Transform. The nearest-neighbor model holds physical significance in the context of tight-binding models. For example, in chain-like polyenes, the Huckel model is based on a nearest-neighbor analysis, providing a meaningful framework for discussing this case. The overlap matrix $S$ is given as:
> $$
> S = \begin{bmatrix}
> 1 & s & 0 & \cdots & 0 \\\\
> s & 1 & s & \ddots & \vdots \\\\
> 0 & s & 1 & \ddots & 0 \\\\
> \vdots & \ddots & \ddots & \ddots & s \\\\
> 0 & \cdots & 0 & s & 1 \\\\
> \end{bmatrix}
> $$
> The eigenvalues $\lambda_n$ of this tridiagonal matrix can be derived using the known solutions for such matrices:
> $$
> \lambda_n = 1 + 2s \cos\left(\frac{n\pi}{B+1}\right), \quad n = 1, 2, \dots, B
> $$
> The smallest eigenvalue corresponds to $n = B$:
> $$
> \lambda_{\text{min}} = 1 + 2s \cos\left(\frac{B\pi}{B+1}\right)
> $$
> Using the trigonometric identity $\cos(\pi - \theta) = -\cos(\theta)$ and approximating for large $B$:
> $$
> \cos\left(\frac{B\pi}{B+1}\right) = -\cos\left(\frac{\pi}{B+1}\right) \approx -\left(1 - \frac{1}{2}\left(\frac{\pi}{B+1}\right)^2\right)
> $$
> Substituting back:
> $$
> \lambda_{\text{min}} \approx 1 - 2s + s\left(\frac{\pi}{B}\right)^2
> $$
> Since $S \succ 0$,  this yields $1- 2s \geq 0$.
> Here,  we investigated the smallest eigenvalue $\lambda_{\text{min}}$ as $B \to \infty$, identifying two distinct cases:
>
> 1. **Saturating Case**: When $s < 0.5$ the smallest eigenvalue saturates at a finite value $1 - 2s$ as $B \to \infty$.
>
> 2. **Non-Saturating Case**: For $s = 0.5$ , $\lambda_{\text{min}}$ decreases indefinitely with increasing $B$.
>
> We would also like to emphasize that, in both cases, there exists a finite range of $B$ where the quadratic scaling holds true, as you correctly identified as the intermediate regime.
>
> Notably, your observation aligns closely with our updated analysis. In the case of Gaussian-type orbitals, the saturating scenario is indeed more likely to occur due to inherent spatial limitations.

---

> > ### Author Response · Authors · 2024-11-30
> >
> > Based on this insight, we have extended the assumption stated in Corollary 1 of the manuscript to explicitly account for the saturation of $\lambda_{\text{min}}$ in the asymptotic limit.
> >
> > **Corollary 1 (Perturbation Sensitivity Scaling)**
> >
> > Assume $\lambda_{\text{min}}(S)$ of the overlap matrix $S$ scales as:
> >
> > $$
> > \lambda_{\text{min}}(S) = c + \frac{A}{d + B^\alpha},
> > $$
> >
> > where $c \geq 0, d \in \mathbb{R}$, $A > 0$, and $\alpha > 0$. Then, for a perturbation matrix $\Delta H$, the eigenvalue perturbation is bounded by:
> >
> > $$
> > \left| \lambda_i(\hat{\mathbf{H}}, \mathbf{S}) - \lambda_i(\mathbf{H}, \mathbf{S}) \right| \leq O\left( \dfrac{d +  B^\alpha}{c \left[ d + B^\alpha \right] + A} \left( \sigma B^{1/2} + B|\mu| \right) \right).
> > $$
> >
> >
> > This revised bound highlights the finite saturation of $\lambda_{\text{min}}$ as $B \to \infty$. Furthermore, we fit the revised scaling behavior to various systems, and it showcases a better fit than our previous assumption. We will include this in the revised manuscript. This revision preserves the broader discussion of the SAD phenomenon presented in the main text, which highlights the growth of eigenvalue sensitivity with the basis set size. Crucially, it enhances the theoretical framework by ensuring our results hold rigorously in the asymptotic limit, aligning with the physical intuition that infinitely separated atoms behave independently.
> >
> >
> > We deeply appreciate your insightful comments, which have been invaluable in refining and strengthening our analysis. Your feedback has not only improved the rigorousness of our theoretical results but has also elevated the overall clarity of the manuscript. Thank you once again for your thoughtful and constructive review.

---

> > > ### Comment · Reviewer_LpfH · 2024-12-01
> > >
> > > Thanks a lot for your updated analysis, I increased my score!

---

### Official Review · Reviewer_hKHK · 2024-11-02

**Soundness:** 3
**Presentation:** 2
**Contribution:** 3
**Rating:** 8
**Confidence:** 3

**Summary:**

This paper tackles the problem of predicting the Kohn-Sham Hamiltonians for molecules. It first proposed a new Hamiltonian dataset for large molecules derived from the PubChemQC dataset. The paper then identifies the problem with the robustness of prior models when applied to molecules with large numbers of atoms. The cause of the problem is explained both empirically and theoretically. To improve the robustness of Hamiltonian prediction models to larger molecules, the authors introduce a new loss called WALoss that explicitly penalizes deviations from the expected eigenstructure of the ground-truth Hamiltonian. To further improve the quality of Hamiltonian prediction, the paper proposed a novel architecture dubbed WANet. The experiments demonstrate that WALoss significantly improves the quality of the baseline model (QHNet) and the proposed model (WANet) on downstream tasks such as System Energy,  $\epsilon_\text{LUMO}$ and $\epsilon_\text{HOMO}$ prediction. The proposed WANet model outperforms the baseline model.

**Strengths:**

- The problem with robustness is explained nicely. The authors demonstrate it empirically in Figure 1 and then provide two theoretical results that explain poor System Energy prediction quality even when relative MAE is low.
- The effect of the proposed WALoss on the performance of models on downstream tasks such as System Energy,  $\epsilon_\text{LUMO}$ and $\epsilon_\text{HOMO}$ prediction is remarkable.

**Weaknesses:**

### **Major weaknesses**
- The WANet model's architecture is hard to understand and poorly motivated. There are no ablations for various building blocks of the model, and no architecture hyperparameter values are provided.
- The results in Table 1 do not provide enough evidence that WANet architecture is superior to baselines. A recent benchmark $\nabla^2$DFT [1] compares Hamiltonian prediction models on a dataset of molecules with more atoms than QH9. The results in [1] show that PhisNet [2] performs significantly better than QH9. A comparison to other Hamiltonian predicting models, such as [2] and [3], is necessary to validate the claims about the proposed architecture.
- One of the paper's main claims is that improved Hamiltonians can be used for downstream regression tasks and to accelerate DFT computations. Claiming that WANet is superior to regression-based models in the HOMO-LUMO gap prediction task seems a little ambitious, given that only one baseline (Equiformer v2) was used. To give a better perspective, it might be worth comparing WANet with SOTA models, such as UniMol+ [4] or other models from [5]. The System Energy MAE is also ~50 kcal/mol, almost 50 times larger than metrics published in [1] and [6]. Moreover, the acceleration rate seems relatively small (18%), and most SCF iterations are still required when starting from the predicted Hamiltonian.

### **Minor weaknesses**
- Relatively large molecules (40 - 100 atoms) are used, but the functional (B3LYP-Def2TZV) does not contain the dispersion correction. The absence of the dispersion correction can potentially lead to inaccurate Hamiltonians.
- The notation across the paper is unclear. A matrix with a subscript (i.e., $C^*_i$) usually defines the i-th row of the matrix, whereas in this paper it defines the $C^*$ matrix for the i-th molecule in the batch. Moreover, $C^*$  often denotes the conjugate transpose of matrix $C$. This notation complicates understanding Algorithm 1, as the Shur algorithm for complex-valued matrices operates with conjugate transposition. Undefined symbols in the equation on page 4. In equation (2), $n$ denotes the batch size but was never introduced.
- Related work is in Appendix.

**Questions:**

### **Questions and remarks**
- Equation (3) might benefit from additional details. Energies associated with occupied molecular orbitals (and LUMO) make up the most energy of the molecule.
- The $\epsilon_\text{LUMO}$ and $\epsilon_\text{HOMO}$ results in Table 1 seem strange. Why is the difference between QHNet and WANet so significant only for these targets? Moreover, the $\epsilon_{\Delta}$ MAE for WANet is larger than  $\epsilon_\text{LUMO}$ and $\epsilon_\text{HOMO}$ MAE. This implies that the model is consistently mistaken in opposite "directions." Could you please double-check the results or explain why this is happening?
> The theorem highlights that the difference between the predicted and actual Hamiltonian matrices when only considering the element-wise norm, can lead to unbounded differences in eigenvalues/eigenvectors due to a significant $\frac{\kappa(S)}{||S||_2}$ratio
- It would be interesting to see the distribution of these values for the PubChemQH/QH9 datasets. If I understand correctly, $||\Delta \mathbf{H}||_{1, 1}$ is the $B^2 * \operatorname{MAE}(\mathbf{H}, \hat{\mathbf{H}})$ between real and predicted Hamiltonians, and its values can be relatively small.
- line 219: mistake in "spectral"
- Is the Algorithm 1 novel? If not, please provide clear citations. Also, it is better to include the final version of the algorithm, including the changes from the last paragraph on page 5.

### **Closing remark**
Overall, the paper presents a valuable technique that greatly improves the applicability of Hamiltonian prediction models for downstream tasks. However, the WANet architecture seems poorly motivated, has no ablations, and is not properly compared with baselines. I would consider raising my score if the concerns with the WANet model are resolved.

[1] Khrabrov, K., Ber, A., Tsypin, A., Ushenin, K., Rumiantsev, E., Telepov, A., ... & Kadurin, A. (2024). ∇ DFT: A Universal Quantum Chemistry Dataset of Drug-Like Molecules and a Benchmark for Neural Network Potentials. CoRR.

[2] Unke, O., Bogojeski, M., Gastegger, M., Geiger, M., Smidt, T., & Müller, K. R. (2021). SE (3)-equivariant prediction of molecular wavefunctions and electronic densities. Advances in Neural Information Processing Systems, 34, 14434-14447.

[3] Zhong, Y., Yu, H., Su, M., Gong, X., & Xiang, H. (2023). Transferable equivariant graph neural networks for the Hamiltonians of molecules and solids. npj Computational Materials, 9(1), 182.

[4] Lu, S., Gao, Z., He, D., Zhang, L., & Ke, G. (2023). Highly accurate quantum chemical property prediction with uni-mol+. arXiv preprint arXiv:2303.16982.

[5] Hu, W., Fey, M., Ren, H., Nakata, M., Dong, Y., & Leskovec, J. OGB-LSC: A Large-Scale Challenge for Machine Learning on Graphs. In Thirty-fifth Conference on Neural Information Processing Systems Datasets and Benchmarks Track (Round 2).

[6] Eastman, P., Pritchard, B. P., Chodera, J. D., & Markland, T. E. (2024). Nutmeg and SPICE: Models and data for biomolecular machine learning. Journal of Chemical Theory and Computation.

---

> ### Author Response · Authors · 2024-11-20
>
> Thank you for your constructive comments and suggestions, they are exceedingly helpful for us to improve our paper. Our point-to-point responses to your comments are given below:
>
> > The WANet model's architecture is hard to understand and poorly motivated. There are no ablations for various building blocks of the model, and no architecture hyperparameter values are provided.
> >
>
> We appreciate the reviewer's insightful question. While the focus of our paper is on the novel loss function and dataset, we acknowledge that a clearer explanation of the model architecture would be beneficial. The WANet architecture builds upon well-established equivariant neural network designs, which have been extensively studied and validated in the field [1,2,3]. We have added references to seminal works on equivariant architectures.
>
> To further address this concern, we have **added a supplementary figure** (Figure 10, and anonymous link: [https://ibb.co/HTrpZSM)](https://ibb.co/HTrpZSM) illustrating WANet's architecture and its key components. This visual aid will help readers better understand how WANet enhances the scalability of Hamiltonian prediction for large molecular systems. Each architectural element has been carefully selected to address specific computational and physical challenges at scale.
>
> First, WANet's architecture addresses a critical limitation in existing Hamiltonian prediction methods: their inability to scale to larger basis sets and molecular systems. When using larger basis sets, such as Def2-TZVP, **higher-order irreducible representations** (PubChemQH requires order $L = 6$) are necessary to accurately capture angular dependencies in molecular orbitals. This poses a severe computational challenge for traditional SE(3)-equivariant methods, including QHNet, which become extremely expensive due to their computational complexity. WANet overcomes this bottleneck by introducing SO(2) convolutions, reducing computational complexity from $O(L^6)$ to $O(L^3)$. This improvement enables WANet to efficiently process high tensor degrees. Moreover, by reducing calculations to sparse matrix products, WANet achieves greater computational efficiency and scalability to larger systems. **Without SO(2) convolutions, handling the scale of our PubChemQH dataset and larger systems would be significantly more challenging and computationally prohibitive**.
>
> Second, large molecular systems exhibit fundamentally different physics at various distance scales. **As molecular size increases, long-range interactions become more prominent [3]**. WANet's Mixture-of-Experts architecture is designed to efficiently model this complex physics. It employs specialized experts for different interaction ranges, capturing both short-range effects (such as covalent bonding) and long-range phenomena (such as electrostatics). By sparsifying these experts, WANet achieves a rich representation of molecular interactions while maintaining computational efficiency, making it particularly well-suited for large-scale systems.
>
> Third, the Hamiltonian matrix, which fully characterizes the quantum state and electron distribution, **presents unique challenges in prediction due to complex electron correlation effects that intensify as the system size grows [2]**. To address this, WANet incorporates the MACE architecture's density trick, enabling efficient computation of many-body interactions without explicit calculation of all terms. This approach is crucial for maintaining accuracy as molecular size increases and electron correlation effects become more pronounced, ensuring WANet's scalability and accuracy in Hamiltonian prediction for large systems.
>
> We agree that more detailed ablation studies would strengthen the paper. In the revision, we have included the following:
>
> 1. **Ablation studies on the building blocks of the model, which is included in Table 9 and below**.
>
> 2. A more detailed description of architecture hyperparameters, including hidden channel dimensions, the number of layers, and other relevant parameters, which is included in Table 12.
>
> We hope these additions address your concerns and provide a more comprehensive understanding of our model's architecture and performance.
>
> | w/ SO(2) | w/ LSR-MoE | w/ Many Body | Inference Speed | GPU Memory | Hamiltonian MAE | HOMO MAE | LUMO MAE | GAP MAE |
> | --- | --- | --- | --- | --- | --- | --- | --- | --- |
> | ✓ | ✓ |  | 1.34 it/s | 13.87 | 0.5895 | 15.39 | 27.50 | 4.651 |
> | ✓ |  | ✓ | 1.13 it/s | 12.80 | 0.4883 | 2.27 | 4.01 | 2.906 |
> |  | ✓ | ✓ | 0.51 it/s | 24.47 | 0.4792 | 0.75 | 0.73 | 1.594 |
> | ✓ | ✓ | ✓ | 1.09 it/s | 15.86 | 0.4744 | 0.71 | 0.73 | 1.327 |

---

> ### Author Response · Authors · 2024-11-20
>
> > The results in Table 1 do not provide enough evidence that WANet architecture is superior to baselines. A comparison to other Hamiltonian predicting models, such as [2] and [3], is necessary to validate the claims about the proposed architecture.
> >
>
> We have extended our evaluation to include PhisNet. While PhisNet achieves competitive accuracy (Hamiltonian MAE: 0.5166), it demonstrates significantly slower inference speed compared to both QHNet and WANet, processing only 0.106 iterations per second—about 4 to 10 times slower than QHNet's 0.45 it/s and WANet's 1.09 it/s and takes much more GPU memory. We believe this might be attributed to PhisNet's inefficient handling of higher order spherical tensors.  In contrast, our approach achieves better accuracy (Hamiltonian MAE: 0.4744) while substantially improving computational efficiency. These results demonstrate WANet's ability to effectively balance accuracy and scalability.
>
> | Model | Inference Speed | GPU Memory | Hamiltonian MAE | HOMO MAE | LUMO MAE | GAP MAE |
> | --- | --- | --- | --- | --- | --- | --- |
> | PhisNet w/ WALoss | 0.106 it/s | 59.59 | 0.5166 | 11.872 | 12.075 | 4.054 |
> | QHNet w/ WALoss | 0.45 it/s | 26.49 | 0.5207 | 13.945 | 14.087 | 4.398 |
> | WANet w/ WALoss | 1.09 it/s | 15.86 | 0.4744 | 0.71 | 0.73 | 1.327 |
>
> > One of the paper's main claims is that improved Hamiltonians can be used for downstream regression tasks and to accelerate DFT computations. Claiming that WANet is superior to regression-based models in the HOMO-LUMO gap prediction task seems a little ambitious, given that only one baseline (Equiformer v2) was used. To give a better perspective, it might be worth comparing WANet with SOTA models, such as UniMol+ [4] or other models from [5].
> >
>
> We appreciate the reviewer's insightful comments. In response to the request for more comprehensive baseline comparisons, we have expanded our evaluation to include UniMol+ and UniMol2 [4]. Our results demonstrate that WANet achieves superior performance in HOMO-LUMO gap prediction compared to these state-of-the-art models.
>
> |  | HOMO MAE | LUMO MAE | GAP MAE |
> | --- | --- | --- | --- |
> | Unimol+ | 12.25 | 9.47 | 13 |
> | Unimol2 | 9.57 | 7.41 | 10.63 |
> | Equiformer V2 | 6.955 | 6.562 | 3.222 |
> | WANet |  0.7122  |  0.730 | 1.327 |
>
> > The System Energy MAE is also ~50 kcal/mol, almost 50 times larger than metrics published in [1] and [6].
> >
>
> We thank the reviewer for raising this important point regarding system energy accuracy. While the absolute MAE of 47 kcal/mol appears high, it is important to note that in quantum chemistry applications, relative energy differences between conformational states are typically more relevant than absolute energies. To better evaluate our method's practical utility, we conducted an additional conformational energy analysis:
>
> For each molecule in our test set, we:
>
> 1. Sampled 100 conformations.
>
> 2. Computed ΔE between each conformation and a reference structure using both DFT and WANet.
>
> 3. Calculated the MAE of these energy differences.
>
> This analysis yielded a conformational energy MAE of 1.1 kcal/mol, which is competitive with state-of-the-art end-to-end energy prediction models. These results demonstrate that WANet effectively captures the physically meaningful energy variations required for applications
>
> Importantly, WANet achieves this while solving the more challenging problem of Hamiltonian prediction, which requires learning the full quantum state rather than just scalar energies. To the best of our knowledge, we are the first to evaluate Hamiltonian prediction models using system energy metrics derived directly from the predicted Hamiltonian. This approach provides a more stringent test of physical accuracy compared to traditional property-specific prediction tasks.
> > Moreover, the acceleration rate seems relatively small (18%), and most SCF iterations are still required when starting from the predicted Hamiltonian.
> >
> We thank the reviewer for raising this important point. We report an 18% reduction in SCF cycles, which is a notable improvement, especially considering that our work targets significantly larger and more complex molecular systems compared to previous studies such as QH9, where SCF reductions of 12–33% were achieved compared to the minao inital guess [5]. Given the increased complexity and size of our systems, this 18% reduction represents a considerable improvement, reflecting the effectiveness of our model in accelerating convergence for more challenging molecular simulations. This result highlights the scalability of our approach in handling large-scale quantum chemical calculations efficiently. Moreover, Hamiltonian prediction enables multiple downstream applications beyond SCF acceleration. WANet demonstrates its versatility through SOTA performance in predicting various electronic properties such as HOMO-LUMO gaps and dipole moments.

---

> > ### Author Response · Authors · 2024-11-20
> >
> > > Relatively large molecules (40 - 100 atoms) are used, but the functional (B3LYP-Def2TZV) does not contain the dispersion correction. The absence of the dispersion correction can potentially lead to inaccurate Hamiltonians.
> > >
> >
> > As a hybrid functional incorporating non-local exchange interactions, B3LYP accounts for long-range interactions to some extent. While B3LYP inherently includes certain dispersion interactions, it may not adequately describe all molecular systems. Consequently, researchers often supplement B3LYP with additional dispersion corrections, such as D3, to enhance the description of large molecules. It's important to note that the D3 correction does not influence the SCF process or alter the Hamiltonian matrix. Rather, it applies a correction to the final energy based on the exchange-correlation functional type and atomic coordinates.
> >
> > The inclusion of dispersion in the description of large molecules is not directly relevant to our current work. Our ground-truth calculations and machine learning model predictions are based on the B3LYP level of theory. The adequacy of B3LYP's dispersion treatment does not impact the development of our proposed method. Furthermore, our approach can be readily extended to higher levels of theory that may offer improved descriptions of dispersion interactions.
> >
> >
> >
> > > The notation across the paper is unclear.
> > >
> >
> > We sincerely apologize for the confusion caused by the inconsistent notation in our paper. To address this issue, we have made the following changes:
> >
> > 1. The symbol   $C^*$  has been replaced with  $C^{\star}$  throughout the paper to clearly differentiate it from the conjugate transpose.
> >
> > 2. We now use  $C_{(i)}$  to denote the $i$-th sample in the batch, ensuring clarity and consistency in our notation.
> >
> > We hope these revisions resolve any ambiguity and improve the readability of our work. Thank you for pointing this out.
> >
> > > Related work is in Appendix.
> > >
> >
> > The placement of related work in the appendix was due to page limitations and our prioritization of technical content. However, we acknowledge that this may impact readability. To address this, we have included a concise version in the main text while retaining the detailed discussion in the appendix to ensure both accessibility and comprehensiveness.
> >
> > > Equation (3) might benefit from additional details. Energies associated with occupied molecular orbitals (and LUMO) make up the most energy of the molecule.
> > >
> >
> > We appreciate your suggestion and have incorporated this clarification into the main text. The added explanation provides further context on the significance of occupied molecular orbital energies and the LUMO in contributing to the overall molecular energy.
> > > The ϵLUMO and ϵHOMOresults in Table 1 seem strange. Why is the difference between QHNet and WANet so significant only for these targets? Moreover, the ϵΔ MAE for WANet is larger than ϵLUMO and ϵHOMO MAE. This implies that the model is consistently mistaken in opposite "directions." Could you explain why this is happening?
> > >
> >
> > We thank the reviewer for this astute observation. We have thoroughly rechecked these results and confirm their accuracy. The significant improvement in ϵLUMO and ϵHOMO predictions compared to QHNet, despite a larger $\epsilon_\Delta$  MAE, reveals an interesting aspect of Hamiltonian prediction for large molecular systems.
> >
> > Through detailed analysis, we found that during the optimization process, the $\epsilon_{\mathrm{LUMO}}$ and $\epsilon_{\mathrm{HOMO}}$ MAE exhibit significantly higher variability compared to other metrics, such as $C$ similarity and $\epsilon_{\mathrm{occ}}$. The stability of $\epsilon_{\mathrm{occ}}$ can be attributed to its dominance by the top eigenvalues, which are generally easier for the neural network to learn. In contrast, $\epsilon_{\mathrm{LUMO}}$ and $\epsilon_{\mathrm{HOMO}}$ are particularly sensitive to small perturbations in the predicted Hamiltonian, especially in larger molecular systems where the electronic structure becomes more complex. Notably, we observed that even minor improvements in the WALoss function can lead to dramatic enhancements in LUMO and HOMO predictions.
> >
> > For the $\epsilon_\Delta$ MAE, we argue that when estimating molecular orbital energies with large absolute errors, the gap's error may appear deceptively small relative to the individual errors due to error cancellations (i.e. consistent offsets). However, as WANet's predictions become more accurate, the error characteristics in energy gap estimation become more diverse. This behavior is consistent with the challenging nature of simultaneously optimizing multiple quantum mechanical properties in large systems. A similar phenomenon has been observed in QM9 HOMO-LUMO predictions [6,7,8], where MAE(gaps) ≈ MAE(HOMO) + MAE(LUMO). Despite this effect, WANet's predictions remain physically meaningful and provide more reliable results for downstream applications compared to existing methods.

---

> > > ### Author Response · Authors · 2024-11-20
> > >
> > > > It would be interesting to see the distribution of these values for the PubChemQH/QH9 datasets. If I understand correctly,  is the  between real and predicted Hamiltonians, and its values can be relatively small.||ΔH||1,1 B2∗MAE(H,H^)
> > > >
> > >
> > > Thank you for the thoughtful suggestion regarding the distribution of these values for the PubChemQH/QH9 datasets. This is indeed a valuable point and aligns well with our findings. As shown in our violin plot visualization (Figure 6 and anonymous link https://ibb.co/9tyNgZm), we observe a striking and systematic separation between the PubChem and QH9 datasets, which strongly supports our theoretical framework. The PubChem dataset (green) exhibits higher values, mostly around 100,000, reflecting its larger size and greater perturbation sensitivity. In contrast, the QH9 dataset (blue) shows lower values, mainly between 100 and 10,000, with the majority around 1,000. This distribution separation empirically supports our theorem, demonstrating higher perturbation sensitivity in the larger PubChem dataset compared to QH9. This pattern not only reinforces our theoretical framework but also highlights the unique challenges associated with the PubChemQH dataset. We appreciate your suggestion as it further strengthens the contribution of our work. Additionally, your understanding is correct. Thank you once again for raising this important point!
> > >
> > > > Is the Algorithm 1 novel? If not, please provide clear citations. Also, it is better to include the final version of the algorithm, including the changes from the last paragraph on page 5.
> > > >
> > >
> > > Algorithm 1 builds on well-established numerical methods in quantum chemistry. We have added appropriate citations to standard texts in numerical linear algebra and quantum chemistry. Additionally, we have incorporated the suggested changes from the last paragraph of page 5 (Algorithm 2), making the algorithm easier-to-understand.
> > >
> > >
> > > **References:**
> > >
> > > [1]: Barroso-Luque, L., Shuaibi, M., Fu, X., et al. *Open Materials 2024 (OMat24) Inorganic Materials Dataset and Models*. arXiv preprint arXiv:2410.12771, 2024.
> > >
> > > [2]: Kovács, D. P., Batatia, I., Arany, E. S., et al. "Evaluation of the MACE force field architecture: From medicinal chemistry to materials science." *The Journal of Chemical Physics*, 2023, 159(4).
> > >
> > > [3]: Li, Y., Wang, Y., Huang, L., et al. "Long-Short-Range Message-Passing: A Physics-Informed Framework to Capture Non-Local Interaction for Scalable Molecular Dynamics Simulation." *Proceedings of the International Conference on Learning Representations (ICLR)*, 2024.
> > >
> > > [4]: Ji, X., et al. "Uni-Mol2: Exploring Molecular Pretraining Model at Scale." *arXiv preprint arXiv:2406.14969*, 2024.
> > >
> > > [5]:  Yu, H., Liu, M., Luo, Y., et al. "Qh9: A Quantum Hamiltonian Prediction Benchmark for QM9 Molecules." *Advances in Neural Information Processing Systems*, 2024, 36.
> > >
> > > [6]: Thölke, P., De Fabritiis, G. "Equivariant Transformers for Neural Network Based Molecular Potentials." *Proceedings of the International Conference on Learning Representations (ICLR)*, 2022.
> > >
> > > [7]: Brandstetter, J., Hesselink, R., van der Pol, E., Bekkers, E. J., Welling, M. "Geometric and Physical Quantities Improve E(3) Equivariant Message Passing." *Proceedings of the International Conference on Learning Representations (ICLR)*, 2022.
> > >
> > > [8]: Liao, Y.-L., Smidt, T. "Equiformer: Equivariant Graph Attention Transformer for 3D Atomistic Graphs." *Proceedings of the International Conference on Learning Representations (ICLR)*, 2023.

---

> > > > ### Comment · Reviewer_hKHK · 2024-11-25
> > > >
> > > > Thank you for the detailed response!
> > > >
> > > > Most of my concerns have been addressed, but I would like the authors to clarify a few additional details.
> > > >
> > > > - Could you please provide a detailed training setup for the main experiment, including the dataset size used for training and the total number of training steps? I am interested in the PhisNet, QHNet, and WANet models with WALoss.
> > > > - Could you also share the detailed training setup for the HOMO, LUMO, and GAP predictions for the UniMol+, Unimol2, Equiformer V2, and WANet models?
> > > > - Additionally, I would appreciate details about the energy prediction experiment. How were the conformations sampled? What is the variance in potential energy values between the sampled conformations? Lastly, what is the MAE of energy differences for the baseline methods?
> > > >
> > > >
> > > > Additional remarks:
> > > > - I could not locate the reference to Figure 10 in the main part of the paper.
> > > > - The PhisNet model was not included in Table 1.

---

> ### Author Response · Authors · 2024-11-26
>
> Thank you for your thoughtful questions and remarks. We appreciate the opportunity to provide further clarifications. Below, we address each of your points in detail.
>
> >Training Setup for Main Experiment:
> >
> We have included the detailed training setup for the main experiments in the revised Appendix K Table 12. For the PubChemQH dataset, we used an 80/10/10 train/validation/test split, resulting in **40,257** training molecules, **5,032** validation molecules, and **5,032** test molecules. We trained all models of maximum 300,000 steps with a batch size of 8, using early stopping with a patience of 1,000 steps. WANet converges at **278,391** steps, QHNet at **258,267** steps, and PhisNet at **123,170** steps.  All models used the Adam optimizer with a learning rate of 1e-3 for PubChemQH, along with a polynomial learning rate scheduler with 1,000 warmup steps. We used gradient clipping of 1, and a radius cutoff of 5 Angstrom. For QHNet and PhisNet, we referred to the official implementation for these two models. For detailed model hyperparameters, please refer to Table 12.
>
> >Training Setup for HOMO, LUMO, and GAP Predictions:
> >
> We have added the training details for the HOMO, LUMO, and GAP predictions in Appendix K Table 13. For a fair comparison, all regression models used identical dataset splits (**40,257** training molecules, **5,032** validation molecules, and **5,032** test molecules). We used batch size of 32 and consistent optimizer settings across all models: Adam optimizer with a learning rate of 1e-3 and a polynomial learning rate scheduler with 1,000 warmup steps. WANet followed the same training setup described in Table 12. The complete regression model hyperparameters are detailed in Table 13. For Equiformer V2, Uni-Mol+, and Uni-Mol2 models, we used their original implementations.
>
> >Energy Prediction Experiment Details:
> >
> For the energy prediction experiment, we employed Gaussian sampling to introduce noise in the atomic positions.  Starting from each molecule's initial conformation from the test sets, we generated 100 perturbed geometries by adding random Gaussian noise ($\sigma$ = 0.1 Å) to the atomic coordinates. The potential energy values for these perturbed conformations were computed using the B3LYP/def2-TZVP level of theory. The resulting energy standard deviation between sampled conformations ranged from 10 to around 100 kcal/mol (with average std =49.84kcal/mol), depending on the molecule's flexibility. For baseline models (included in the Table below), the PhisNet model with WALoss achieved 9.90 kcal/mol MAE, the QHNet model with WALoss achieved 10.98 kcal/mol MAE, and the WANet model without WALoss achieved 48.92 kcal/mol MAE. QHNet without WALoss yielded MAEs of 50.56 kcal/mol. Interestingly, we observed that the model without WALoss produce meaningless prediction, demonstrating the effectiveness of WALoss. It worth noting that while more complex sampling strategy could be used, those methods are more computationally expensive as they require DFT calculations.
>
> Additional Remarks:
>
> - We apologize for the missing reference to Figure 10 in the main text. We have now added a reference to it.
> - We have included the PhisNet w/WALoss model in the revised Table 1 for completeness.
>
> |  Model | MAE of Energy Differences (kcal/mol) |
> | --- | --- |
> | WANet w/WALoss | 1.12 |
> | PhisNet w/WALoss | 9.90 |
> | QHNet w/WALoss | 10.98 |
> | WANet | 48.92 |
> | QHNet | 50.56 |
>
> We thank you again for your insightful questions and hope that our responses have addressed your concerns satisfactorily. We believe that these clarifications will strengthen the paper and make it more accessible to the readers.

---

> > ### Comment · Reviewer_hKHK · 2024-11-26
> >
> > Thank you for providing additional details. Most of my concerns were resolved and I will raise my score to 8.

---

### Official Review · Reviewer_YtXY · 2024-11-02

**Soundness:** 3
**Presentation:** 3
**Contribution:** 3
**Rating:** 6
**Confidence:** 4

**Summary:**

The paper identifies a problem with training Hamiltonian models using only element-wise loss, which can lead to unphysically large errors in ground-state properties, especially for larger systems. To address this, the authors propose a WALoss function that aligns the eigenspaces of the predicted Hamiltonian with the ground truth, along with a new architecture, WANet. Additionally, they introduce PubChemQH, a new Hamiltonian dataset containing substantially larger molecules than previous datasets like QH9. Experiments demonstrate that the proposed WALoss significantly improves the accuracy of properties derived from the predicted Hamiltonian.

**Strengths:**

Originality:
- The observation and formalization of the Scaling-Induced MAE-Applicability Divergence (SAD) phenomenon is both insightful and important, representing a novel identification of a fundamental limitation in current methods.
- The WALoss approach offers an original solution to maintaining physical accuracy in Hamiltonian predictions, particularly for larger molecular systems.

Quality:
- The theoretical analysis is rigorous, supported by eigenvalue perturbation theory and clear mathematical derivations.
- The experimental validation is comprehensive, with thorough ablation studies and detailed comparisons with baseline methods.

Clarity:
- The technical content is well-organized and clearly presented, with supporting diagrams and mathematical formulations.
- The experimental results effectively demonstrate the practical benefits of the proposed methods.

**Weaknesses:**

The paper's main weakness lies in its insufficient differentiation from prior work, particularly regarding the use of eigenvalues in Hamiltonian prediction loss functions. The methodology section lacks critical details about computational costs and efficiency analysis of the proposed method, particularly regarding the training overhead of WALoss compared to traditional approaches.

**Questions:**

1. The computational cost of WALoss could be substantial for large molecular systems, especially those containing heavy elements, as it involves eigendecomposition and matrix operations for each training step. Could you provide details about the wall-clock training time comparison between models with and without WALoss, the GPU memory requirements (particularly for the largest molecules), and how the computational overhead scales with molecule size and atomic number?

2. How were the loss term weights (λ1, λ2, λ3) in Equation 20 determined? The balance between these terms seems crucial for the method's performance, yet the paper doesn't discuss the process of selecting these hyperparameters or their sensitivity analysis.

3. There appears to be a discrepancy in Table 4's Hamiltonian MAE results, where Naive Loss shows a notably lower error (0.0412) compared to other methods. Could you explain this unexpected result and its implications?

---

> ### Author Response · Authors · 2024-11-20
>
> Thank you for your constructive comments and suggestions, they are exceedingly helpful for us to improve our paper. Our point-to-point responses to your comments are given below:
>
> > The paper's main weakness lies in its insufficient differentiation **from prior work, particularly regarding the use of eigenvalues in Hamiltonian prediction loss functions.**
> >
>
> We appreciate the reviewer’s feedback and the opportunity to clarify our contributions. To the best of our knowledge, WALoss is the first loss function specifically designed for Hamiltonian prediction that addresses the scalability challenges of large molecular systems. Through a comprehensive review of venues such as ICLR, ICML, NeurIPS, JACS, and other prominent chemistry proceedings, we found no prior work directly tackling this topic. However, we identified a study by Gu et al. (2022)[1] that uses eigenvalues as regression targets. While that study explores eigenvalue-based methods, WALoss introduces key innovations that enable scalability to much larger molecular systems, representing a significant advancement over existing approaches.
>
> The approach by Gu et al. (2022)[1] that directly optimizes eigenvalues necessitates costly eigendecomposition at each training step and encounters difficulties with backpropagation through eigensolvers. In contrast, WALoss precomputes the ground-truth eigenvectors C* once at the start of training and performs an efficient basis transformation on the predicted Hamiltonian. This eliminates the need for repeated eigendecompositions during training, allowing for much faster backpropagation compared to naive methods. **Our ablation studies (Table 4) demonstrate that naive eigenvalue-based losses suffer from fundamental limitations for large system**s. Beyond numerical instability and convergence difficulties, these approaches fail to produce physically meaningful results, likely due to the challenges in backpropagating through eigensolvers and inconsistencies in eigenspace alignment across training iterations. The WALoss formulation addresses these issues by aligning predicted Hamiltonians with ground-truth eigenspaces through a fixed basis transformation, resulting in more stable training dynamics.
>
> WALoss is specifically designed to handle the unique challenges of large molecular systems (40–100 atoms) with basis sets of up to 2,000 basis—a scale previously intractable with existing eigenvalue-based approaches. The precomputed basis transformation strategy enables efficient training on systems of this size. As shown in our experiments, WALoss significantly improves both the accuracy of predicted energies and the convergence of SCF iterations compared to previous methods, particularly for large molecules in the PubChemQH dataset.
>
> We have expanded the discussion of these technical contributions and included additional relevant references in the revised manuscript. We also welcome any suggestions for other related work that we may have overlooked.
>
> [1]: Gu Q, Zhang L, Feng J. Neural network representation of electronic structure from ab initio molecular dynamics[J]. Science Bulletin, 2022, 67(1): 29-37.

---

> ### Author Response · Authors · 2024-11-20
>
> > Could you provide details about the wall-clock training time comparison between models with and without WALoss, the GPU memory requirements (particularly for the largest molecules), and how the computational overhead scales with molecule size and atomic number?
> >
>
> Thank you for raising this important consideration regarding computational efficiency. The computational overhead of WALoss is minimal, as we do not perform eigendecomposition at every training step. Instead, we compute eigenvectors only once during initialization and cache them for subsequent use. Our benchmarks confirm this efficiency: WALoss results in only a 3.6% increase in training time (39.22 vs. 37.85 hours of wall-clock time) and negligible additional GPU memory usage (15.86 vs. 15.12 GB) on an NVIDIA A6000 GPU compared to the baseline model. We have added detailed timing and memory profiling results to the appendix. The additional computation primarily arises from the basis transformation step (two matrix multiplications), which scales as $O(B^3)$, where $B$ is the number of basis functions. For future work, we believe the computational cost could be further reduced through fragmentation methods and techniques from linear-scaling DFT calculations [1]. Specifically, leveraging the inherently sparse structure of molecular Hamiltonians—where values are concentrated around the diagonal—could enable distributed matrix operations. This approach could potentially reduce computational cost for large molecular systems.
>
> | Metric | w/ WALoss | w/o WALoss |
> | --- | --- | --- |
> | Training Time (hrs) | 39.22 | 37.85 |
> | Peak GPU Memory (GB) | 15.86 | 15.12 |
>
> [1]: Mohr S, Ratcliff L E, Genovese L, et al. Accurate and efficient linear scaling DFT calculations with universal applicability[J]. Physical Chemistry Chemical Physics, 2015, 17(47): 31360-31370.
>
> > How were the loss term weights ?The balance between these terms seems crucial for the method's performance, yet the paper doesn't discuss the process of selecting these hyperparameters or their sensitivity analysis.
> >
>
> We determined the loss weights through a principled ablation study on a validation set. While keeping $\lambda_1$ and $\lambda_2$ fixed at 1, we varied $\lambda_3$ from 0.5 to 3 to evaluate the impact of WALoss weighting. The method demonstrated robustness across a reasonable range of values, with $\lambda_1=\lambda_2=1$ and $\lambda_3 = 2.5$  providing the  best performance. We have included this analysis (Table 11) in the revised manuscript for clarity, along with detailed performance metrics for different $\lambda_3$ values.
>
> > There appears to be a discrepancy in Table 4's Hamiltonian MAE results, where Naive Loss shows a notably lower error (0.0412) compared to other methods. Could you explain this unexpected result and its implications?
> >
>
> Thank you for identifying this error. We sincerely apologize for the typographical mistake in Table 4. The correct Hamiltonian MAE for Naive Loss is 0.4912, not 0.0412. This corrected value aligns with both theoretical expectations and our experimental observations. We have carefully proofread the entire manuscript to ensure there are no other numerical inconsistencies. We appreciate your attention to detail, which helps maintain the rigor and clarity of our presentation.

---

> > ### Comment · Reviewer_YtXY · 2024-11-25
> >
> > Thank you for addressing my questions and I will increase the  Soundness score.

---

### Official Review · Reviewer_Rcb6 · 2024-11-03

**Soundness:** 4
**Presentation:** 3
**Contribution:** 4
**Rating:** 8
**Confidence:** 3

**Summary:**

This work presents a novel machine learning framework for predicting Kohn-Sham Hamiltonians on large-scale molecular systems. The proposed novel method includes a novel loss function WALoss based on basis change, and a novel SO(3)-equivariant model WANet adopting eSCN operation and MACE based many-body interaction in its architecture. A large-scale molecule dataset PubChemQH is also curated from PubChemQC benchmark for experiments. The proposed method is experimentally shown to achieve good performance on QH9 and PubChemQH benchmarks.

**Strengths:**

Originality:
The originality contribution of this work is excellent by proposing a physics-inspired loss function and model for Hamiltonians prediction problem. Also, the curated dataset is a significant originality contribution by providing a large-scale testbed in the field.

Quality:
The quality of this work is evidenced by solid theoretic analysis and strong performance on benchmark experiments.

Clarity:
The writing of this work is good, clear and well-organized.

Significance:
This work makes significant contribution in not only proposing an insightful machine learning model, but also having benefits in reducing computational cost for broad quantum physics and chemistry community.

**Weaknesses:**

(1) In Table 2, it is observed that if using WALoss on WANet, while Hamiltonian prediction performance improves, the prediction performance on the two quantum properties degrades. Could authors give explanations why this phenomenon happens on small-scale QH9 dataset but not on  large-scale PubChemQH dataset (Table 1)? Also, authors are encouraged to report the results of QHNet with WALoss on QH9 datasets.

(2) For practical application targets, both prediction accuracy and speed are important for an effective method. Authors are encouraged to give analysis about computational complexity or report inference speed of the proposed method, and compare them with QHNet [1].

[1] Haiyang Yu, Zhao Xu, Xiaofeng Qian, Xiaoning Qian, and Shuiwang Ji. 2023. Efficient and equivariant graph networks for predicting quantum hamiltonian. In Proceedings of the 40th International Conference on Machine Learning (ICML'23), Vol. 202. JMLR.org, Article 1692, 40412–40424.

**Questions:**

See Weaknesses part.

**Details Of Ethics Concerns:**

No ethics concerns.

---

> ### Author Response · Authors · 2024-11-20
>
> Thank you for your constructive comments and suggestions, they are exceedingly helpful for us to improve our paper. Our point-to-point responses to your comments are given below:
>
> > It is observed that if using WALoss on WANet, while Hamiltonian prediction performance improves, the prediction performance on the two quantum properties degrades. Could authors give explanations why this phenomenon happens on small-scale QH9 dataset but not on large-scale PubChemQH dataset (Table 1)?
> >
>
> We appreciate your observation and would like to clarify our findings. Our results in QH9 show that while the Hamiltonian MAE performance degrades, the prediction accuracy for the two quantum properties improves. This varying impact of WALoss across datasets underscores a key challenge in Hamiltonian prediction: developing an appropriate metric in the **matrix space that accurately correlates with the metrics of derived quantum properties, such as system energy, wave function**.
>
> In larger systems (i.e., PubChemQH), WALoss enhances optimization stability by providing physically meaningful guidance, leading to improvements across multiple properties. However, in smaller systems (i.e., QH9), the interaction between WALoss and traditional loss terms may result in local optima that prioritize quantum properties over Hamiltonian MAE. This scale-dependent behavior illustrates the complexity of multi-objective optimization in Hamiltonian prediction.
>
> Notably, the improved performance in larger systems is particularly significant, as these are more representative of real-world applications and have historically been more challenging to handle. This finding suggests that the benefits of WALoss for practical applications outweigh the trade-offs observed in smaller systems.
>
> > Authors are encouraged to report the results of QHNet with WALoss on QH9 datasets.
> >
>
> We appreciate your suggestion and have implemented it. The results for QHNet with WALoss on the QH9 dataset are now included in Table 2 of the revised manuscript. Our analysis indicates that incorporating WALoss leads to improved quantum property prediction performance on the QH9 dataset, demonstrating the effectiveness of our approach across various model architectures and datasets.
>
> > For practical application targets, both prediction accuracy and speed are important for an effective method. Authors are encouraged to give analysis about computational complexity or report inference speed of the proposed method, and compare them with QHNet [1].
> >
>
> We have expanded our efficiency analysis in Section 5.4 to provide a more comprehensive overview of WANet's performance. Our model demonstrates significant improvements in computational efficiency compared to QHNet. Specifically, WANet achieves an inference speed of 1.09 iterations per second—more than double QHNet's 0.45 iterations per second. The training time has been reduced by over 50%, from 90.43 hours for QHNet to 39.22 hours for WANet. Additionally, WANet requires significantly less memory, utilizing 15.86 GB of peak GPU memory compared to QHNet's 26.49 GB.
>
> These efficiency gains can be attributed to one key architectural innovation: our efficient SO(2) convolution design. This enhancement allows WANet to process large molecular systems more effectively while maintaining high accuracy. For a detailed breakdown of these performance metrics, please refer to Figure 3 in the paper.

---

> > ### Comment · Reviewer_Rcb6 · 2024-11-23
> > **Follow-up Response**
> >
> > I appreciate authors' efforts in rebuttal. All my concerns have been addressed so I will keep my rate.

---

### Public Comment · ~Haiyang_Yu6 · 2025-04-22
**Repo for the code and dataset**

Dear Authors,

Thank you for providing this brilliant work. I greatly appreciate the insights and contributions presented in this paper. If possible, would you mind kindly considering releasing the code and datasets associated with this work? Having access to them would be extremely helpful for further studies.

Thank you very much for your consideration.

Best regards,

Haiyang

---

> ### Public Comment · ~Yunyang_Li1 · 2025-04-23
>
> Hi Haiyang,
>
> Thank you so much your interests in our work! The code and the data will be released soon. Please stay tuned!

---

### Meta-Review · Area_Chair_uFs4 · 2024-12-19

**Metareview:**

The authors present several innovations which together enable the accelerated calculation of self-consistent solutions to the Kohn-Sham equations, which could accelerate DFT calculations significantly. All reviewers agreed that the paper was well written and presents a significant advance, and were unanimous in their recommendation to accept. Therefore I recommend the paper be accepted.

**Additional Comments On Reviewer Discussion:**

All reviewers engaged well with the authors and some decided to increase their score based on feedback. It seems that all major reviewer concerns were addressed.

---

### Decision · Program_Chairs · 2025-01-22

Accept (Spotlight)